# Phosphorylation of ELYS promotes its interaction with VAPB at decondensing chromosomes during mitosis

Christina James [1], Ulrike Möller[1], Christiane Spillner[1], Sabine König[2,3], Olexandr Dybkov [3], Henning Urlaub[2,3], Christof Lenz [2,3] & Ralph H Kehlenbach [1✉]

## Abstract

ELYS is a nucleoporin that localizes to the nuclear side of the nuclear pore complex (NPC) in interphase cells. In mitosis, it serves as an assembly platform that interacts with chromatin and then with nucleoporin subcomplexes to initiate post-mitotic NPC assembly. Here we identify ELYS as a major binding partner of the membrane protein VAPB during mitosis. In mitosis, ELYS becomes phosphorylated at many sites, including a predicted FFAT (two phenylalanines in an acidic tract) motif, which mediates interaction with the MSP (major sperm protein)-domain of VAPB. Binding assays using recombinant proteins or cell lysates and co-immunoprecipitation experiments show that VAPB binds the FFAT motif of ELYS in a phosphorylation-dependent manner. In anaphase, the two proteins co-localize to the non-core region of the newly forming nuclear envelope. Depletion of VAPB results in prolonged mitosis, slow progression from meta- to anaphase and in chromosome segregation defects. Together, our results suggest a role of VAPB in mitosis upon recruitment to or release from ELYS at the non-core region of the chromatin in a phosphorylation-dependent manner.

**Keywords** VAPB; ELYS; Mitosis; FFAT-motif; Nuclear Envelope
**Subject Categories** Cell Cycle; Membranes & Trafficking; Signal Transduction

## Introduction

The Vesicle-Associated-membrane Protein-associated protein B (VAPB) is a tail-anchored membrane protein with well-described functions at the endoplasmic reticulum (ER). It serves as a tethering factor that is involved in the formation of contact sites between the ER and other organelles, e.g., mitochondria (De Vos et al, 2012; Gómez-Suaga et al, 2019; Kim et al, 2022; Stoica et al, 2014), the Golgi-apparatus (Kuijpers et al, 2013) or peroxisomes (Costello et al, 2017; Kors et al, 2022). Besides its characteristic C-terminal transmembrane domain, VAPB contains a central coiled-coil domain (Nishimura et al, 1999). It is a member of a larger protein family (VAP-family), including VAPA, a protein closely related to VAPB, and also the Motile Sperm Domain-containing proteins 1, 2 and 3 (MOSPD1, MOSPD2 and MOSPD3 (Cabukusta et al, 2020; Di Mattia et al, 2018; Loewen and Levine, 2005)). All these proteins have a characteristic MSP (Major Sperm Protein) domain that is of particular importance for protein-protein interactions. The crystal structure of an MSP-domain of VAPA revealed seven immunoglobulin-like β-sheets (Kaiser et al, 2005; Shi et al, 2010). The domain interacts with characteristic peptide sequences of target proteins called FFAT-motifs (two phenylalanines in an acidic stretch) (Kaiser et al, 2005; Loewen and Levine, 2005), for example of the oxysterol-binding proteins (OSBPs) and of OSBP-related proteins (ORPs) which connect ER- and Golgi-membranes (Mesmin et al, 2013). Several variants of such motifs have been described: the conventional FFAT motif with its seven core residues (EFFDAxE) has an upstream acidic flanking region and binds in a characteristic manner to the MSP-domain where several amino acid residues are crucial for the interaction, for example VAPB K87 and M89 (Kaiser et al, 2005). FFAT-like motifs deviate from the conventional motif and are extremely diverse in their sequences, e.g., with a tyrosine- instead of a phenylalanine-residue (Murphy and Levine, 2016). Moreover, there are unconventional FFAT-related motifs with a neutral instead of an acidic tract (called FFNT, two phenylalanines in a neutal tract), which are favored by MOSPD1 and 3 (Cabukusta et al, 2020). Other prominent examples for protein-protein interaction partners of VAPB are the protein tyrosine phosphatase-interacting protein-51 (PTPIP51), connecting ER- and mitochondria (Yeo et al, 2021) and the acyl-CoA binding domain protein ACBD5 which forms a link between the ER and peroxisomes (Costello et al, 2017) and also interacts with VAPA (Hua et al, 2017). In general, members of the VAP-family are known to have many common interaction- or proximity partners with more than 250 proteins identified in proteomic screens (Huttlin et al, 2015; James and Kehlenbach, 2021; James et al, 2019; Murphy and Levine, 2016; Slee and Levine, 2019). The biological

[1]Department of Molecular Biology, Faculty of Medicine, GZMB, Georg-August-University Göttingen, Humboldtallee 23, 37073 Göttingen, Germany. [2]Bioanalytics Group, Institute of Clinical Chemistry, University Medical Center Göttingen, Robert-Koch-Straße 40, 37075 Göttingen, Germany. [3]Bioanalytical Mass Spectrometry Group, Max-Planck-Institute for Multidisciplinary Sciences, Am Fassberg 11, 37077 Göttingen, Germany. ✉E-mail: rkehlen@gwdg.de

significance of most of these interactions, however, has not been investigated in detail.

Both, the canonical FFAT- and the FFAT-like motifs can be regulated by phosphorylation. The negative charge that can be introduced by phosphorylation of serine or threonine residues can promote binding of the respective protein to the MSP-domain. Examples for proteins with such a motif are STARD3 (StAR-related lipid transfer domain-3), a protein that mediates cholesterol transport from the ER to endosomes (Di Mattia et al, 2020), PTPIP51, which tethers mitochondria to the ER (De Vos et al, 2012; Di Mattia et al, 2020) and CERT (ceramide transfer protein), which is required for the transfer of ceramide from the ER to the Golgi apparatus (Kumagai et al, 2014). For a more detailed discussion of the different types of FFAT-motifs see (Di Mattia et al, 2020; James and Kehlenbach, 2021; Murphy and Levine, 2016; Slee and Levine, 2019).

Using functional assays and immunoelectron microscopy, we (James et al, 2019) and others recently showed that VAPB (Saiz-Ros et al, 2019) and the yeast VAP homologue (Scs2) which plays a role in the regulation of SUMOylation (Brickner and Walter, 2004; Ptak et al, 2021; Saik et al, 2023; Saiz-Ros et al, 2019) not only localizes to the ER, but also reaches the inner nuclear membrane (INM). Proteins with a cytoplasmic region smaller than 60 kDa are thought to passively diffuse from the ER via the outer nuclear membrane (ONM) and the nuclear pore complexes (NPCs) to the INM (Zuleger et al, 2012). Hence, a portion of the cellular VAPB with its cytoplasmic domain of 24 kDa is expected to reach the INM and the question arises whether the protein fulfills a function at this localization. To address this question, we searched specifically for nuclear interaction/proximity partners of VAPB using an APEX2-based approach for the identification of potential binding partners. Our new method, RAPIDS (Rapamycin and APEX dependent Identification of proteins by SILAC), identified several membrane proteins of the INM, e.g., emerin and LAP1 (lamina-associated polypeptide 1, also known as Torsin-1A-interacting protein 1, TOR1AIP1). Another prominent hit we obtained was the nucleoporin ELYS (embryonic large molecule derived from yolk sac), also known as AHCTF1 (AT-hook containing transcription factor 1) and Mel28 in *C. elegans* (Galy et al, 2006). It was originally identified in mice as a large protein of 2243 amino acid residues (2266 in human cells), containing characteristic nuclear localization signals (NLSs) and nuclear export signals (NESs). Furthermore, an AT-hook DNA-binding domain could contribute to the nuclear localization of the protein (Kimura et al, 2002). Later, it was realized that ELYS is a component of the nuclear pore complex (NPC) and also localizes to kinetochores during mitosis (Rasala et al, 2006). At the level of the NPC, it interacts with the Nup107-160 complex, also known as the Y-complex (Galy et al, 2006; Rasala et al, 2006). Very detailed analyses of NPC-structures revealed that ELYS localizes to the nuclear side of the nuclear envelope (NE) (Bley et al, 2022). Importantly, ELYS is required for the assembly of post-mitotic NPCs (Franz et al, 2007; Galy et al, 2006; Gillespie et al, 2007; Rasala et al, 2006). Specifically, it binds to the decondensing chromosomes via its AT-hook and then recruits other NPC-components, e.g., other proteins of the Y-complex. This initial hub was then suggested to interact with membrane vesicles containing the integral NPC-membrane proteins POM121 and NDC1 (Rasala et al, 2008), followed by the recruitment of additional soluble nucleoporins. Additional proteins on mitotic membrane fragments that interact with ELYS and could be involved in NE-assembly and/or promote post-mitotic NPC formation include the lamin B receptor (LBR; (Clever et al, 2012)) and the reticulon-like protein REEP4 (Golchoubian et al, 2022). In summary, a picture emerges where ELYS serves as the crucial initiation factor for post-mitotic NPC-assembly (for review see (Kutay et al, 2021)). Remarkably, ELYS is not required for NPC-biogenesis during interphase (Hampoelz et al, 2019; Vollmer et al, 2015).

In this study, we first aimed to identify proteins that specifically interact with the MSP-domain of VAPB and compared lysates of asynchronous and of mitotic cells as a source of potential interaction partners. ELYS turned out to be a major binding partner of VAPB during mitosis. Furthermore, our results show that the MSP-domain of VAPB interacts with FFAT-like motifs of ELYS in a phosphorylation-dependent manner. A wild-type fragment of ELYS had a profound effect on the cell cycle which was altered by single-point mutations in an FFAT motif that affected VAPB-binding. VAPB localizes to chromatin, in particular to the non-core region of the segregating chromosomes, together with ELYS, and depletion of VAPB resulted in prolonged mitosis and lagging chromosomes. Together our findings suggest that the ELYS-VAPB interaction is regulated in mitosis where ELYS recruits VAPB to the reforming NE in a phosphorylation-dependent manner.

## Results

### ELYS is a major interaction partner of VAPB in mitosis

Many of the known interaction partners of VAPB are found at contact sites between organelles during interphase. To the best of our knowledge, the fate of such interactions during mitosis has not been investigated at a larger scale. We therefore immobilized the MSP-domain of VAPB fused to a GST-tag (or GST alone as a control) on beads and used lysates from asynchronous and from mitotic HeLa cells as a source of potential binding partners. As shown in Fig. 1A, the pattern of interacting proteins clearly differed between the two lysates, with a most prominent protein with an apparent molecular weight larger than 250 kDa that was strongly enriched from the mitotic lysate. For a thorough quantitative analysis, proteins eluted from the beads were then subjected to data-independent acquisition mass spectrometry (DIA-MS). As a first quality control of our approach, we compared proteins from asynchronous cells that interacted with GST or GST-MSP. Indeed, a large number of proteins (284; in red) specifically interacted with GST-MSP, including many of the known binding partners of VAPB like ACBD5 and several oxysterol binding proteins (Fig. 1B). An even larger number (427 proteins) was identified from a lysate from cells synchronized in mitosis, with 230 proteins found under both conditions (Fig. 1D; see also Fig. EV1A). Next, we directly compared proteins from the two types of lysates with respect to binding to immobilized GST-MSP. As shown in Fig. 1C, many proteins (>160; in red) from the mitotic lysate were clearly enriched compared to proteins from the lysate derived from asynchronous cells. Several of the well-known binding partners of VAPB, by contrast, rather bound to GST-MSP when derived from asynchronous cells, e.g., ACBD5 and RMDN2 (Di Mattia et al, 2020), a

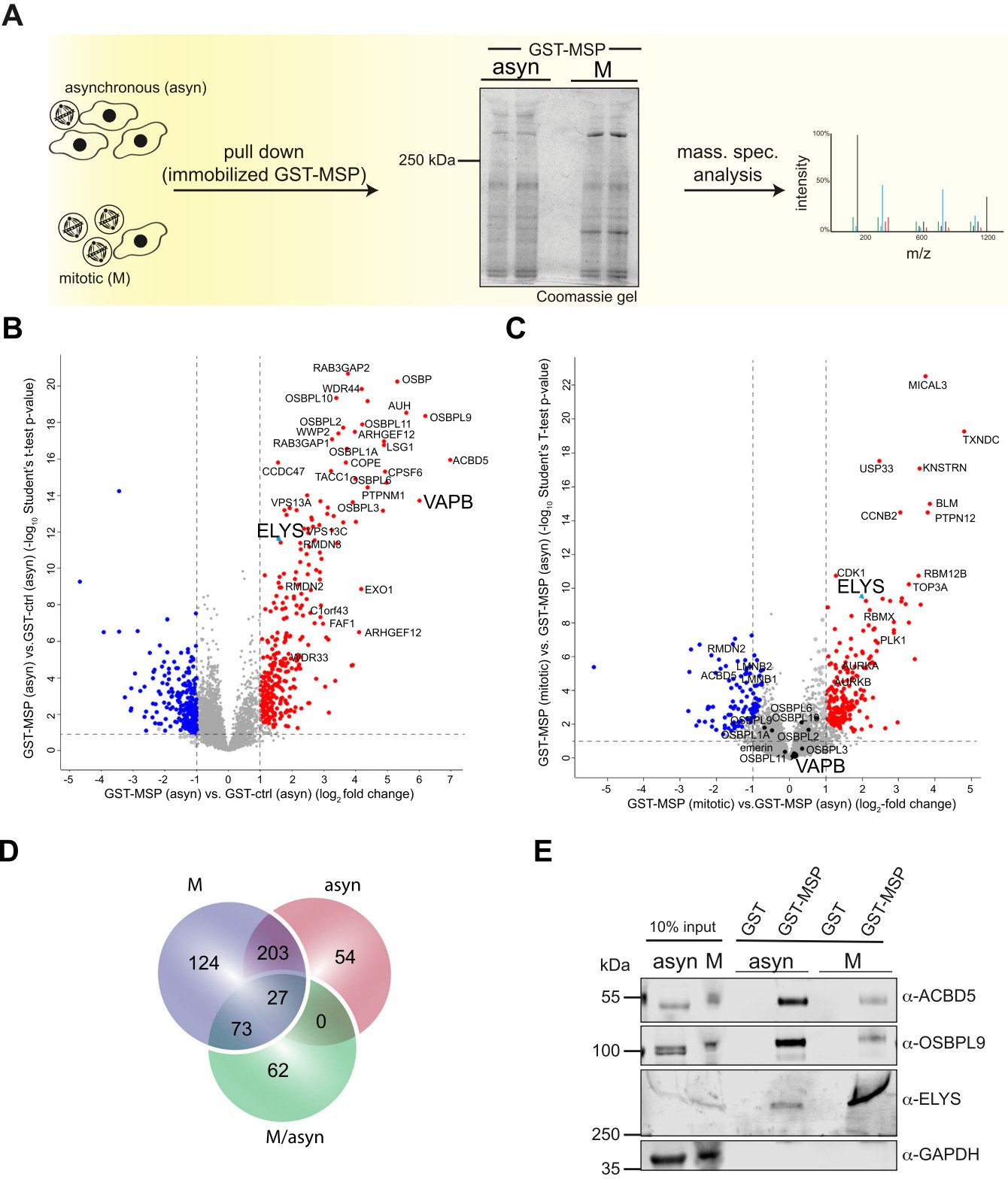

Figure 1.  ELYS is a major interaction partner of VAPB in mitosis.

(A) HeLa cells were synchronized using DME and cellular lysates obtained from asynchronous (asyn) and mitotic cells (M) were incubated with GST or GST-MSP immobilized on GST-selector beads. The Coomassie gel shows two lanes each of proteins that had been eluted from the beads with SDS-sample buffer. Similar samples were also analyzed by MS-DIA. (B) MS-DIA-results comparing proteins from asynchronous cells bound to GST or GST-MSP. The graph shows the combined results of four independent experiments with two technical replicates each. A two-sided Student's T-test was performed using normalized $\log_2$ ratios. For the analysis, a Permutation-based FDR was applied and a threshold value of 0.05 was chosen. Proteins enriched on GST-MSP beads are depicted in red. (C) MS-DIA-results comparing proteins from asynchronous and mitotic cells bound to GST-MSP. The graph shows the combined results of four independent experiments with two technical replicates each. A two-sided Student's T-test was performed using normalized $\log_2$ ratios. For the analysis, a Permutation-based FDR was applied and a threshold value of 0.05 was chosen. Proteins from mitotic lysates that are enriched on GST-MSP beads are depicted in red. (D) Venn diagram summarizing the results of different analyses: proteins from mitotic cells (M, blue circle; compare Fig. EV1) or asynchronous (asyn, red circle; as in (B)) specifically enriched on GST-MSP- versus GST-beads. Green circle, proteins from mitotic cells enriched on GST-MSP-beads compared to proteins from asynchronous cells; as in (C)). Individual proteins are also listed in Datasets EV1–4. (E) Cell lysates (10% input) and proteins eluted from GST-beads were obtained as in (A) and analyzed by SDS-PAGE followed by Western blotting, detecting ACBD5, OSBPL9, and ELYS. GAPDH was used as loading control. Source data are available online for this figure.

protein that is homologous to PTPIP51 (Fig. 1C, proteins in blue). The nucleoporin ELYS was one of the proteins that was found to be enriched on GST-MSP beads under all three conditions (i.e., on GST-MSP vs. GST incubated with the two types of lysates and, strikingly, on GST-MSP incubated with the mitotic lysate vs. the lysate derived from asynchronous cells; see Fig. 1D). ELYS is the only protein of this group that we had previously identified as a nuclear proximity partner of VAPB using RAPIDS (James et al, 2019). Indeed, we could demonstrate enhanced binding of ELYS to GST-MSP from mitotic lysates also by Western-blotting in both HeLa and Hek293T cells (Figs. 1E and EV1B), suggesting that the large protein (> 250 kDa) seen in Fig. 1A is in fact ELYS. This was also confirmed by mass spectrometry upon cutting the corresponding band from the Coomassie gel. In contrast to ELYS, ACBD5 and OSBPL9 were preferentially detected as binding partners of immobilized GST-MSP when derived from lysates of asynchronous cells (Fig. 1E). Together, these results show that ELYS interacts specifically with the MSP-domain of VAPB and does so in particular during mitosis.

## Phosphorylation of ELYS regulates VAPB binding

We next investigated binding of endogenous full-length ELYS to overexpressed or endogenous VAPB under different conditions. First, we generated HeLa cells stably expressing GFP-VAPB in a tetracycline-inducible manner. Cells were enriched in the G1/S-phase of the cell cycle or in mitosis, or left unsynchronized (asyn) and subjected to a co-immunoprecipitation protocol using nanobodies (GFP-Selector). In the input, ELYS from the mitotic lysate migrated as two distinct bands on the SDS-gel (Fig. 2A), suggesting a post-translational modification of the protein. Compared to the input fraction, a large proportion of GFP-VAPB was precipitated under our experimental conditions. ELYS could not be detected as a coprecipitating protein when the lysate from asynchronous cells was used. Lysates from G1/S-cells and from mitotic cells, by contrast, yielded a small amount of co-precipitating ELYS (Fig. 2A,B). ELYS was not detected when the cells had not been induced with tetracycline to express GFP-VAPB. Next, we performed an immunoprecipitation experiment using antibodies against endogenous VAPB, now using normal HeLa cell lysates from asynchronous and mitotic cells. Similar to the results in Fig. 2A, the signal of co-precipitating ELYS was more pronounced for the mitotic lysate (Fig. 2C). When purified IgG was used as a control, only background levels of ELYS were detected. In a next step, we analyzed binding of endogenous ELYS

to immobilized versions of the MSP-domain of VAPB, purified from bacteria as GST-fusion proteins. As shown in Fig. 2D (see also Fig. 1E), strong binding of ELYS to wild-type GST-VAPB-MSP was observed when a mitotic lysate was used. Binding was much weaker with a lysate from asynchronous cells and no binding to immobilized GST was detected. As a specificity control, we used MSP-K87D/M89D (KD/MD), an established mutant of VAPB that abolishes the interaction with FFAT-motifs (Kaiser et al, 2005). Importantly, the mutations do not affect folding and do not lead to protein aggregation (Kaiser et al, 2005). As expected for a typical MSP-based interaction, binding of ELYS (from the mitotic lysate) was abolished when the KD/MD-version of GST-VAPB-MSP was used as an immobilized protein.

Many nucleoporins are extensively phosphorylated at the onset or during mitosis (Blethrow et al, 2008; Glavy et al, 2007; Macaulay et al, 1995). Although multiple phosphorylation sites have been identified in ELYS (Hornbeck et al, 2015; Sharma et al, 2014; Ullah et al, 2016), potential consequences of ELYS-phosphorylation have not been investigated. Hence, we directly addressed the question of a possible phosphorylation-dependent interaction of VAPB and ELYS. We first treated our lysates with λ-phosphatase to dephosphorylate cellular proteins. Indeed, this treatment resulted in an increased electrophoretic mobility of ELYS in the mitotic lysate (Fig. 2E; input). Phosphorylation of ELYS could also be confirmed using Phos-tag gels (see Appendix Fig. S1). Lysates that had been treated with or without λ-phosphatase were then incubated with immobilized GST-VAPB-MSP. Both, the weak interaction seen with ELYS from a lysate from asynchronous cells and the much stronger interaction seen with ELYS from a mitotic lysate was abolished upon λ-phosphatase treatment (Fig. 2E,F). Together, these results show that VAPB, via its MSP-domain, and ELYS interact with each other most prominently during mitosis and that this interaction can be regulated by phosphorylation.

## FFAT motifs of ELYS interact with VAPB

ELYS was identified previously as a potential interacting partner of VAPs by us (James et al, 2019) and by others (Huttlin et al, 2015; Murphy and Levine, 2016). We now decided to investigate the interaction of VAPB and ELYS in detail. First, we analyzed ELYS for potential FFAT motifs that could bind to the MSP-domain of VAPB. Three FFAT motifs were predicted in ELYS based on a position weight matrix strategy to identify short linear motifs (Di Mattia et al, 2020; Murphy and Levine, 2016; Slee and Levine, 2019), all of which lack the hydrophobic phenylalanine residues at

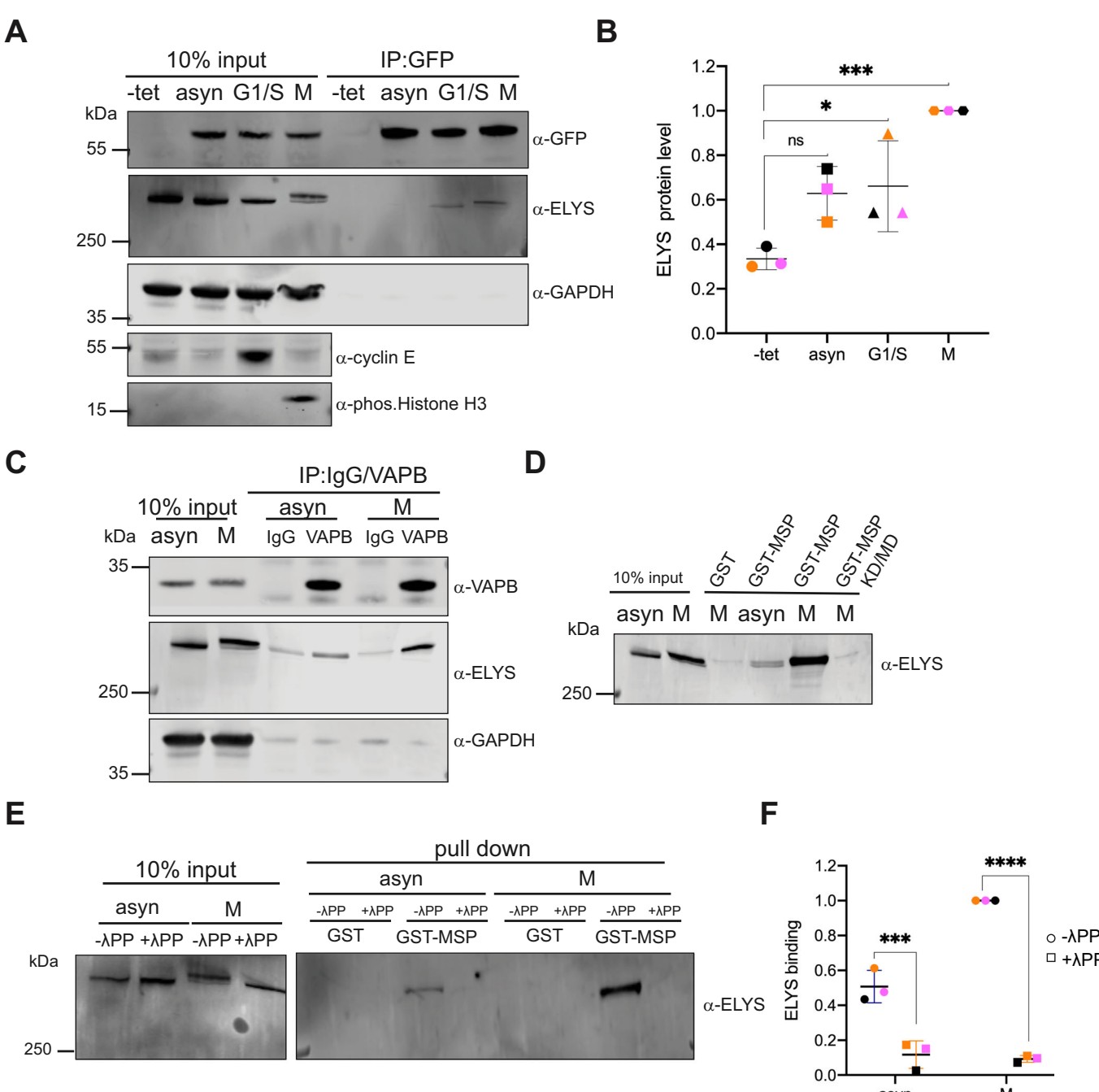

position one but all have an aromatic residue (tyrosine or phenylalanine) at position two (Loewen et al, 2003). FFAT-1 lies in the α-solenoid domain of ELYS, whereas FFAT-2 and FFAT-3 are found in its C-terminal region, which is predicted to be unstructured (Fig. 3A). On first inspection, FFAT-2, largely corresponds to a conventional FFAT motif with an aspartic residue in the fourth position of the core. FFAT-1 and FFAT-3, on the other hand, resemble phospho-FFAT motifs, as they contain a serine or threonine residue, respectively, at the fourth position. These residues could be phosphorylated to promote binding to VAPB (Di Mattia et al, 2020). In order to determine whether the three FFAT-motifs are indeed interaction sites for VAPB, we used a

peptide binding approach (Di Mattia et al, 2020). Biotinylated peptides (Fig. 3B) were immobilized on beads and incubated with a total HeLa-cell lysate. For FFAT-1 and FFAT-3, phosphorylated and non-phosphorylated versions were used ($pS_{825}$ and $pT_{1717}$, respectively). A random peptide (Di Mattia et al, 2020) was used as negative control. As shown in Fig. 3C, strong binding of VAPB was detected for the FFAT-2 peptide. Much weaker binding was observed for the non-phosphorylated FFAT-1 peptide, whereas the phosphorylated version showed somewhat stronger binding. No binding of VAPB to the FFAT-3 peptides and the control peptide was observed. Next, we performed similar binding assays using immobilized peptides and GST-tagged versions of the MSP-domain

**Figure 2. Binding of VAPB to ELYS is regulated by phosphorylation.**

(A) HeLa Flp-In T-REx cells with stable expression of GFP-VAPB were treated with or without (-tet) tetracycline to induce expression. Cells were then synchronized either in the G1/S phase by a double thymidine block or in mitosis (M) by treatment with DME. Total cell lysates obtained from asynchronous (asyn) or synchronized (G1/S and M) cells were subjected to co-immunoprecipitation (IP) using the GFP-Selector. Cell synchronization at G1/S phase and mitosis was confirmed by staining for Cyclin E and phospho-Histone H3, respectively. (B) Quantification of ELYS protein levels in the bound fraction after immunoprecipitation from three independent experiments as in (A) (symbols in black, orange, and pink). Maximum protein levels coprecipitated from mitotic lysates were assigned an arbitrary value of 1. The data are shown as mean ± standard deviation ($n = 3$; biological replicates). ***$p = 0.0004$, *$p < 0.0323$, ns, non-significant, $p = 0.0527$ (Bonferroni's multiple comparisons test). (C) HeLa cells were treated as in A to obtain asynchronous or mitotic lysates, which were subjected to co-IP using antibodies against VAPB or purified IgG as a control. (A, B) Precipitated proteins were analyzed by SDS-PAGE, followed by Western blotting using antibodies against GFP, ELYS, VAPB, and GAPDH as indicated. Note the slower migrating form of ELYS in mitotic lysates. (D) Purified proteins (GST, the wild type MSP-domain of VAPB (GST-MSP) and the KD/MD mutant of MSP (GST-MSP-KD/MD) were immobilized on GST-Selector beads and incubated with lysates from asynchronous (asyn) or mitotic (M) HeLa cells. Interacting proteins were analyzed by SDS-PAGE, followed by Western blotting using antibodies against ELYS. (E) Lysates from asynchronous (asyn) or mitotic (M) HeLa cells were treated with (+) or without (-) λ-phosphatase (λPP) and incubated with purified proteins (GST or GST-MSP) bound to GST Selector agarose beads. Interacting proteins were analyzed by SDS-PAGE, followed by Western blotting using antibodies against ELYS. (F) Quantification of ELYS binding to GST-MSP with or without λPP treatment normalized to the input protein levels from three independent experiments as in (E) (symbols in black, orange, and pink). The data are shown as the mean (± standard deviation) of three biological replicates. ****$p < 0.0001$, ***$p = 0.0001$ (Sidak's multiple comparisons test). Source data are available online for this figure.

of VAPB instead of cellular lysates. As described above, the MSP-K87D/M89D (KD/MD) mutant of VAPB abolishes the interaction with FFAT-motifs (Kaiser et al, 2005). MSP-K43L, on the other hand, was reported as a mutant of VAPB that abrogates binding to phospho-FFAT motifs, but not to conventional FFAT-motifs (Di Mattia et al, 2020). As an additional control, we also used a peptide derived from ORP1 (also known as OSBPL1A), a protein with a strong FFAT motif (Kaiser et al, 2005; Murphy and Levine, 2016). Similar to the results shown in Fig. 3C, binding of the wild-type MSP-domain of VAPB (GST-MSP) was stronger to the phosphorylated compared to the non-phosphorylated version of FFAT-1 (Fig. 3D). Strong binding was also observed for the FFAT-2 peptide, whereas FFAT-3 showed only background binding. As expected, the KD/MD mutation in the MSP-domain largely abrogated binding to FFAT-1 and FFAT-2 peptides and also to the ORP1-peptide. Surprisingly, the K43L-mutation clearly affected the interaction of GST-MSP with the strong FFAT motif of ORP1 as well. Accordingly, the K43L-mutation also affected binding of GST-MSP to the FFAT-1 and FFAT-2 peptides. Importantly, no binding of purified GST to any of the immobilized peptides was detected. Together, these data show that VAPB interacts with ELYS-peptides via its MSP-domain and that the interaction can be mediated by different types of FFAT-motifs, which may in part be regulated by phosphorylation.

## FFAT-2 of ELYS is regulated by phosphorylation

In light of the strong dependency of VAPB-binding to ELYS on phosphorylation, we decided to reinvestigate the phosphorylation state and potential phosphorylation sites of ELYS. HeLa cell lysates from asynchronous or mitotic cells were incubated with GST-VAPB-MSP immobilized on beads, as shown in Fig. 1A. As before, proteins were then subjected to mass spectrometry for the identification of phosphorylated peptides. Figure 4A shows the phosphorylation sites on ELYS identified by our approach (34 sites), and also sites that had previously been reported by others (Hornbeck et al, 2015; Sharma et al, 2014; Ullah et al, 2016). Despite a total of 156 identified phosphorylation sites, our candidate phospho-FFAT-motif (FFAT-1) was not identified as a hit. FFAT-2, however, is close to a phospho-cluster and also contains two serine residues that can be phosphorylated, serine 1314 and serine 1326 (Ullah et al, 2016). We therefore decided to look in more detail into potential effects on VAPB-binding,

depending on the phosphorylation status of these sites. Following the approach described in Fig. 3, several biotinylated versions of the FFAT-2 peptide were synthesized: the non-phosphorylated version (as in Fig. 3), two phosphorylated versions with phospho-serine residues at positions 1314 and 1326, respectively, a double-phospho-version with both residues phosphorylated and one version where both serines are changed to alanines (Fig. 4B). Again, peptides immobilized on beads were incubated with a total-cell lysate and interacting proteins were analyzed. Figure 4C,D shows that phosphorylation of serine 1314, which is upstream of the core region of the FFAT motif, strongly promotes binding of VAPB to the peptide. Phosphorylation of the core residue serine 1326, by contrast, slightly reduced binding of VAPB, in particular when serine 1314 was also phosphorylated. Likewise, the alanine-containing peptides allowed only very little binding of endogenous VAPB. In a next step, we analyzed binding of purified proteins to the immobilized peptides, using GST and the wild-type- and the KD/MD-version of the MSP-domain of VAPB as potential interaction partners. Similar to the results described above, phosphorylation of serine 1314, but not serine 1326 of the FFAT-2 motif enhanced binding of the wild-type MSP-fusion protein to the peptide (Fig. 4E,F). Only background binding was observed when the mutant version of the VAPB-MSP domain (KD/MD) was used, demonstrating the specificity of the interaction.

Next, we investigated binding of the MSP-domain of VAPB to larger fragments of ELYS containing the wild-type FFAT-2 sequence or mutant (S1314A, S1314D, S1326A, S1326D) versions thereof. ELYS$_{1018-1642}$-fragments containing N-terminal His- and C-terminal MBP-tags (Fig. 5A) were expressed in bacteria and immobilized on beads via the MBP-tags. As shown in Fig. 5B,C, binding of the MSP-domain of VAPB (GST-MSP-VAPB) above background was observed when the phosphomimetic S1314D-mutant of the ELYS-fragment had been immobilized, suggesting that the negative charge of the aspartic acid residue promoted the interaction.

To address possible interactions inside cells, we generated inducible cell lines expressing GFP-tagged versions of the same ELYS-fragment (1018–1642; Fig. 5D).

Cells were enriched in mitosis and lysates were subjected to co-immunoprecipitation experiments using anti-GFP-nanobodies (GFP Selector). As shown in Fig. 5E,F, somewhat higher levels of coprecipitating endogenous VAPB were observed when the cell line expressing the ELYS-fragment with the phosphomimetic mutation

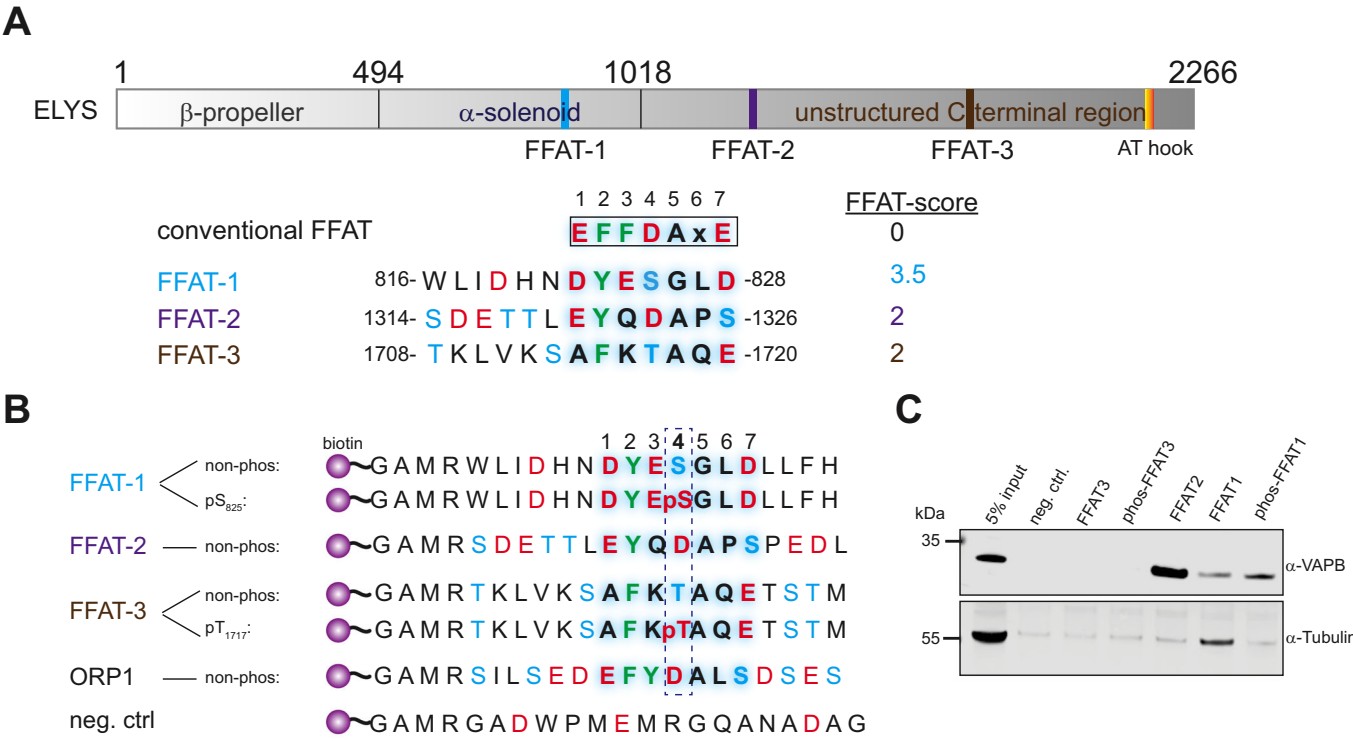

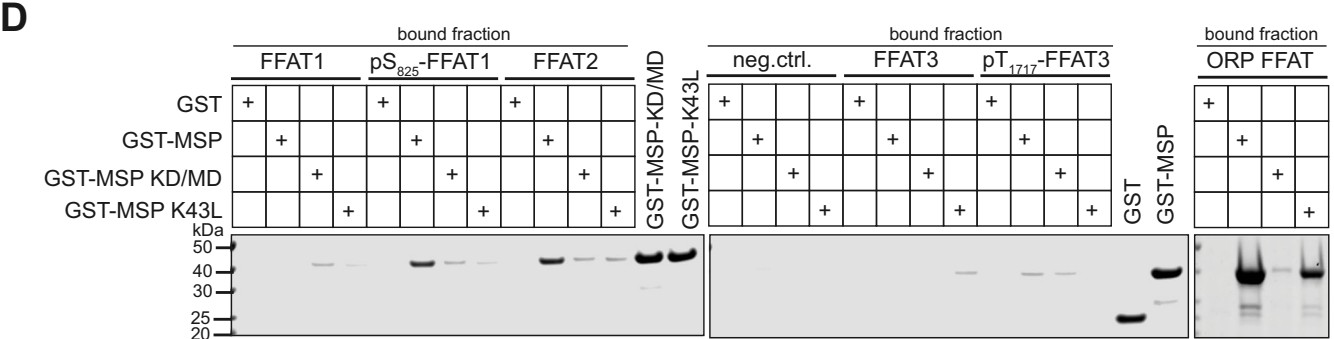

**Figure 3. VAPB-ELYS interactions are mediated by FFAT- and phospho-FFAT motifs.**

(A) Schematic representation of the domain structure of ELYS. The predicted FFAT motifs are shown in blue, purple and brown with the respective sequences and FFAT scores (the lower the score, the more similar it is to the conventional FFAT motif) below (with six acidic tract residues and seven core residues). The AT-hook is shown in orange. (B) Sequences of ELYS-, OPR1- and negative control peptides used for pull down assays. Peptides contain biotin at the N-terminus, a linker sequence (GAMR) and the FFAT sequences of ELYS (FFAT-1 and FFAT-3, either with or without phosphorylated serine- (pS) or threonine (pT) residue at position 4 of the core motif, and non-phosphorylated FFAT-2). The ORP1-peptide (amino acid residues 469-485) was used as a positive and a random peptide as a negative control. (C) Peptides as in B were immobilized on beads and incubated with total HeLa-cell lysate. Interacting proteins were analyzed by SDS-PAGE, followed by Western blotting using antibodies against VAPB or tubulin. (D) Peptides as in B were immobilized on beads and incubated with purified proteins (GST, the wild type MSP-domain of VAPB (GST-MSP), the KD/MD mutant (GST-MSP-KD/MD) or the K43L mutant of the MSP-domain (GST-MSP-K43L). Interacting proteins were analyzed by SDS-PAGE and Coomassie-staining. 10% of the input of the recombinant proteins were loaded as indicated. Source data are available online for this figure.

S1314D was used. We also noted that the mutant fragment containing alanine at position 1314, which cannot be phosphorylated, yielded lower levels of co-precipitating VAPB than the S1314D-mutant.

Together, these results suggest that phosphorylation at S1314 of FFAT-2, as it may occur in particular during mitosis, can control binding of VAPB to ELYS. To what extent phosphorylation of the many other phospho-sites in ELYS contribute to the phosphorylation-dependent interaction of the two proteins

(Figs. 2 and 4) remains to be investigated. Unfortunately, we did not manage so far to obtain cell lines expressing full-length GFP-ELYS (or mutant versions thereof).

## VAPB co-localizes with ELYS in anaphase

We have previously shown that ELYS and VAPB are found in close proximity at the INM of interphase cells (James et al, 2019). ELYS is known to localize to kinetochores in metaphase (Rasala et al,

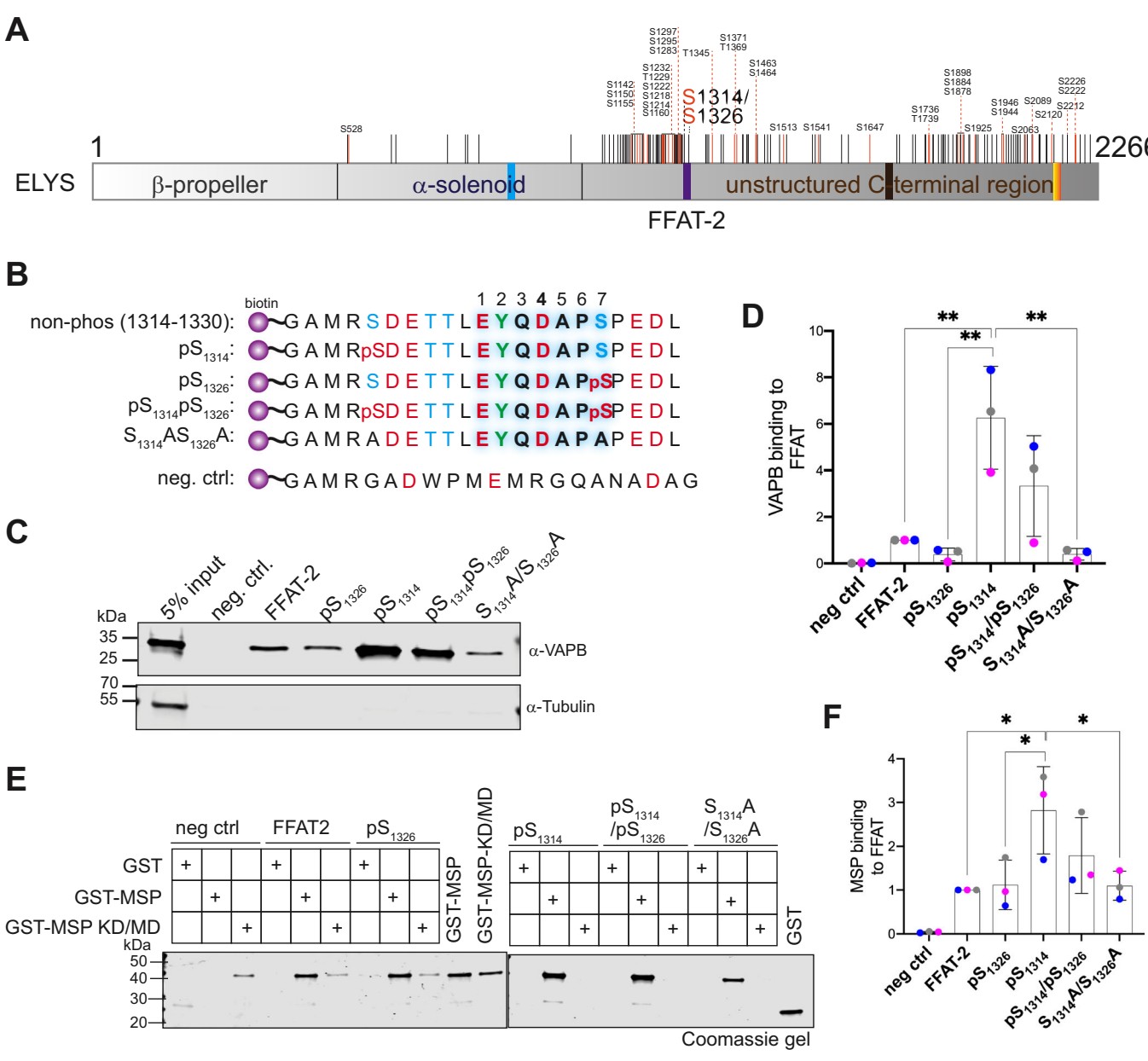

**Figure 4. Phosphorylation of serine 1314 in the acidic tract of FFAT-2 enhances the interaction of ELYS with VAPB.**

(A) Lysates from asynchronous and mitotic HeLa cells were subjected to pull down reactions using the MSP-domain of VAPB bound to GST Selector (compare Figs. 1E and 2D). The bound proteins were subjected to in-solution digestion with trypsin (Hughes et al, 2019) and analyzed by mass spectrometry to identify phosphorylation sites. Phosphorylation sites identified by mass spectrometry (orange lines; see also Dataset EV5) as well as sites that had been identified before (Hornbeck et al, 2015; Sharma et al, 2014; Ullah et al, 2016); black lines) are indicated on the ELYS protein. Two phospho-serine sites (S1314 and S1326) are within the FFAT-2-motif. (B) Sequences of the ELYS-peptides used for pull down assays. Peptides contain biotin at the N-terminus, a linker sequence and the FFAT-2 motif either with or without phosphorylated serine-residues at position 1314 and/or 1326 or alanine residues at these positions. A random peptide served as negative control. (C) Peptides as in C were immobilized and incubated with total HeLa cell lysates. Bound proteins were analyzed by SDS-PAGE, followed by Western blotting using antibodies against VAPB and tubulin. (D) Quantification of experiments as in (C) (dots in gray, blue, and magenta). VAPB-binding to FFAT-peptides was normalized to non-phosphorylated FFAT-2. The data are shown as the mean (± standard deviation) of three independent biological replicates. **$p = 0.0030$, **$p = 0.0012$, **$p = 0.0012$ ($p$ values from left to right; Tukey's multiple comparisons test). (E) Peptides as in C were immobilized and incubated with GST or GST-tagged wild type (GST-MSP) or the mutant version (GST-MSP KD/MD) of the MSP-domain. Bound proteins were analyzed by SDS-PAGE and Coomassie staining. The input fractions are loaded as indicated. (F) Quantification of three independent experiments as in (E) (dots in gray, blue, and magenta). GST-MSP-binding to FFAT-peptides was normalized to non-phosphorylated FFAT-2. The data are shown as the mean (± standard deviation) of three biological replicates. *$p = 0.0276$, *$p = 0.0413$, *$p = 0.0388$ ($p$ values from left to right; Tukey's multiple comparisons test). Source data are available online for this figure.

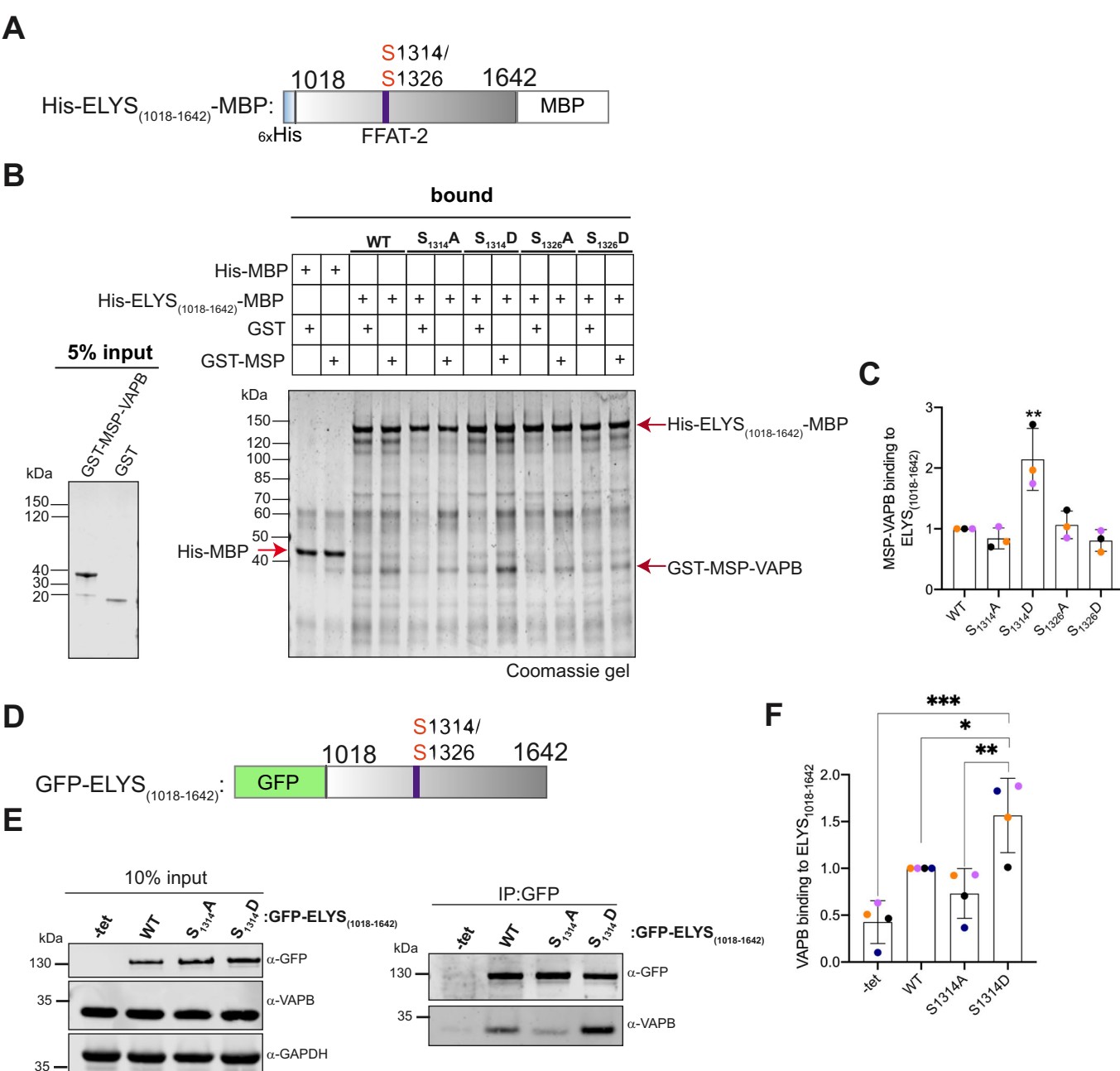

**Figure 5. Interaction of ELYS-fragments with VAPB.**

(A) Scheme of the fragments His-ELYS$_{(1018-1642)}$-MBP, purified from bacteria and used in (B) and (C). (B) Different versions of His-ELYS$_{(1018-1642)}$-MBP (wild-type (WT), S1314A, S1314D, S1326A, and S1326D) were immobilized on MBP-selector beads and incubated with GST-MSP-VAPB or GST as a control. Bound proteins were eluted with SDS-sample buffer and analyzed by SDS-PAGE, followed by Coomassie staining. (C) Quantification of three independent experiments (dots in black, orange, and pink) as in (B). MSP-VAPB binding to different ELYS$_{(1018-1642)}$ versions was normalized to the WT version. The data are shown as the mean (± standard deviation) of the three biological replicates. **$p = 0.0032$ (Holm-Sidak's multiple comparisons test). (D) Scheme of the fragments GFP-ELYS$_{(1018-1642)}$ used for transfection experiments as in (E) and (F). (E) Stable cell lines expressing different versions of GFP-ELYS$_{(1018-1642)}$ were treated with or without (-tet, for wild-type (WT) tetracycline and synchronized with DME to accumulate cells in mitosis. Cell lysates were subjected to co-immunoprecipitation experiments using GFP-selector beads. Proteins in the input lysates (left) and proteins eluted from the beads were analyzed by SDS-PAGE, followed by Western blotting detecting GFP and VAPB. GAPDH was used as loading control. (F) Quantification of four independent experiments (dots in black, blue, orange, and pink) as in (E). VAPB binding to different ELYS$_{(1018-1642)}$ versions was normalized to the WT version. The data are shown as the mean (± standard deviation) of four biological replicates. *$p = 0.0457$; **$p = 0.0038$; ***$p = 0.0003$ (Tukey's multiple comparisons test). Source data are available online for this figure.

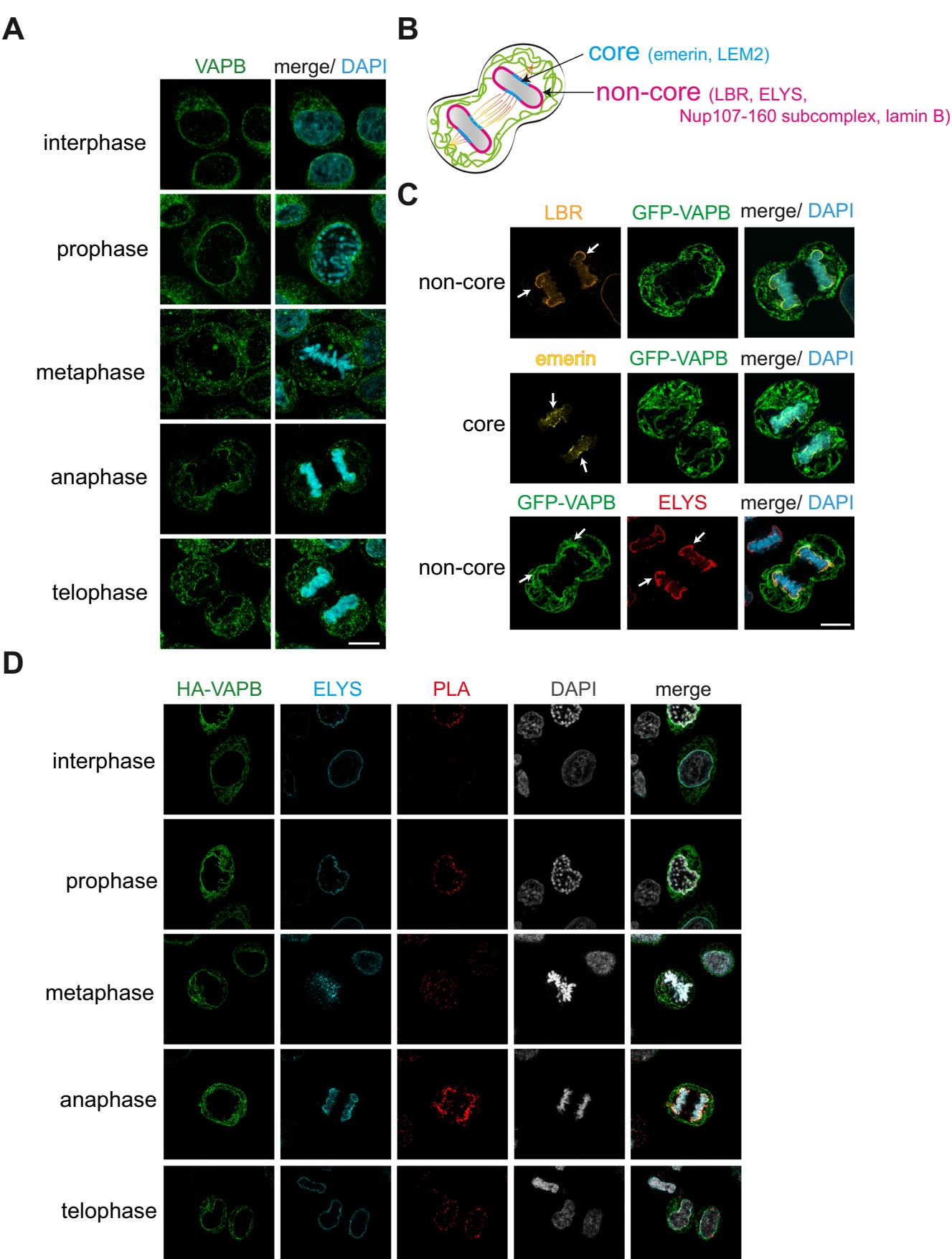

◄ **Figure 6.  VAPB localizes to the non-core regions on mitotic chromosomes.**

(A) Intracellular localization of VAPB in interphase, prophase, metaphase, anaphase, and telophase. HeLa P4 cells were synchronized by double thymidine block and released for 9 h before fixing and staining with antibodies against VAPB. DNA was stained with DAPI and cells were analyzed by confocal microscopy. Scale bar, 10 µm. (B) Schematic representation of anaphase cell with core (blue) and non-core (magenta) nuclear envelope (NE) subdomains. Green lines indicate ER-membranes. (C) HeLa Flp-In T-REx cells stably expressing GFP-VAPB were synchronized and released as in (A) and subjected to indirect immunofluorescence staining using antibodies against LBR, emerin, and ELYS. DNA was stained with DAPI and cells were analyzed by confocal microscopy. Scale bar, 10 µm. Arrows indicate the core and non-core NE subdomains, respectively. (D) HeLa Flp-In T-REx cells stably expressing HA-VAPB were synchronized and released from thymidine block as described in (A) and subjected to PLAs using antibodies against HA and ELYS. HA-VAPB and ELYS were detected by indirect immunofluorescence. Intranuclear signals as observed for example in prophase could derive from extended ER-cisternae (Lu et al, 2011). DNA was stained with DAPI and cells were analyzed by confocal microscopy. Scale bar, 10 µm. Source data are available online for this figure.

2006) and then, later in mitosis, to the decondensing chromosomes where it functions as an interaction platform for the formation of post-mitotic NPCs (Franz et al, 2007; Hampoelz et al, 2019). The localization of VAPB in mitosis has, to the best of our knowledge, not been analyzed before. We first used polyclonal antibodies against endogenous VAPB to detect the protein during different stages of mitosis in HeLa cells. In interphase and in early prophase, VAPB was mainly found at the level of the ER and also at the NE (Fig. 6A), as shown before. In metaphase, VAPB was largely distributed over the cells and also detected at the extended ER-cisternae (Lu et al, 2011), but was clearly absent from the chromatin region where the mitotic spindle is thought to create a zone that excludes organelles and membrane structures from this area of the cell (Lu et al, 2009; Smyth et al, 2012). In anaphase and telophase, the protein was then detected adjacent to the chromatin, suggesting that it is present at the newly forming NE. In these stages of mitosis, the spindle microtubules are in contact with the "core region" of the segregating chromosomes, which also has a characteristic protein composition (Fig. 6B; for review see (Liu and Pellman, 2020)). ELYS and a number of other proteins, by contrast, are known to preferentially localize to the "non-core region" of the chromosomes. For a detailed analysis of the localization of VAPB at mitotic chromosomes, we made use of inducible cell lines expressing HA- or GFP-tagged version of VAPB. As shown in Fig. 6C, LBR was found in non-core regions of segregating chromosomes during anaphase, whereas emerin was detected mainly in core regions, as described previously (Haraguchi et al, 2000; Lee et al, 2001). For GFP-VAPB, we found a clear colocalization with ELYS at the non-core region. In a next step, we further characterized a possible interaction of ELYS and VAPB during mitosis using proximity ligation assays (PLAs (Soderberg et al, 2006)). As shown in Fig. 6D, we could detect the strongest PLA-signals for the ELYS-HA-VAPB pair in anaphase cells, whereas other mitotic stages and interphase cells yielded much lower signals. The specificities of PLA signals and the antibodies used were supported by knocking down either VAPB or ELYS (Fig. EV2A,B) and also by single antibody controls (EV3).

In PLAs, tagged versions of a protein of interest offer the possibility to compare wild-type proteins with proteins containing pertinent mutations at key residues. As a control to show the feasibility of such an approach, we compared PLA-signals resulting from interactions of HA-VAPB with its established MSP-dependent binding partner OSBPL9 in interphase cells. Indeed, the wild-type version of HA-VAPB yielded many more PLA-interactions at the Golgi (James et al, 2019) than HA-VAPB-KD/MD which is expected to show reduced binding to OSBPL9 (Fig. 7A,B, top panels). In anaphase cells, by contrast, we observed

less overall PLA signals and no specific ones at the core or the non-core region of the chromosomes. Furthermore, no difference was detected between the wild-type and the mutant version of HA-VAPB, suggesting that the PLA under this specific condition effectively monitors *proximity* rather than a direct interaction. As shown above (Figs. 2D, 3D and 4E), the KD/MD-mutation in the MSP-domain of VAPB also largely abolishes interaction with ELYS. We therefore compared PLA-signals around the chromosome masses in anaphase cells expressing either wild-type HA-VAPB or the mutant HA-VAPB-KD/MD. We did not observe a significant difference in the localization of both VAPB versions in mitotic cells. However, the mutant protein yielded less PLA-interactions with ELYS compared to wild-type VAPB, suggesting a direct interaction of VAPB and ELYS at the segregating chromosomes in anaphase (Fig. 7A,B). As controls, we also analyzed two proteins, for which the MSP-domain of VAPB is probably not the major/only interaction site, LBR and emerin. For these, the detected PLA-interactions around chromatin were indeed very similar for the mutant and the wild-type version of HA-VAPB, further validating the results obtained for the VAPB-ELYS interaction.

## Effects of ELYS and VAPB on mitosis

The specific interplay of ELYS and VAPB at the chromosomes suggested that the two proteins together might also have a function during mitosis. As a first step to address this issue, we used the cell lines described above, expressing GFP-tagged versions of an ELYS-fragment (ELYS$_{1018-1642}$) comprising the FFAT2-motif (Fig. 5D). Cells were treated with or without tetracycline to induce the expression of the transgenes and analyzed by flow cytometry. As shown in Fig. 8A, tetracycline had no effect on the percentage of cells in the G2- or M-phase of the cell cycle in untransfected cells. Tetracycline-induced expression of the wild-type-version of the ELYS-fragment, however, increased this proportion from 21% to 39%. Interestingly, the S1314A-version of the fragment had a somewhat larger, the S1314D-version a somewhat smaller effect. Two other fragments (S1326A and S1326D) were rather similar to the wild-type fragment in this assay. These results suggest that the propensity of the fragment to interact with VAPB modulates its effect on the cell cycle.

Next, we directly addressed a possible function of VAPB in mitosis and investigated key mitotic parameters in control cells and in VAPB-depleted cells. To monitor the progression through mitosis, HeLa cells that had been treated with siRNAs against VAPB or with non-targeting siRNAs (Fig. 8B) were first subjected to a synchronization protocol, involving a double-thymidine block and the reversible kinesin inhibitor dimethylenastron (DME)

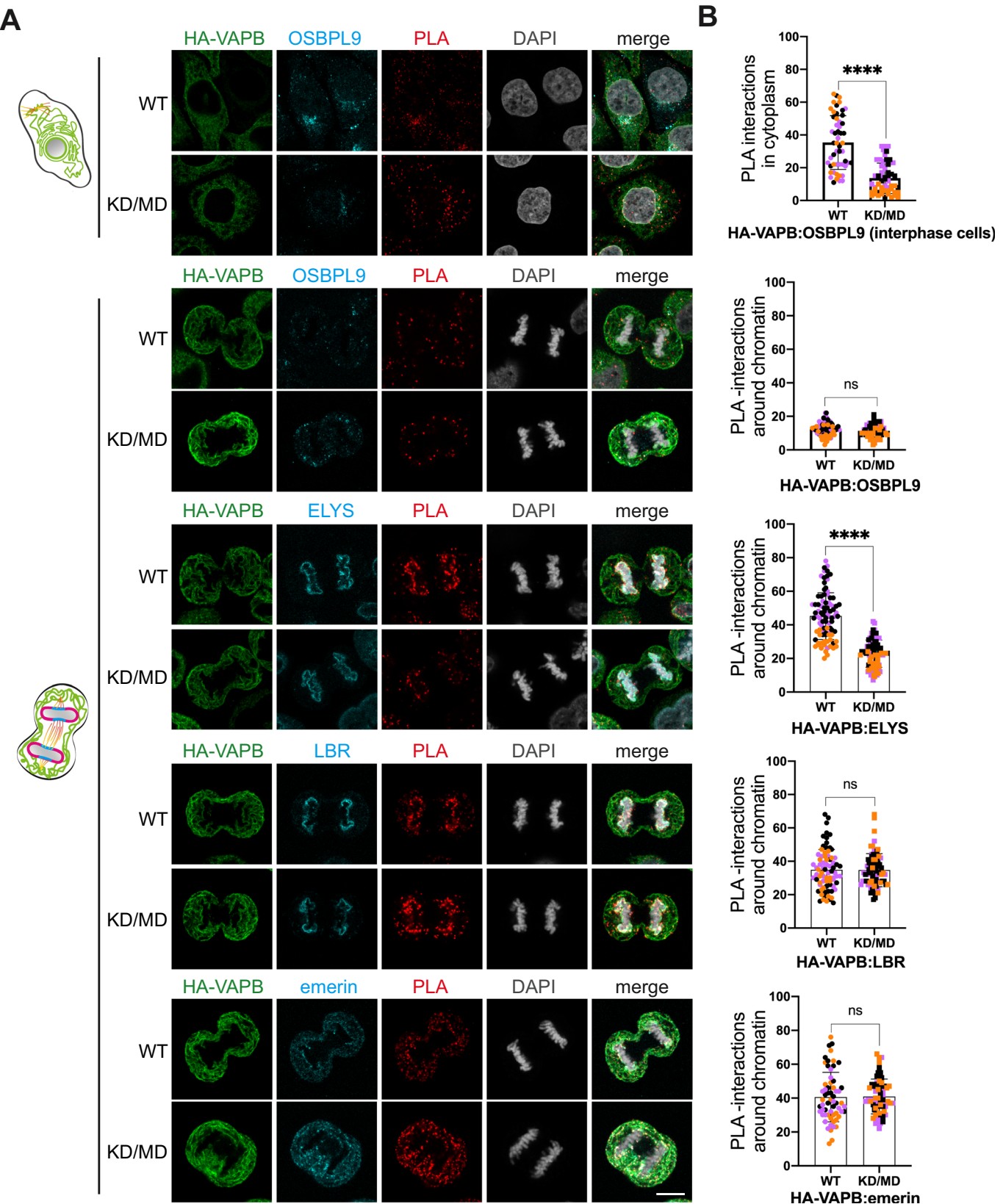

◄ **Figure 7. The MSP-domain of VAPB affects its interaction with ELYS in anaphase.**

(A) Stable cell lines expressing wild-type (WT) HA-VAPB or the KD/MD-mutant version were subjected to double thymidine block and release for synchronization. Cells in interphase (top (double) rows) or in anaphase (four lower rows) were then analyzed by PLAs, detecting proximities between HA-VAPB and OSBPL9 (first and second row), and HA-VAPB and ELYS (third row), LBR (fourth row), and emerin (last row), respectively. The specificity of PLA signals and the antibodies used was validated by single antibody controls (Fig. EV3). Scale bar, 10 μm. (B) Quantification of PLA interactions as detected in (A). First graph, total cellular interactions in 53 cells; remaining graphs, PLA interactions around chromatin, analyzing a total of 35, 42, 40, and 32 anaphase cells (i.e., 2n chromatin regions), respectively. The data are shown as the mean (± standard deviation) of three biological replicates. ****$p < 0.0001$ (two-tailed unpaired parametric t-test). PLA signals were quantified using Cell Profiler (Carpenter et al, 2006). A region around the chromatin was defined by the DAPI signal and by the ExpandorShrinkObjects function with a pixel number of 4 for expansion. The PLA signals were then calculated by the pipeline. Black, orange and pink dots represent individual data of three biological replicates. Source data are available online for this figure.

(Müller et al, 2007) which induces a mitotic arrest in prometaphase (Ertych et al, 2014) (Fig. 8C). After release from the DME-arrest, cells were harvested every 15 min for a total of 2 h and analyzed by flow cytometry. As shown in Fig. 8D,E, VAPB-depleted cells displayed a slower progression through mitosis compared to the control cells, with a clear delay right after the release. We also depleted the VAPB-paralog VAPA which can also localize to the INM and observed only a minor effect on mitotic progression (Fig. EV4A,B; note that the depletion of either protein was not complete, perhaps explaining the small effects (Fig. EV4C)). In addition, we tried to co-deplete VAPA and VAPB, but the efficiency of knockdown in this case was rather low and effects on mitotic progression were not significant. For a more detailed analysis of possible perturbations during mitosis in VAPB-depleted cells, we resorted to a live-cell imaging approach, using cells expressing an mCherry-tagged version of histone H2B. Cells that had been treated with siRNAs against VAPB or with non-targeting siRNAs were subjected to a double-thymidine block, released and imaged to capture mitotic cells. About 90% of the VAPB-depleted cells exhibited a chromosome segregation defect with lagging chromosomes in anaphase, compared to only 10% of the control cells (Fig. 8F,G and Videos EV1 and EV2). A detailed inspection of the lengths of individual mitotic phases further revealed that the depleted cells are clearly slowed down with respect to the transition from meta- to anaphase (Fig. 8H). Other mitotic phases were not affected by the siRNAs against VAPB.

In summary, our results clearly demonstrate phosphorylation-dependent interactions of VAPB and ELYS in mitosis. They are in line with a scenario where such interactions play a role in organizing membranes around the decondensing chromosomes and orchestrating the progression through mitosis.

## Discussion

### ELYS is a preferential binding partner of VAPB in mitosis

VAPB has been extensively studied as an ER-resident protein that serves as a binding partner for other membrane proteins, supporting the formation and/or maintenance of cellular contact sites between the ER and other organelles (De Vos et al, 2012; James and Kehlenbach, 2021; Kim et al, 2022). VAPB was also shown to localize to the INM where we detected it in close proximity to ELYS during interphase (James et al, 2019). Here, we identified ELYS as a *major* direct interaction partner of VAPB in mitotic cells (Fig. 1). In mitosis, contact sites are expected to be partially severed to allow an equal distribution of organelle fragments to the two daughter cells. Indeed, several established

interaction partners of VAPB like ACBD5 or OSBPL9 rather interacted with an immobilized MSP-domain of VAPB derived from asynchronous as opposed to mitotic cells. ELYS, by contrast, was found to preferentially interact with the MSP-domain or with endogenous VAPB, when lysates from mitotic cells were used in pull-down- (Figs. 1 and 2D–F) or co-immunoprecipitation- (Fig. 2A–C) experiments. To the best of our knowledge, ELYS is the first protein that is shown to interact specifically and predominantly with VAPB during mitosis.

### Phosphorylation-dependent binding of VAPB to ELYS

Many nucleoporins and also nuclear lamins (Gerace and Blobel, 1980; Kutay et al, 2021) are known to be phosphorylated at the onset of mitosis. Phosphorylation of Nup98, for example, is a rate-limiting step in the disassembly of the NPC (Laurell et al, 2011). Our results clearly show that ELYS is also heavily phosphorylated during mitosis, indicated by its distinct shift in mobility in SDS-PAGE in mitotic lysates compared to lysates from asynchronous cells (Fig. 2A,C; Appendix Fig. S1). In light of the very large total number of phosphorylation sites in ELYS (>150), it will be very difficult to determine the role of individual sites in protein–protein interactions and mitotic progression. We therefore focused on sites that are part of the predicted FFAT motifs of ELYS to investigate in detail possible consequences on VAPB-ELYS interactions. What is the basis for these interactions? Our initial biotinylation-approach (James et al, 2019) revealed ELYS as a *proximity*-partner of VAPB and did not discriminate between direct and indirect interactions. Using purified protein fragments of VAPB and immobilized ELYS-peptides and ELYS-protein fragments, we now show that the proteins can indeed interact with each other, without the need for additional bridging factors (Figs. 3–5). Furthermore, our co-immunoprecipitation experiments (Figs. 2 and 3) combined with peptide-binding assays clearly show that the interaction is regulated by phosphorylation. Of particular interest here is a sequence (FFAT-2) in ELYS that was identified as an FFAT-like motif, as phosphorylation of a serine residue (S1314) in the acidic tract strongly promoted binding of the MSP-domain of VAPB. Another serine residue in the core region (S1326) had also been identified as a phospho-site in ELYS (Ullah et al, 2016). We note, however, that these sites were not among the phosphosites identified by us, possibly because the corresponding tryptic fragment (GNSSVSITS(1314)DETTLEYQDAPS(1326)PEDLEETVFTASKPK) is very large and may have escaped detection. In our assays, phosphorylation of serine 1326 resulted in reduced binding of endogenous VAPB (Fig. 4C,D) or the purified MSP-domain of VAPB (Fig. 4E,F). Examples for such a regulation by phosphorylation within FFAT motifs have been described before (Kors et al, 2022). Overall, phosphorylation clearly promoted binding of VAPB to ELYS, as treatment of lysates with λ-phosphatase largely abrogated the interaction. FFAT-2 is found in a

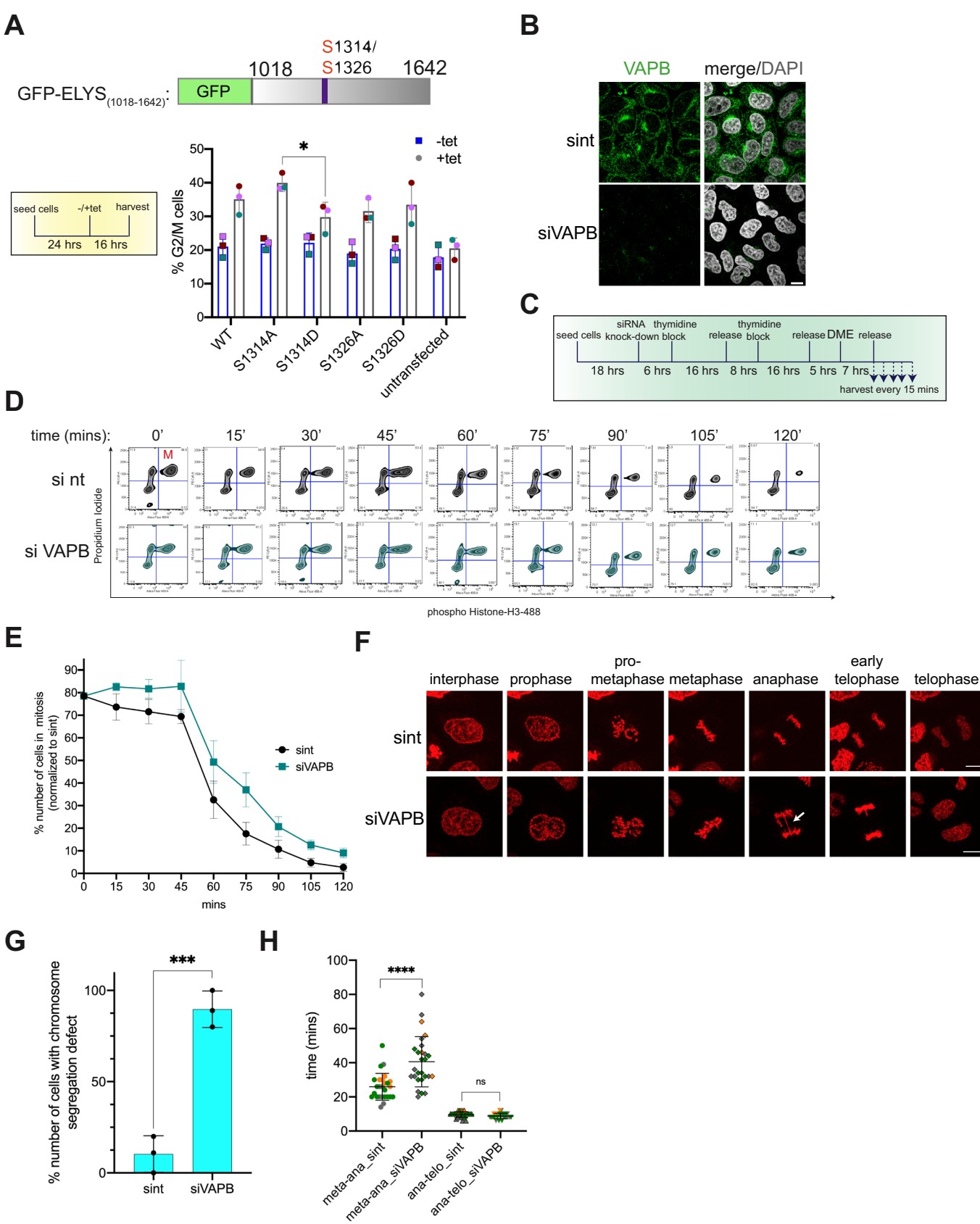

◀ **Figure 8.  Depletion of VAPB delays mitosis and leads to chromosome segregation defects.**

(A) Stable cell lines expressing different versions of GFP-ELYS$_{(1018-1642)}$ were treated with (+tet) or without (-tet) tetracycline for 16 h and harvested. Cells were then analyzed by flow cytometry using propidium iodide. The bar graph shows the quantification of three independent experiments (individual data in green, red, and pink), depicting the percentage of G2/M cells. The data are shown as the mean (± standard deviation) of three biological replicates. *$p = 0.0231$ (Tukey's multiple comparisons test). See Appendix Fig. S3 for flow cytometry data. (B) HeLa P4 cells were treated with siRNAs against VAPB (siVAPB) or non-targeting siRNAs (sint) and analyzed by indirected immunofluorescence, detecting endogenous VAPB, and confocal microscopy. Scale bar, 10 µm. (C) Schematic representation of synchronization- and release steps for HeLa P4 cells after knocking down VAPB using siRNA. Cells were transfected using either non-targeting or siRNA against VAPB. (D) HeLa P4 cells were transfected with siRNAs against VAPB (siVAPB) or non-targeting siRNAs (sint) and synchronized and further treated as depicted in (C). Cells were then analyzed by flow cytometry using propidium iodide and antibodies against phospho-Histone H3. Gated mitotic cells are observed in the upper right quadrant. (E) Quantification of three independent experiments as in (D), depicting the percentage of mitotic cells as determined by flow cytometry. The data are shown as the mean (± standard deviation) of three biological replicates. (F) Live cell imaging of HeLa Flp-In T-REx cells stably expressing mCherry-H2B after knock down of VAPB. Cells transfected with either non-targeting siRNAs (sint) or siVAPB were synchronized by a double thymidine block. After releasing the block for 6 h, the cells were imaged every 2 min for a total of 2 h after the onset of mitosis. Examples from different phases of mitosis are shown. The arrows indicate lagging chromosomes. Scale bar, 10 µm. See also Movies EV1 and EV2. (G) Cells were treated as in (D) and the percentage of mitotic cells showing chromosome segregation defects were quantified from live cell imaging experiments performed as in (F). Error bars indicate the standard deviation from the mean of three independent experiments. ***$p = 0.0006$ (two-tailed unpaired parametric t-test). (H) Cells were treated as in F and the progression from metaphase to anaphase and from anaphase to telophase was measured using live cell imaging. Three independent experiments were performed (dots in gray, green and orange), analyzing a total of 26 cells for si nt and 27 cells for siVAPB. The data are shown as the mean (± standard deviation) of three biological replicates. ****$p < 0.0001$; ns, not significant, $p = 0.9859$ (Tukey's multiple comparisons test). Source data are available online for this figure.

region of ELYS without a predicted structure, as it is typically the case for FFAT motifs (Mikitova and Levine, 2012). The FFAT-2 region is highly conserved between species, suggesting that it is important for protein functions (Fig. EV4D). Another motif, FFAT-1, which is found in a structured region of ELYS, was initially identified with an FFAT-score of 3.5 (Di Mattia et al, 2020), indicating a comparatively low level of similarity to the conventional FFAT-sequence. Nevertheless, we detected binding of endogenous VAPB and of purified MSP-domains to the corresponding peptide (Fig. 3C,D), which was further enhanced upon phosphorylation of a serine residue (S825). It remains to be investigated, how phosphorylation of two or more residues affects the interaction with VAPB in the context of full-length ELYS. Likewise, it remains unknown, which kinases are involved in phosphorylation of ELYS during mitosis. In our pull-down experiment (Fig. 1C), we detected several kinases that were enriched on the immobilized MSP-domain of VAPB when lysates from mitotic cells were used as starting material, e.g., CDK1 and Aurora kinases A and B. It has not been investigated if these proteins interact directly with the MSP-domain, or indirectly, for example via a potential substrate like ELYS. Interestingly, ELYS also interacts with protein phosphatase 1 (PP1; (Hattersley et al, 2016)), which may also affect the phosphorylation status of adjacent proteins like LBR (Mimura et al, 2016). The proposed docking site for PP1 on ELYS is very close to its FFAT-2 motif (Hattersley et al, 2016), but it is unknown if PP1-activity is required for de-phosphorylation of ELYS itself.

## Mitotic roles of ELYS and VAPB

Every interpretation of VAPB-knockdown effects is complicated by the fact that the protein has a vast number of interaction partners (Huttlin et al, 2015; James and Kehlenbach, 2021; James et al, 2019). The majority of these interactions, however, are mostly relevant during interphase. Hence, we consider ELYS, clearly a preferred binding partner of VAPB during mitosis, as a likely component of the VAPB-network that is involved in the effects observed in this study. ELYS, a nucleoporin that does not contain a transmembrane domain, directly interacts with chromatin during mitosis via its AT-hook and recruits nucleoporin-subcomplexes to initiate the formation of post-mitotic NPCs. Concurrent with the separation of chromosomes during anaphase, membranes derived

from the mitotic ER assemble around the chromatin masses to ultimately re-assemble the NE ((Anderson and Hetzer, 2007; Anderson et al, 2009), for review see also (Güttinger et al, 2009)). This process has to be tightly coordinated with the biogenesis of the NPC and VAPB could have a function here, as it interacts with ELYS, particularly in anaphase cells. A first analysis of NPC-numbers by STED-microscopy, however, revealed no significant changes upon depletion of VAPB (Appendix Fig. S2). Binding of ELYS to chromatin is an early event in post-mitotic NPC-biogenesis (Franz et al, 2007; Rasala et al, 2008) that is followed by the recruitment of components of the Y-complex (Franz et al, 2007; Gillespie et al, 2007). In the complete absence of ELYS, nuclear envelopes can be formed but lack NPCs entirely (Franz et al, 2007). It was also reported that further oligomerization of original nucleoporin subcomplexes requires the recruitment of membrane components to the site of NPC-assembly (Rotem et al, 2009) and that nuclear envelope formation always precedes NPC-assembly (Lu et al, 2011), although alternative scenarios have been suggested (for review see (Liu and Pellman, 2020)). Whatever the exact sequence of events, integral membrane proteins are expected to play important roles in these steps. Indeed, several proteins of the INM can directly interact with chromatin, for example LBR (Ye and Worman, 1994) and Lap2β (Foisner and Gerace, 1993). Furthermore, INM-proteins in general are enriched in basic domains that may facilitate membrane interactions with DNA (Ulbert et al, 2006). VAPB localized largely to the non-core region of anaphase chromosomes, together with ELYS. Depletion of VAPB resulted in prolonged mitosis, with particularly lengthened transition times from metaphase to anaphase (Fig. 8D–E,H), i.e., at stages in mitosis where the close proximity/interaction of ELYS and VAPB was most pronounced (Figs. 6D and 7). These results are in line with a very recent publication where the authors reported a delay in cell cycle progression upon knockout of VAPB in medulloblastoma cells (Faria Assoni et al, 2023).

Similar effects as for VAPB have been observed upon knock down of several INM proteins like LBR, Lap2β, and MAN1. However, reduction in any one of these proteins did not block NE formation and is thus indicative of a redundant system (Anderson et al, 2009). A delay at individual stages, however, can lead to chromosome segregation defects (Afonso et al, 2014), as seen for VAPB (Fig. 8F–H),

with lagging chromosomes or chromosome bridges. Ultimately, such defects could also lead to the generation of micronuclei and genome instability, although this was not observed in our studies (see Videos 1 and 2). Hence, it appears that the cells are able to cope with the lack of VAPB, allowing more time for a faithful segregation of chromosomes and thus avoiding deleterious effects. Similar observations were made in cells overexpressing Nup107 or lamin B, which enforced a chromosome separation checkpoint to prevent NE reformation on incompletely separated chromosomes (Afonso et al, 2014). Clearly, a function of VAPB in mitosis is not essential, as VAPB-knockout cells (Dorsch et al, 2021) as well as knockout mice (Silbernagel et al, 2018) are viable. Other cellular components that could function here as rescue factors are VAPA, a paralog very similar in sequence and localization to VAPB (Lev et al, 2008), and MOSPD2 (Di Mattia et al, 2018). In our experiments, knockdown of VAPA alone resulted in a minor delay in mitosis (Fig. EV4A,B). Hence, it seems likely that independent mechanisms apply as well, relying on different proteins that mediate and coordinate the interaction of membranes with chromatin. The large number of INM proteins with the ability to function in membrane-recruitment may provide a safety mechanism in post-mitotic NE-formation where loss of a single protein may affect the efficiency of the process but is overall tolerated. As an alternative to a positive function, where VAPB helps to recruit membranes to chromatin, binding of VAPB to phosphorylated ELYS could also exert negative effects, e.g., by preventing premature NPC-biogenesis. After dephosphorylation of ELYS (e.g., at later stages of anaphase), VAPB would be released from the complex so that ELYS could fulfill its functions. Obviously, these alternatives are not mutually exclusive. For a better understanding of these mechanisms, a precise determination of the schedule of phosphorylation- and dephosphorylation events on ELYS during mitosis (or even during subphases of meta- and anaphase) will be required.

# Methods

## Plasmids

For pcDNA5/FRT/TO-GFP, the GFP sequence was amplified by PCR using primers G2594 and G2595 and pEGFP-C3 (Addgene #2489) as template (oligonucleotides are listed in Appendix Table S1). The PCR product was cloned into pcDNA5-FRT-TO vector (Thermo Fischer Scientific) via AflII and HindIII. The VAPB coding sequence was amplified using primers G1390 and G1386 and pCAN-myc-VAPB (Fasana et al, 2010) as a template and inserted into pcDNA5/FRT/TO-GFP via KpnI and BamHI. To obtain pcDNA5/FRT/TO-HA-VAPB, the HA-tag was inserted into pcDNA5/FRT/TO by oligo annealing using oligos G2599 and G2600. The VAPB coding sequence was amplified using primers G2601 and G2602 and pCAN-myc-VAPB (Fasana et al, 2010) as a template and inserted into pcDNA5/FRT/TO-HA via EcoRV and NotI. Site-direct mutagenesis (QuikChange Site-Directed Mutagenesis kit (Agilent Technologies)) of pcDNA5/FRT/TO-HA-VAPB was performed using primers G2403, G2404, G2405, G2406 to generate pcDNA5/FRT/TO-HA-VAPB-K87D/M89D.

For pGEX-6P-1-MSP-VAPB, the MSP domain of VAPB was amplified by PCR using primers G2387 and G2388 and pCAN-myc-VAPB (Fasana et al, 2010) as a template. The PCR product was cloned into pGEX-6P-1 through EcoRI and BamHI. To

generate the MSP-KD/MD mutant, site-directed mutagenesis was performed as described above using primers G2403, G2404, G2405, G2406. To generate the MSP-K43L mutant, site-directed mutagenesis was performed using primers G2772 and G2773. For pcDNA5/FRT/TO-GFP-ELYS$_{1018-1642}$, ELYS$_{1018-1642}$ was amplified using primers G2712 and G2651 and HeLa cDNA as template. The PCR product was cloned into pcDNA5/FRT/TO-GFP via BamHI and NotI. To generate ELYS$_{1018-1642}$-S1314A, S1314D, S1326A, and S1326D mutants, the QuikChange Site-Directed Mutagenesis kit (Agilent Technologies) and primers G2781/G2782, G2783/G2784, G2785/G2786, and G2787/G2788 were used. To generate pMal-His-ELYS$_{1018-1642}$, the ELYS$_{1018-1642}$ fragment was amplified using G2764 and G2482 and HeLa cDNA as template and inserted into pMal-His-MBP via EcoRI and BamHI. All plasmids obtained were confirmed by sequencing (Eurofins Genomics).

## Cell culture, transfection, and drug treatment

Parental HeLa Flp-In T-REx (Hafner et al, 2014) and H2B-cherry HeLa Flp-In T-REx cells used to generate stable tetracycline-inducible cell lines were a gift from T. Mayer (University of Konstanz, Germany). HeLa P4 cells (Charneau et al, 1994) were obtained from the NIH AIDS Reagent Program. Cells were grown in DMEM supplemented with 10% (v/v) normal or tetracycline-free FBS (Thermo Fischer Scientific), 100 U ml$^{-1}$ penicillin, 100 µg ml$^{-1}$ streptomycin and 2 mM l-glutamine (Thermo Fischer Scientific) at 37 °C in the presence of 5% $CO_2$ and tested for contamination by mycoplasma on a regular basis.

siRNA-mediated knockdown was carried out using Lipofectamine RNAiMAX (Thermo Fisher Scientific), following instructions of the manufacturer. VAPB siRNAs (GCUCUUGGCUCUGGUG-GUUUU and AAAACCACCAGAGCCAAGAGC; Sigma), ELYS siRNAs (GAAUGAUCGUGAUCCUCGU and ACGAGGAUCAC-GAUCAUUC; Sigma), VAPA siRNAs (GGCAAAACCUGAU-GAAUUA and UAAUUCAUCAGGUUUUGCC; Sigma) and ON-Target-plus nontargeting siRNA (D-001810-01-50, Dharmacon, Lafayette, CO) were used at a final concentration of 50 nM. Transfections for the generation of stable cell lines were performed using X-tremeGENE™ 9 DNA transfection reagent (Merck), following the instructions of the manufacturer. Briefly, pcDNA5/FRT/TO and the pOG44 vector (three times the concentration of pcDNA5/FRT/TO plasmid) (Thermo Fischer Scientific) were incubated for 15 min at room temperature with X-tremeGENE™ 9 DNA. The mixture was added to the cells, which were then grown as described above.

Cells were arrested in G1/S phase by treatment with thymidine (Sigma) or in mitosis by treatment with DME (Eg5 inhibitor III, Enzo Life Sciences). For a double thymidine block, cells were first treated with 2 mM thymidine for 16 h, followed by a release from the block for 8 h. The cells were treated again with thymidine for 16 h and released for 8 to 9 h. For flow cytometry analysis, cells were synchronized by a double thymidine block after 6 h of transfection with siRNAs. After the second block, cells were released for 5 h. After addition of 1 µM of DME from a 10 mM stock in DMSO and incubation for 7 h, cells were harvested every 15 min for 120 min from the time of release.

For λ-protein phosphatase (λPP) treatment, asynchronous or mitotic HeLa cell lysates were incubated with 0.9 mM MnCl$_2$ and 7 U/µl λPP (New England Biolabs) or 0.9 mM MnCl$_2$ and water for 1 h at 30 °C.

## Generation of stable cell lines

HeLa Flp-In T-REx cells were seeded onto 24-well plates and cultured for 24 h. Cells were then co-transfected with 0.5 μg of one of the pcDNA5/FRT/TO-plasmids and 1.5 μg of pOG44 using 1 μl of X-tremeGENE™ 9 DNA. On the next day, cells from three wells were pooled, reseeded in 10 cm dishes and selected using hygromycin (200 μg/ml) and blasticidin (5 μg/ml). Transgene expression was induced by the addition of tetracycline (1 μg/ml) and further incubation for 24 h.

## Protein expression and purification

GST-MSP constructs were expressed in *E. coli* BL21 (DE3) codon⁺ strain at 16 °C for 16 h upon induction with 1 mM isopropyl β-D-1-thiogalactopyranoside (IPTG). After washing with PBS, cells were suspended in lysis buffer (50 mM Tris-HCl, pH 7.5, 300 mM NaCl, 1 mM MgCl$_2$, 1 mM DTT, 1 mg/ml each of aprotinin, pepstatin, and leupeptin, and 0.1 mM PMSF) and lysed using an Emulsiflex-C3 (Avestin, Germany). The lysate was centrifuged at $100,000 \times g$ at 4 °C for 30 min and the supernatant was incubated using Glutathione Sepharose-4-Fast Flow resin for 1 h at 4 °C followed by washing steps with lysis buffer. Proteins were eluted using lysis buffer supplemented with 15 mM glutathione and dialyzed overnight at 4 °C against lysis buffer.

His-ELYS$_{1018-1642}$-MBP constructs were expressed in *E. coli* BL21 (DE3) codon⁺ strain at 16 °C for 16 h upon induction with 0.5 mM IPTG. Cells were suspended in His-lysis buffer (50 mM Tris HCL, pH 8, 300 mM NaCl, 10 mM imidazole, 0.1% Triton X-100, 2 mM β-mercaptoethanol, 5% glycerol, 1 mg/ml each of aprotinin, pepstatin and leupeptin and 0.1 mM PMSF and lysed using an Emulsiflex-C3. The lysate was centrifuged at $30,000 \times g$ at 4 °C for 30 min and the supernatant was incubated using Ni Sepharose High Performance (GE Healthcare) for 1.5 h at 4 °C followed by washing with wash buffer (His-lysis buffer containing no Triton X-100). Proteins were eluted using 250 mM imidazole in wash buffer and dialyzed overnight at 4 °C against wash buffer.

## Peptide pull down assays

Synthetic biotinylated peptides were purchased from ProteoGenix, France. Peptide pull down assays were performed as described by (Di Mattia et al, 2020). Briefly, for peptide pull down assays from total HeLa-cell lysate, 30 nmol of biotinylated peptides were incubated with 30 μl neutravidin agarose resin in 1 ml of buffer 1 (50 mM Tris–HCl, pH 7.4, 75 mM NaCl, 1 mM EDTA, 1% Triton X-100 and protease inhibitor cocktail tablet (cOmplete, Roche)) at 4 °C for 1 h. The immobilized beads were then washed twice with buffer 2 (50 mM Tris–HCl, pH 7.4, 500 mM NaCl, 1 mM EDTA, 1% Triton X-100 and protease inhibitor cocktail tablet (cOmplete, Roche)) and twice with buffer 1. HeLa P4 cells from one 10 cm dish (8 million cells) per condition were lysed using 500 μl of buffer 3 (50 mM Tris–HCl, pH 7.4, 75 mM NaCl, 1 mM EDTA, 1% Triton X-100, protease inhibitor cocktail (cOmplete, Roche) and phosphatase inhibitor (phosSTOP) tablet (Roche)). The cell lysate was incubated for 5 min on ice and centrifuged at $9600 \times g$ at 4 °C for 10 min. The cleared lysate was incubated with peptide-coupled neutravidin beads at 4 °C for 2 h. The beads were then washed four times with 1 ml buffer 3 and bound proteins were eluted using 30 μl of SDS sample buffer (4% (w/v) SDS,

125 mM Tris, pH 6.8, 10% (v/v) glycerol, 0.02% (v/v) bromophenol blue and 10% (v/v) β-mercaptoethanol)).

For binding assays using peptides and recombinant proteins, 20 nmol of biotinylated peptides were incubated with 20 μl neutravidin agarose resin in 500 μl of buffer 4 (50 mM Tris–HCl, pH 7.4, 75 mM NaCl, 1 mM EDTA, 1% Triton X-100, 1 mM DTT and protease inhibitor cocktail tablet (cOmplete, Roche)) at 4 °C for 1 h. The immobilized beads were then washed thrice with buffer 4 and blocked using 1.25 mg/ml hemoglobin from bovine blood (Sigma) at 4 °C for 1 h. The beads were washed thrice with buffer 4. Thirty micrograms of purified proteins were incubated in 500 μl buffer 5 (50 mM Tris–HCl, pH 7.4, 75 mM NaCl, 1 mM EDTA, 1% Triton X-100, 0.25 mM DTT and protease inhibitor cocktail (cOmplete, Roche)) at 4 °C for 2 h. The beads were then washed four times with buffer 5 and bound proteins were eluted using 25 μl of SDS sample buffer.

## Immunoprecipitation

$2 \times 10^6$ HeLa Flp-In T-REx cells were seeded per 10 cm dish with or without the addition of 1 μg/ml tetracycline. After 24 h, the cells were treated with either 2 mM thymidine or 0.5 μM DME for 14 h to arrest them in G1/S or in mitosis, respectively. The cells were lysed using lysis buffer (10 mM Tris–HCl, pH 7.4, 400 mM NaCl, 2 mM EDTA, 1% Triton X-100, 0.1% SDS, 1 mM DTT, 10 U/ml benzonase (EMD Millipore), protease inhibitor cocktail (cOmplete, Roche) and phosphatase inhibitor (phosSTOP; Roche) and then centrifuged for 15 min at $14,800 \times g$ at 4 °C. The clarified lysate was then diluted 3.75 times using dilution buffer (10 mM Tris–HCl, pH 7.4, 2 mM EDTA, 1 mM DTT, protease inhibitor cocktail (cOmplete, Roche) and phosphatase inhibitor (phosSTOP, Roche)). For immunoprecipitation of GFP-ELYS$_{1018-1642}$ fragments, cells synchronized in mitosis by 1 μM DME treatment were lysed using lysis buffer (10 mM Tris–HCl, pH 7.4, 400 mM NaCl, 2 mM EDTA, 1% Triton X-100, 0.1% SDS, 1 mM DTT, protease inhibitor cocktail (cOmplete, Roche) and phosphatase inhibitor (phosSTOP, Roche) and then centrifuged for 15 min at $16,100 \times g$ at 4 °C. The clarified lysate was diluted 3.75 times using dilution buffer (as above) containing 1% Triton X-100. For immunoprecipitation of GFP-VAPB and GFP-ELYS fragments, 50 μl of GFP-Selector (NanoTag) were washed thrice with 3.75 times diluted lysis buffer. The lysate was incubated with the beads for 3 h at 4 °C. The beads were then washed four times with diluted lysis buffer and bound proteins were eluted with 40 μl of SDS sample buffer. For immunoprecipitation of endogenous protein complexes from total HeLa cell lysate, mitotic cells were enriched as described above. 4 μg of rabbit anti-VAPB, or IgG as a control were immobilized on 50 μl of Protein A–Sepharose 4 Fast Flow beads (GE Healthcare) for 3 h and incubated with clarified lysates.

## GST pull down

GST proteins (200 pmol) were immobilized on GST-Selector beads (Nanotag, Germany) equilibrated in diluted lysis buffer as described above (see immunoprecipitation) for 1 h at 4 °C. The beads were washed thrice using diluted lysis buffer. Cell lysates (see immunoprecipitation protocol) were then incubated with immobilized beads for 3 h. Bound proteins were eluted with 40 μl of SDS sample buffer.

## Binding assays

Purified His-ELYS$_{1018-1642}$-MBP fragments (100 pmol) were immobilized on MBP-Selector beads (Nanotag, Germany) equilibrated in buffer 6 (50 mM Tris HCl, pH 7.4, 150 mM NaCl, 1 mM EDTA, 1 mM DTT, protease inhibitor cocktail tablet) for 1 h at 4 °C. The immobilized beads were washed thrice with buffer 6 and blocked with hemoglobin (1.25 mg/ml) for 1 h at 4 °C. The beads were washed thrice with buffer 7 (50 mM Tris HCl, pH 7.4, 150 mM NaCl, 1 mM EDTA, 0.25 mM DTT, protease inhibitor cocktail tablet). 300 pmol of either GST or GST-MSP-VAPB were added to the immobilized beads and incubated in buffer 7 containing hemoglobin (1.25 mg/ml) for 2 h at 4 °C. The beads were then washed five times with buffer 7 and bound proteins were eluted using 25 μl of SDS sample buffer.

## Flow cytometry

HeLa P4 cells released from a double thymidine block were fixed with 70% ethanol overnight at 4 °C. The fixed cells were permeabilized with 0.25% Triton X-100 in PBS for 5 min and blocked with 1% BSA for 5 min. The cells were stained with an Alexa Fluor 488-labeled anti-phospho-histone H3 (Ser28) antibody (Thermo Fischer Scientific) for 90 min. After staining, cells were resuspended in PBS containing RNase A (Machery-Nagel, Germany) (0.1 mg/ml) and incubated overnight at 4 °C. Propidium iodide (Thermo Fischer Scientific) was added to a concentration of 0.03 μg/μl and cells were analyzed using a BD FACSCanto II (Becton-Dickinson). For flow cytometry analysis of GFP-ELYS$_{1018-1642}$ expressing cells, dead cells and debris were eliminated from the analysis using forward and side scatter parameters. DNA content in GFP-positive cells was measured by propidium iodide staining. Data were analyzed with FlowJo software.

## MS-DIA

MS-DIA analysis was largely performed according to published protocols (Lambert et al, 2013; Meier et al, 2020; Zhang et al, 2015). Protein samples were reconstituted in Laemmli buffer, loaded onto a 4–12% NuPAGE Novex Bis-Tris Minigels (Invitrogen), and run into the gel for 1.5 cm. Following Coomassie staining, the protein areas were cut out, diced, and subjected to reduction with dithiothreitol, alkylation with iodoacetamide and finally overnight digestion with trypsin. Tryptic peptides were extracted from the gel, the solution was dried in a Speedvac and kept at −20 °C for further analysis.

Protein digests were analyzed on a nanoflow chromatography system (Vanquish neo) linked to a hybrid quadrupole-obritrap mass spectrometer (Exploris 480, all Thermo Fisher Scientific) using an EasySpray ion source. In brief, 150 ng equivalents of peptides were dissolved in loading buffer (2% acetonitrile, 0.1% trifluoroacetic acid in water), enriched on a reversed-phase C18 trapping column (0.3 cm × 300 μm, Thermo Fisher Scientific) and separated on a reversed-phase C18 column (Aurora 25 cm × 75 μm, IonOpticks) using a 45 min linear gradient of 5–35% acetonitrile/ 0.1% formic acid (v:v) at 300 nl min$^{-1}$, and a column temperature of 50 °C. Data-independent acquisition analysis was performed using a custom 30 variable window isolation scheme from $m/z$ 350 to 1150 to include the 2 +/3 +/4+ population. Two technical replicates per biological replicate were acquired.

Protein identification and quantification was achieved using the Pulsar algorithm in Spectronaut Software version 18.1 (Biognosys) using default settings. All DIA data were searched against the UniProtKB Homo sapiens reference proteome (revision 08-2023) combined with a set of 51 known common laboratory contaminants at default settings. For quantitation, up to the 6 most abundant fragment ion traces per peptide, and up to the 10 most abundant peptides per protein were integrated and summed up to provide protein area values. Mass and retention time calibration as well as the corresponding extraction tolerances were dynamically determined. Both identification and quantification results were trimmed to a False Discovery Rate (FDR) of 1% using a forward-and-reverse decoy database strategy. Perseus software version 1.6.15 (Max Planck Institute for Biochemistry, Martinsried, Germany) was used for statistical evaluation of relative protein quantitation values from the MaxQuant software results.

## Phosphopeptide enrichment and HPLC-MS

Asynchronous or mitotic HeLa cell lysates were incubated with GST-VAPB-MSP immobilized on GST-selector beads and bound proteins were digested in-solution with trypsin using the SP3 protocol (Hughes et al, 2019). Phosphopeptides were enriched by TiO2 chromatography as described (Oellerich et al, 2009). In brief, tryptic peptides were dissolved in 60 μl of the buffer A (5% (v/v) glycerol, 80% (v/v) acetonitrile (ACN), 5% (v/v) trifluoroacetic acid), and bound to a self-packed TiO$_2$ tip-column (Titansphere 5 μm, GL Sciences Inc.). The column was washed 3 times with 60 μl of the buffer A, 3 times with 60 μl of the buffer B (80% (v/v) ACN, 5% (v/v) trifluoroacetic acid) and once with the buffer B2 (60% (v/v) ACN, 0.1% (v/v) trifluoroacetic acid). Phosphopeptides were eluted from the TiO$_2$ column by incubating it 3 times with 40 μl of 1.18% ammonia in water, pH ≥ 10.5.

The samples were analyzed using an Orbitrap Exploris 480 mass spectrometer coupled to a Dionex Ultimate 3000 UHPLC system equipped with a PEPMAP100 C18 nano-trap column (all Thermo Fisher Scientific) and an in-house-packed 30 cm analytical C18 capillary column (360 μm outer diameter, 75 μm inner diameter, ReproSil-Pur 120 C18-AQ 3 μm beads, Dr. Maisch GmbH). Peptides were separated using a linear gradient from 7.2 to 30% (v/v) ACN in 0.1% (v/v) formic acid for 70 min. The mass spectrometers were operated in a data-dependent acquisition mode using a cycle time of 3 s at a resolution of 120,000 and 30,000 for MS1 and MS2 scans, respectively. Raw files were searched with MaxQuant software (version 1.6.17.0) against a database containing a GST-fusion of truncated VAPB protein, the UniProt human reference proteome (release 2023-06-28, 20598 entries) and protein contaminants commonly found in MS samples. Phosphorylation of serine, threonine, and tyrosine residues as well as methionine oxidation and protein N-terminal acetylation were considered as variable modifications. Cysteine carbamidomethylation was set as a fixed modification. The precursor tolerance was set to 10 ppm and the MS/MS tolerance to 20 ppm. Results were filtered to FDR 1%.

## Western blot analyses

Western blotting was performed according to standard protocols using IRDye (LI-COR) secondary antibodies. 3–8% NuPAGE Tris-Acetate gels (Thermo Fischer Scientific) were used to separate ELYS

by SDS-PAGE. For other proteins, 4–12% NuPAGE Bis-Tris gels were used. After transfer to PVDF or nitrocellulose, the membranes were incubated in blocking buffer (3% BSA in TBS-T (24.8 mM Tris, pH 7.4, 137 mM NaCl, 2.7 mM KCl, 0.05% (v/v) Tween 20)) for 1 h at room temperature and with primary antibodies (Appendix Table S2) overnight at 4 °C. Incubation with IRDye-labeld secondary antibodies (Appendix Table S2; diluted 1:10,000 in blocking buffer) for 1 h at room temperature was followed by three washing steps with TBS-T. Proteins were detected using LI-COR Odyssey-CLx imaging system and analyzed by Image Studio Lite software (LI-COR). For statistical analyses, ordinary one-way ANOVA, followed by Brown-Forsythe and Welch tests were performed using GraphPad Prism 8 software and a confidence interval of 95% was set.

### Immunofluorescence, confocal microscopy, and live cell imaging

For confocal microscopy, cells were grown on coverslips and fixed with 4% (v/v) paraformaldehyde. For immunofluorescence, fixed cells were permeabilized with 0.5% (v/v) Triton X-100 in PBS for 5 min at room temperature and blocked with 3% (w/v) BSA in PBS for 20 min at room temperature. Staining was performed for 1 h at room temperature using appropriate primary antibodies and fluorescently labeled secondary antibodies (Appendix Table S2), which were diluted in 3% BSA in PBS. Cells were embedded in MOWIOL supplemented with 1 μg ml$^{-1}$ DAPI. Cells expressing fluorescently labeled proteins were mounted directly with MOWIOL containing DAPI.

An LSM510 confocal laser scanning microscope using a 100X/1.3 oil immersion lens (Zeiss, Germany) was used for microscopic analysis. For live cell imaging, cells were grown on ibidi μ-Dish$^{35\,mm,}$ $^{low}$ and imaged using an LSM800 confocal laser scanning microscope using a 40X/1.3 oil DIC M27 immersion lens (Zeiss) under 5% $CO_2$ at 37 °C. Images were analyzed using Zen Blue software (Zeiss). For statistical analysis, ordinary one-way ANOVA followed by Brown-Forsythe and Welch tests was performed and a confidence interval of 95% was set.

### Proximity ligation assay (PLA)

HeLa Flp-In T-REx cells expressing HA-VAPB or HA-VAPB-KD/ MD were seeded at a density of 35,000 cells/well in 24-well plates. After synchronizing the cells in mitosis, cells were fixed with 4% paraformaldehyde and permeabilized with 0.5% (v/v) Triton X-100. The Duolink In Situ PLA kit (Sigma Aldrich) was used for PLA. Cells were blocked and incubated with mouse anti-HA or rabbit anti-ELYS antibodies and thereafter with the corresponding PLA probes. After ligation and amplification using the kit reagents, cells were counterstained for HA and ELYS and mounted using Duolink mounting medium with DAPI. Images were acquired on a LSM510 confocal laser scanning microscope using 100X/1.3 oil immersion lens. Anaphase cells were analyzed for PLA interactions using CellProfiler (Carpenter et al, 2006). A pipeline was generated to measure PLA signal intensities and numbers around the chromatin. Chromatin was identified by DAPI staining. The region around the chromatin was defined by the function ExpandOrShrinkObjects with a pixel number of 4 for expansion. An unpaired t-test was performed for statistical analysis using GraphPad Prism 8 software and a confidence interval of 95% was set.

## Data availability

The mass spectrometry proteomics data have been deposited to the ProteomeXchange Consortium via the PRIDE (Perez-Riverol et al, 2019) partner repository with the dataset identifiers PXD047720 for DIA-MS and PXD048054 for phosphopeptide analysis.

## Peer review information

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

## Acknowledgements

We would like to thank Lisa Neuenroth, Monika Raabe, and Olivia Langer for expert technical support, Dr. Thomas Mayer (University of Konstanz, Germany) for stable cell lines, Dr. Holger Bastians (University of Göttingen, Germany) for very helpful advice and reagents and Dr. Larry Gerace, La Jolla,

USA for very fruitful discussions. We also appreciate the very helpful comments from the anonymous reviewers. The project was supported by the Deutsche Forschungsgemeinschaft (DFG, SFB1190, P07, and Z02). We also acknowledge support by the Open Access Publication Funds/ transformative agreements of the Göttingen University.

## Author contributions

**Christina James**: Conceptualization; Data curation; Formal analysis; Validation; Investigation; Visualization; Methodology; Writing—original draft; Writing—review and editing. **Ulrike Möller**: Investigation. **Christiane Spillner**: Investigation. **Sabine König**: Investigation. **Olexandr Dybkov**: Formal analysis; Investigation. **Henning Urlaub**: Supervision; Funding acquisition; Methodology. **Christof Lenz**:  Formal analysis; Supervision. **Ralph H Kehlenbach**: Conceptualization; Formal analysis; Funding acquisition; Writing—original draft; Project administration; Writing—review and editing.

## Funding

## Disclosure and competing interests statement

The authors declare no competing interests.

# Expanded View Figures

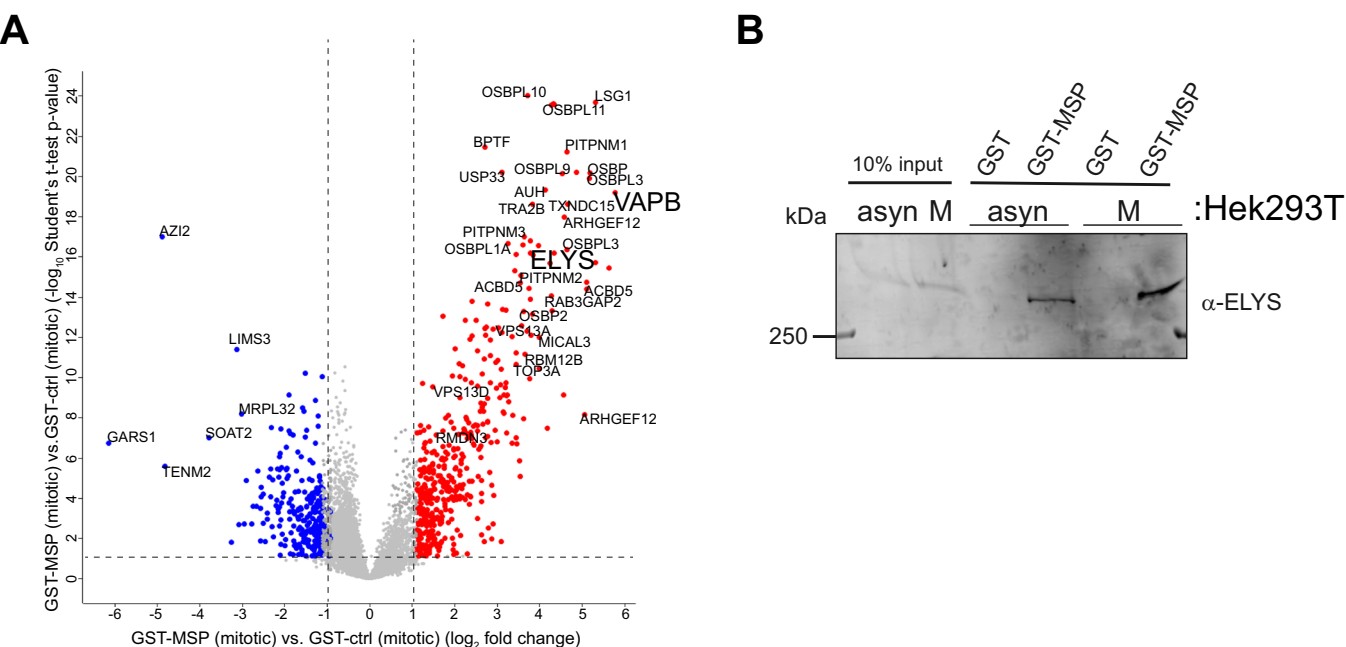

**Figure EV1. The MSP-domain of VAPB interacts with ELYS in mitosis.**

(A) MS-DIA-results comparing proteins from mitotic cells bound to GST or GST-MSP. Proteins enriched on GST-MSP beads are depicted in red. The graph shows the combined results of four independent experiments with two technical replicates each. A two-sided Student's T-test was performed using normalized $\log_2$ ratios. For the analysis, a Permutation-based FDR was applied, and a threshold value of 0.05 was chosen. (B) HEK293T-cells were subjected to a synchronization protocol and lysates from asynchronous (asyn) and mitotic (M) cells were obtained. Lysates were then incubated with immobilized GST or GST-MSP and bound proteins were analyzed by SDS-PAGE followed by Western blotting detecting ELYS. Compare Fig. 1E.

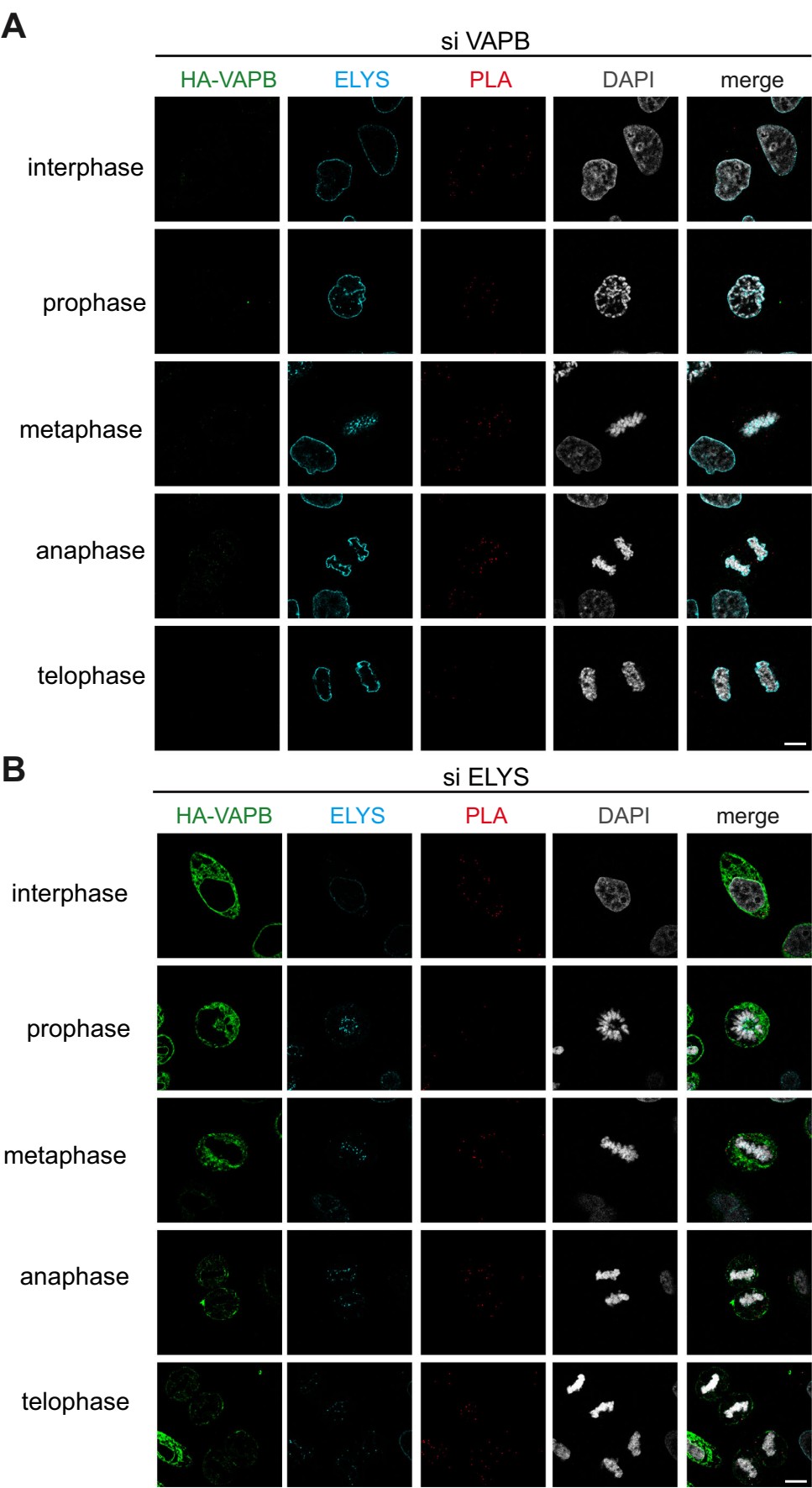

◄ **Figure EV2. Specificity of VAPB-ELYS interactions in anaphase shown by knockdown of either VAPB or ELYS.**

HeLa T-REx Flp-In cells stably expressing HA-VAPB were treated with siRNAs against VAPB (**A**) or ELYS (**B**) and synchronized and released as described in Fig. 6A and subjected to PLAs. Indirect immunofluorescence was performed to detect HA-VAPB and ELYS. Scale bar, 10 μm.

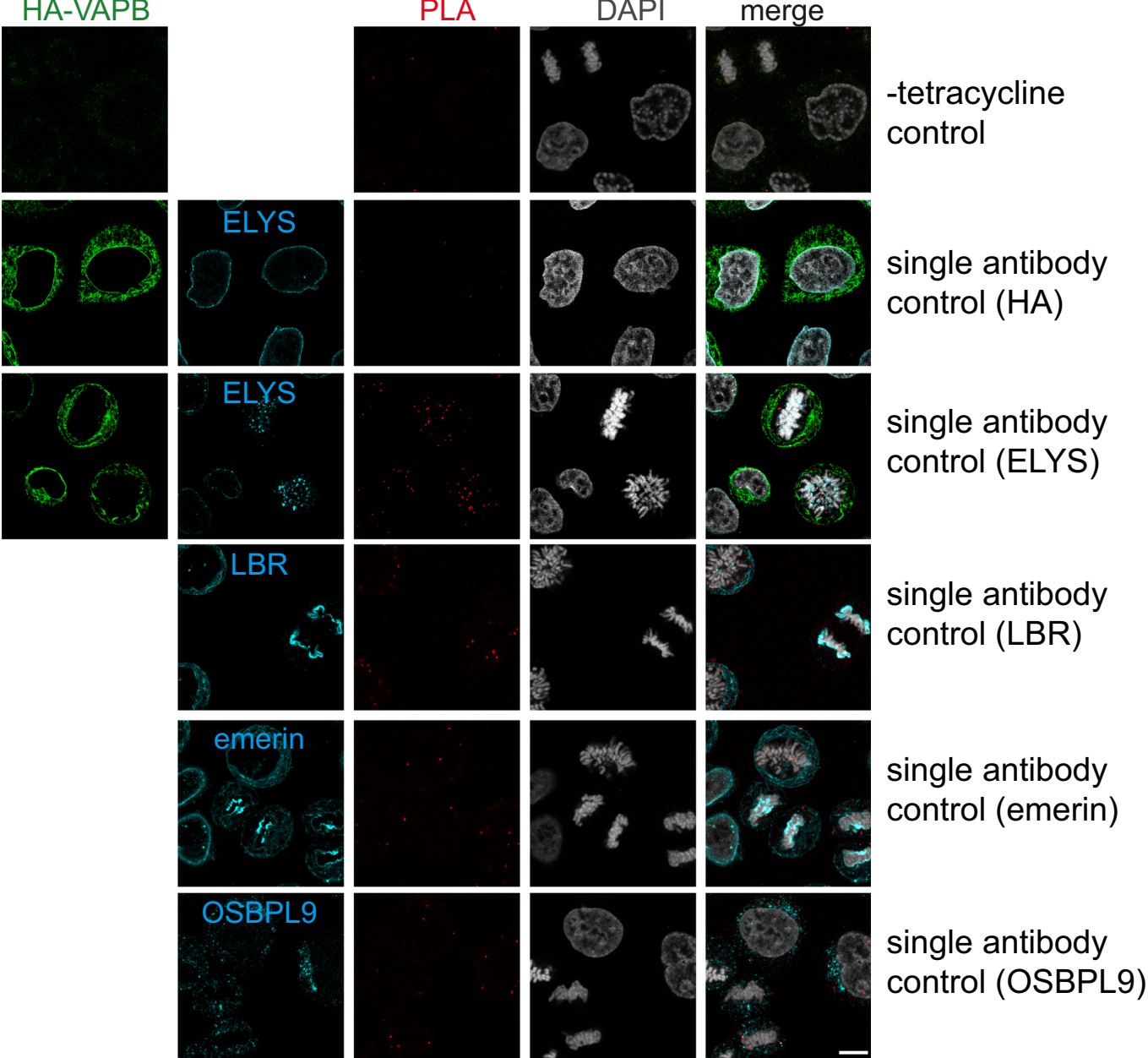

**Figure EV3.**   Single antibody controls using antibodies against HA, ELYS, LBR, emerin, and OSBPL9 to determine the specificity of PLA interactions. Scale bar, 10 μm.

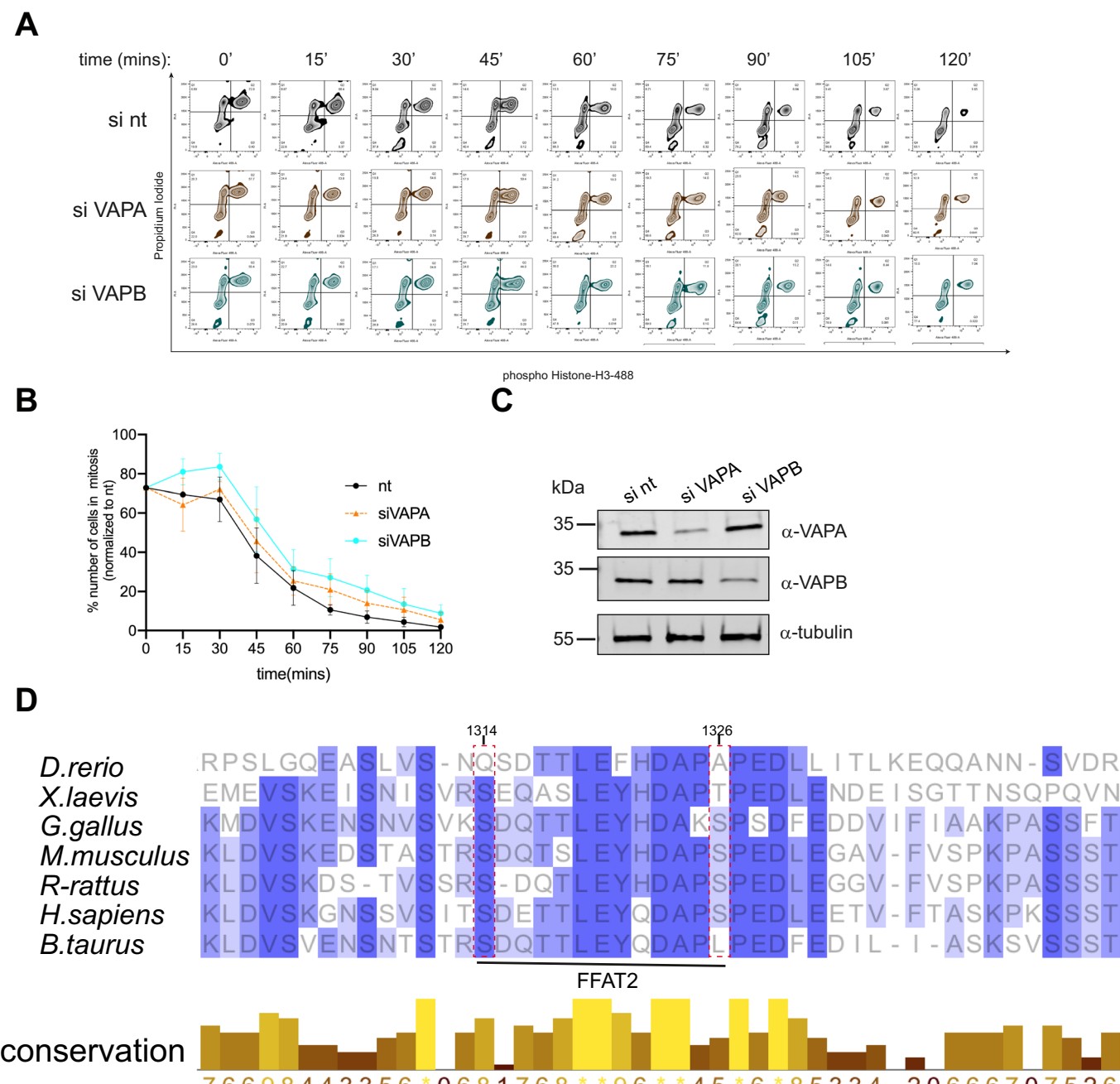

**Figure EV4. Effects of knockdown of VAPA and VAPB in mitosis.**

(**A**) HeLa P4 cells were transfected with siRNAs against VAPA (siVAPA) or VAPB (siVAPB) or non-targeting siRNAs (si nt) and synchronized and further treated as described in the legend to Fig. 8C. Cells were then analyzed by flow cytometry using propidium iodide and antibodies against phospho-Histone H3. Mitotic cells are observed in the upper right quadrant. (**B**) Quantification of the results in (**B**), indicating the percentage of mitotic cells as determined by flow cytometry. The data are shown as the mean (± standard deviation) of three biological replicates. (**C**) HeLa P4 cells subjected to knock down of VAPA or VAPB as indicated in (**A**) were analyzed by SDS-PAGE and Western blotting using antibodies against VAPA, VAPB, and tubulin. (**D**) Mutiple sequence alignment (MSA) of the FFAT2-motif of ELYS from *Danio rerio, Xenopus laevis, Gallus gallus, Mus musculus, Rattus rattus, Homo sapiens, Bos taurus*. Conservation of the amino acid residues are represented from 0 to 9 and *. Phosphorylation sites $S_{1314}$ and $S_{1326}$ are indicated by dotted red boxes. Jalview software (Waterhouse et al, 2009) was used to perform MSA.

