## [Peer Review File · EMBO Reports]

Phosphorylation of ELYS promotes its interaction with VAPB at decondensing chromosomes during mitosis

Christina James, Ulrike Möller, Christiane Spillner, Sabine König, Olexandr Dybkov, Henning Urlaub, Christof Lenz, and Ralph Kehlenbach

Corresponding author(s): *Ralph Kehlenbach (rkehlen@gwdg.de)*

Review Timeline:

Transfer from Review Commons:	15th Sep 23
Editorial Decision:	26th Sep 23
Revision Received:	9th Jan 24
Editorial Decision:	19th Feb 24
Revision Received:	23rd Feb 24
Accepted:	11th Mar 24

Editor: *Martina Rembold*

**Transaction Report: This manuscript was transferred to
EMBO reports following peer review at Review Commons.**

**Review
COMMONS**

Review #1

1. Evidence, reproducibility and clarity:

Evidence, reproducibility and clarity (Required)

Summary

The VAP proteins are well established as tail anchored proteins of the ER membrane. VAPs mediate co-operation between the ER and other organelles by creating a transient molecular tether with binding partners on opposing organelles to form a membrane contact site over which lipids and metabolites are exchanged. Proteins which bind VAPs generally contain a short FFAT motif, of varying sequence which binds the MSP domain of VAP. More recently the FFAT motif has been more extensively analysed in multiple different proteins and differential phosphorylation of the FFAT motif has been shown to either enhance or block VAP binding depending on the position of the phosphosite.

Recent work conducted by the authors demonstrated that a small population of VAPB is not exclusively localised to the ER and can also reach the inner nuclear membrane. They also identified ELYS as a potential interaction partner of VAPB in a screening approach. ELYS is a nucleoporin that can be found at the nuclear side of the nuclear envelope where it forms part of nuclear pore complexes. During mitosis, ELYS serves as an assembly platform that bridges an interaction between decondensing chromosomes and recruited nucleoporin subcomplexes to generate new nuclear pore complexes for post-mitotic daughter cells. In this manuscript, James et al seek to explore this enigmatic potential interaction between ELYS and VAPB to address why VAPB may be found at the inner nuclear membrane.

Peptide binding assays and some co-immunoprecipitation experiments are used to demonstrate that interactions occur via the MSP-domain of VAPB and FFAT-like motifs within ELYS. In addition, it is demonstrated that, for the ELYS FFAT peptides, the interaction is dependent on the phosphorylation status of serine residues of a particular FFAT-motif that can either promote or reduce its affinity to VAPB. Of most relevance is a serine in the acidic tract (1314) which, when phosphorylated increases VAPB binding. This is completely in line with what is already known about the FFAT motif and so is not surprising, in particular when using a peptide in an in vitro assay.

The authors then utilise cell synchronisation techniques to provide evidence that both

phosphorylation of ELYS and its binding to VAPB are heightened during mitosis. Immunofluorescence and proximity ligation assays are used to demonstrate that the proteins co-localise specifically during anaphase and at the non-core regions of segregating chromosomes.

The manuscript is concluded by investigating the effect of VAPB depletion on mitosis with some evidence to suggest that transition from meta-anaphase is delayed and defects such as lagging chromosomes are observed.

****Major comments****

Overall, this manuscript is well written and the data presented in Figures 1-3 convincingly show the nature of the interaction between ELYS and VAPB. Clearly the proteins interact via FFAT motifs and this interaction appears to be enhanced during mitosis. However, the work as is, relies heavily on peptide binding assays and would benefit from additional experiments to further support the results. The authors need to more clearly show that this specific phosphorylation happens during mitosis, they may have this data but it is not clearly explained. In addition, the data that VAPB-ELYS interaction contributes to temporal progression of mitosis (as per the title) is not sufficiently clear. VAPB silencing appears to have some impact on mitosis but this is not the same thing. So this section needs to be strengthened before this statement can be made.

The authors claim that the study "suggests an active role of VAPB in recruiting membrane fragments to chromatin and in the biogenesis of a novel nuclear envelope during mitosis". Given the data presented in Figures 4 and 5, this appears to be rather speculative with little evidence to support it, so data should be provided or this statement toned down. Currently, without additional supporting data the authors may wish to revise the overarching conclusions of the study and change the title.

****Specific points.****

Peptide pull down assays clearly show which FFAT-like motifs are important in facilitating binding. The co-immunoprecipitation systems used in Figure 2 also provide useful information on the interaction in a cell context. The authors should combine these findings by introducing full length ELYS mutants with altered FFAT-like motifs into their stably expressing GFP-VAPB HeLa cell line and then performing Co-IPs to help identify which FFAT motif/s drive the mitotic interaction. Other mutants of ELYS harbouring either phosphomimetic or phospho-resistant residues may also be introduced to further investigate mechanisms of the molecular

switch in a cellular environment to support the work currently done with peptides alone. This is an obvious gap in the work which, based on the other data the authors have shown, should presumably be straightforward and would also lead directly into the next major point.

- Whilst silencing VAPB does appear to delay mitosis, no reference is made to ELYS throughout Figure 5 nor as part of its associated discussion. Given that VAPB has more than 250 proposed binding partners, the observed aberration of mitotic progression could result from a huge number of indirect processes. Further work is needed to link the experiment specifically to the VAPB-ELYS interaction and not just loss of VAPB. We would suggest generating a complementation system where ELYS is either knocked out or silenced and then wild-type ELYS and an ELYS FFAT mutant (which cannot interact with VAPB), and/or a phospho mutant (whose interaction cannot be regulated during mitosis) are introduced. Then the observed effects can be better attributed to the VAPB-ELYS interaction and not just loss of VAPB.

- The immunofluorescence and PLA results in Figure 4 could be strengthened by including other ER markers. This would show that co-localisation of ELYS at the non-core region is specific to VAPB protein, not any ER protein or rather than an artefact of the ER being pushed out of the organelle exclusion zone during mitosis and therefore 'bunching' at the periphery of the nuclear envelope. It would be worthwhile repeating these experiments with candidates such as VAPA, other ER membrane proteins or at least GFP-KDEL, to make this phenomenon more convincing. As part of this the authors should ideally generate a complemented ELYS KO (see point above) to avoid the residual activity attributed to endogenous background in the PLA Figure 4E.

- Authors should clarify if the phosphorylation events (in particular S1314) only occur or are increased during mitosis. This may be data they have from the MS experiment in Figure 3 or it could also be shown using a phospho-antibody (although this can be challenging if a suitable antibody cannot be made).

- The authors should clarify why they need to do these semi in-vitro assays with purified GST-VAPB-MSP on beads and then lysates added and not just a standard co-IP. If this is simply signal intensity due to a very small proportion of VAPB binding to ELYS then this is fine but this should be stated and it should be made clear that ELYS is not a major binding partner - most of VAPB is on the ER. Otherwise, this is misleading.

I estimate that the suggested alterations above would incur approximately 3-6 months of additional experimental work, depending on if KO cell lines were required.

****Minor comments****

- To show that the observed interactions and potential role of VAPB-ELYS interaction is universal it would be useful to have at least a subset of experiments also shown in another cell line or system - this is now also a requirement for some journals.
- Consider re-wording the title of the manuscript to better reflect the data presented within the study. Alternatively, provide further evidence that VAPB-ELYS interactions directly affect temporal progression of mitosis to validate this claim, as discussed above.
- Quantification of blots in Figure 2A could allow measurement of relative binding affinities between VAPB-ELYS throughout the cell cycle. The same could be applied to the effect of phosphorylation on binding affinity in Figure 2D.
- The cells used are never clearly mentioned in the text - I assume this is always in HeLa but this should be added in all cases for clarity
- Page 8: "As shown in Fig. 2A, a large proportion of GFP-VAPB was precipitated under our experimental conditions." - I don't understand how this is shown in this figure as the non-bound fraction is not shown?
- Please provide some controls to demonstrate the extent to which the samples used are asyn, G1/M or M.
- Page 9 - why are Phos-tag gels not shown as this would make this result more convincing?
- Figure 3A - I find the SDS-PAGE gel confusing. Why not show the whole gel and why is the band size apparently reduced in the mitotic fraction when previously it was increased (by phosphorylation)? It would also be useful to see if there were any other band shifts.
- "FFAT-2 of ELYS is regulated by phosphorylation" The way you have setup the experiment leads the reader to think you are going to show which sites are differentially phosphorylated in mitosis, but then this is not the case - so there seems no purpose to doing the experiment this way. If you used TMT MS approach you would be able to potentially quantify the change in phosphorylation at the FFAT motif sites in mitosis. Otherwise what is the purpose of using these 2 samples, mitotic and AS?
- For all of the antibodies used, in particular for the PLA, please provide evidence of validation of the antibodies.
- Just a minor point to consider - In the methods for your lysis buffer you use 400mM NaCl - might this slightly reduce the VAPB-FFAT interaction? Worth considering reducing this?
- "The rather small difference observed between the wild-type and the mutant protein observed in this experiment probably results from the presence of endogenous VAPB in the stable cell lines, which could form dimers with the exogenous HA-tagged versions." If this is the case then please demonstrate that this is happening, or use the KO approach in the major points above.

- "we now show that the proteins can indeed interact with each other, without the need for additional bridging factors (Figs. 1 and 3)." You show that the peptides can bind - but this is not the same thing as the peptide in the full context of the protein - so this should be toned down or removed.
- "Remarkably, this region is highly conserved between species, suggesting that it is important for protein functions (data not shown)". Please show the alignments so the reader can judge for themselves. It is conserved in ALL species and the phosphosites are also conserved??
- "In our experiments, knockdown of VAPA alone did not lead to a delay in mitosis (data not shown). " Why not show this data - as this is a very interesting and potentially important observation? Also add the validation of knockdown of VAPA.
- I find the end to the discussion to the paper rather abrupt. It would be interesting to discuss further how VAPB, but not apparently VAPA reaches the INM and if so why this function is required of an ER adaptor and not another more obvious adaptor protein. In short - why would VAPB be performing this role?

****Referees cross-commenting****

I agree with the comments of the other reviewers, and they are very much in line with my own review. We all seem convinced that VAPB binds ELYS via a pFFAT, and that this interaction is enhanced during mitosis. However the role of this interaction in mitotic progression remains unclear and based on this data should not be claimed in the title or discussion of the paper.

2. Significance:

Significance (Required)

Overall, if the manuscript could be improved with the suggested changes, then this could be a considerable conceptual advance in how we understand the VAP proteins, showing functions beyond those as an ER adaptor. This would be significant for the field.

In the context of the existing literature the work does not advance our knowledge of FFAT-VAP interactions, this has already been shown, but it would give a nice example of how this can be regulated during mitosis and how VAP can contribute beyond just as an ER adaptor at membrane contact sites.

There would be a wide audience in the cell biology field and more widely as mutations in VAPB cause a form of ALS, and many people are working in this area.

My field of expertise is in organelle cell biology and membrane contact sites.

3. How much time do you estimate the authors will need to complete the suggested revisions:

Estimated time to Complete Revisions (Required)

(Decision Recommendation)

Between 3 and 6 months

No

Review #2

1. Evidence, reproducibility and clarity:

Evidence, reproducibility and clarity (Required)

****Summary:****

In this study, James et al. follow up on their prior discovery that the ER contact site protein VAPB localizes to the nuclear envelope and is a putative binding partner of the nucleoporin ELYS, which coordinates nuclear envelope reformation (NER) with nuclear pore complex (NPC) biogenesis at mitotic exit and is also a constituent of the nuclear facing Y-complexes of mature NPCs. Using a series of complementary

biochemical approaches the authors 1) demonstrate that VAPB and ELYS directly interact, 2) map the binding sites on ELYS that are sufficient to bind VAPB, 3) show that mutations that disrupt VAPB-FFAT motif binding also abrogate binding to ELYS including of the full-length protein, 4) define mitotic phosphorylation sites on VAPB-bound ELYS, 5) demonstrate that phosphorylation of ELYS, specifically at the FFAT2 motif, is required for binding to VAPB, and 6) demonstrate that the phosphorylation of ELYS that regulates VAPB binding occurs in mitosis. Turning to cell biology, the authors find that VAPB, which is an established ER protein, has some preference for non-core regions during NER (like ELYS). In addition, PLA analysis suggests that the interaction of VAPB and ELYS is most robust during anaphase and is somewhat disrupted when the binding of VAPB to FFAT motifs is lost due to targeted mutation. Last, the authors demonstrate that depletion of VAPB leads to metaphase delay and lagging chromosomes.

****Major comments:****

The data supporting direct binding of peptides encoding the FFAT 1 and FFAT2 motif derived from ELYS to VAPB in a manner similar to other FFAT sequences is strong, as is the effect of phosphorylation of FFAT1 on the strength of this interaction.

The evidence supporting the mitotic-specific nature of the ELYS-VAPB interaction is strong, and that this interaction is direct, is also strong, and was rigorously tested using a combination of endogenous expression, heterologous expression, and recombinant protein approaches. Moreover, the sensitivity of this interaction to established mutations in VAPB abrogating FFAT interactions reinforces the outlined underlying biochemical interaction mechanism. The essentiality of ELYS phosphorylation (and therefore the mechanism underlying the mitotic specificity of the interaction) is also strongly supported by the data using phosphatase treatment. Although it is an intuitive model, whether the cell biological evidence support the simplest view that the ELYS-VAPB complex bridges the nuclear envelope to chromatin during NER in late anaphase / at mitotic exit is far less solid and, at a minimum, alternative models should be considered/discussed. For example, how a delay in metaphase in the siVAPB condition is consistent with a role in NER, which occurs exclusively post-metaphase, is unclear. Is it not possible that the VAPB-ELYs complex is regulated by phosphorylation during mitotic progression such that VAPB and/or ELYS can only exert their biological effects when released from the complex? In other words, might ELYS be licensed to act in NER only when it is released from VAPB, which could prevent premature NER/NPC biogenesis? Subtleties of when during mitosis the phosphorylation occurs is challenging, and it could be that the anaphase A to anaphase B transition, when many mitotic entry phosphorylation events begin to be reversed, could be relevant here. Along these lines, in Fig. 5 how the

VAPB knock-down does or does not recapitulate the phenotype of an ELYS knock-down in this cell type (and the effect of the combination, to address epistasis) is needed for context, as is whether VAPB knock-down affects ELYS distribution in mitosis. ELYS knock-down would also be very beneficial for the PLA analysis to establish the "floor" of measurable signal. Last, it is also possible that VAPB has other roles in mitosis that should be acknowledged - for example although it is in yeast, it is relevant that a VAPB orthologue Scs2 is required for normal nuclear envelope expansion in mitosis by regulating SUMOylation (Ptak, Saik et al., JCB, 2021 and Saik et al., JCB, 2023) - this work should be referenced as well.

Of course, the ideal experiment would be one in which an ELYS knock-down is complemented with a resistant form that encodes the S to A mutations in the FFAT2 region to assess its localization and to see if it can complement the knock-out function of ELYS in post-mitotic NPC assembly or, as suggested by a sequestration model, it can drive the same metaphase delay seen upon VAPB knock-down. This is technically challenging for sure, particularly given the size of the ELYS gene, but it would address the cell biological function of this interaction in the most direct manner. Several other observations that could warrant further comment or study include 1) is there a VAPB signal at the metaphase poles as suggested by Fig. 4A and, if so, could this represent a distinct mitotic function?; 2) Does the HA-VAPB KD/MD mutant localize differently in mitosis compared to the WT - it appears that it might be less enriched in non-core regions (Fig. 4E)?; 3) does VAPB alter post-mitotic NPC biogenesis/number?

****Minor comments:****

I would suggest avoiding the use of "novel" when describing newly assembled NPCs or post-mitotic nuclear envelope reformation, as its other meaning of "non-standard" makes this wording confusing.

It is unclear whether when the authors state that ELYS localizes "to the nuclear side of the nuclear envelope" they are referring to the nuclear aspect of the NPC and/or a separate pool at the INM - please edit to clarify.

More descriptive y-axes for the plots in Fig. 4F and 5F and related legends would be useful; although the details are in the methods section, it would be nice not to have to hunt them down. Also, please clarify the meaning of blue and orange points in Fig. 4F.

2. Significance:

Significance (Required)

General assessment: The biochemical analysis is rigorous and compelling and establishes the mitotic-specific interaction of VAPB and ELYS including detailed information about the binding interface and its regulation by phosphorylation. The new insight provided into the function of this VAPB-ELYS interaction is somewhat less well developed as the current manuscript, in its current form, does not yet mechanistically define the function of the VAPB-ELYS interaction in mitosis.

Advance: Conceptually, to the best of my knowledge, the idea that VAPB contributes to mitosis in mammalian cells is novel and is therefore impactful and will motivate further work. As the authors connect VAPB biochemically to ELYS, an established factor that promotes the coordination of NER and NPC biogenesis, this interaction is likely to be mechanistically important, although the specific details by which this interaction facilitate normal mitotic progression is not yet clear.

Audience: This work will be of interest to a broad swath of cell biologists including those interested in NPCs, the nuclear envelope, the ER, membrane remodeling, and chromosome segregation.

My expertise is in nuclear envelope dynamics, nuclear pore complexes, and chromatin organization.

3. How much time do you estimate the authors will need to complete the suggested revisions:

Estimated time to Complete Revisions (Required)

(Decision Recommendation)

Between 3 and 6 months

No

Review #3

1. Evidence, reproducibility and clarity:

Evidence, reproducibility and clarity (Required)

This MS contains carefully carried out and well controlled experiments describing a new pFFAT in ELYS. There is a similarly convincing demonstration of functionally relevant colocalisation by proximity ligation assay (PLA), particularly that both ELYS and VAP are nuclear envelope proteins in interphase without interacting (neg control in Fig 4D).

****Major Issue:**** Functional significance

A key conclusion is that experiments prove that "ELYS serves as the crucial initiation factor for post-mitotic NPC-assembly" (p5). However, evidence for this is lacking as this would require reconstitution of NPC assembly with a mutant form of ELYS carefully changing the FFAT motif (e.g. 1321A 1324E) and exclusion of other probable VAP targets in experiments with mutant VAP.

VAPs are among the proteins with the highest number of documented interactors (see Huttlin 2015/7 etc, e.g. PMID 26186194), so knocking down VAP may have pleiotropic effects and quite indirect read-outs in many aspects of cell function. In addition, for this work specifically there are other NE proteins that are known interactors of VAP: Emerin (EMD) and LBR both interact with VAP (high-throughput data, VAPA and VAPB). EMD has a motif similar to the canonical phospho-FFAT: 98 SYFTTTRT 104. LBR has no motif. These findings should not be overlooked in this work. For example, was the interaction with emerin (page 4) sensitive to mutating VAP or ELYS? Could the effect seen in Figure 5 result from interactions with proteins other than ELYS?

Further experiments should be carried out to justify all statements in the current MS of functional significance. Instead of doing more experiments, an alternative for the authors would be to describe the current set of results more cautiously. However, that would require changing much of the impact of the current MS, from the title onwards.

****Moderate Issue:**** VAPA

From the start of the Introduction and some elements of the Discussion, include VAPA in equal measure with VAPB. When describing interactions of ELYS with VAP note that Huttlin et al., reported interactions twice for each of VAPA and VAPB. When describing own results (James et al. 2019) and those of others (Saiz-Ros et al., 2019) that focused on VAPB, clarify if the authors' view is that VAPA would (or would not) have the same interaction.

Is there any evidence that only VAPB is on NE? Note that some refs in the Introduction relate to VAPA: Mesmin (not VAPB); ACBD5: although article titles refer to VAPB, early work (10.1083/jcb.201607055) showed almost identical involvement of VAPA. Also, this redundancy likely explains "function of VAPB in mitosis is not essential," (in Discussion). The lack of effect of VAPA knock-down may indicate that in these cells VAPB is dominant, but does not exclude a role for VAPA when VAPB is reduced. That might be tested by depleting both. Even following that, there is MOSPD2 to consider

Other aspects of the writing

"two amino acid residues are crucial for the interaction (VAPB K87 and M89)." This is wrong. Many residues are critical, these are merely 2 of possibly >10 that were chosen by Kaiser et al (2005) to create their non-binder.. Others have used different mutations to block FFAT binding.

"They may exhibit a certain binding preference to specific members of the VAP ... family...". I cannot think of any example. I note no citation is given.

When listing many or all MSP proteins, the text should state that MOSPD2 is uniquely close to VAPA/B. CFAP65 is typically not mentioned in the VAP-like lists as it does not have any of the conserved sequence that binds FFAT. If however the authors wish to include all human MSP domain protein, they should also include Hydin.

Slightly wrong to cite De Vos et al., 2012 about PTPIP51's FFAT as that paper makes no mention of the motif. Better pick Di Mattia (again)

On VAPB (and also A) on INM: there are references to be cited esp. relating to

intranuclear Scs2 in yeast (Brickner et al 2004, Ptak et al 2021)

Citations for VAP at ER-mito contacts "De Vos et al., 2012; Gómez-Suaga et al., 2019; Stoica et al., 2014)". These all refer to the same bridging protein, PTPIP51. Reduce to one citation. Then mention other proteins at the same site VPS13A, mitoguardin(MIGA)-2 ...

"The domain interacts with characteristic peptide sequences ..." add citation to this sentence

"Several variants of such motifs have been described: (i)" ... "(ii)": (i) and (ii) are entirely unlinked. Delete these and also "Several variants of such motifs have been described." Which is repeated later

"FFAT-like motifs come in different flavors and may even lack the two phenylalanine residues (Murphy and Levine, 2016)": while motifs can tolerate variation at both positions, this text is misleading as it implies much more variation than is known. The 1st F can only be conservatively substituted (Y).

****Minor aspects in Results:****

ORP1L peptide as positive control: cite Kaiser 2005

Was phosphoproteomics done in such a way as to find peptides that have both S1314 and S1326?

Figure 4D, row 2: Comment on intranuclear staining in Prophase (at approx 4 o'clock) of both ELYS & VAP that is PLA positive

****Referees cross-commenting****

I agree with this point from Reviewer #1. We all agree that the main issue can be resolved experimentally to determine the effect of subtle point mutations in ELYS. Both other reviewers have done a good job in finding issues with the experiments that can also be addressed.

2. Significance:

Significance (Required)

This work documents an interaction between the protein ELYS, that is involved in the reformation of nuclear pore complexes after mitosis, and the ER membrane protein VAPB. The interactions was previously known through high-throughput studies, along with many 100's of others for VAP, but here it is studied in detail and with care, identifying how the motif is induced by phosphorylation of ELYS. The two proteins are co-localised using convincing proximity ligation assays. This biochemistry and cell biological localisation is well done.

Functional experiments then show that VAP (in this case VAPB) knock-down affects mitosis and chromosome segregation. While the result is incontrovertible, it has many possible interpretations, mainly because VAP has hundreds of interactions, including with multiple proteins involved in mitosis beyond just ELYS. This means that there are major limitations on how the interaction and co-localisation should be interpreted, reducing the advance associated with the current manuscript to incremental, and the limiting the audience to specialized.

3. How much time do you estimate the authors will need to complete the suggested revisions:

Estimated time to Complete Revisions (Required)

(Decision Recommendation)

Between 3 and 6 months

Yes

Dear Prof. Kehlenbach

Thank you for the submission of your research manuscript to our journal. I have now carefully read your manuscript, the referee comments from Review Commons and your revision plan.

Based on these analyses, we would like to invite you to revise your manuscript along the lines you describe in the revision plan. I want to stress, however, that the functional data on VAPB's effect on mitosis need to be strengthened and at least some evidence should be provided that the effect is specific and depends on its interaction with ELYS. You explain that the suggested epistasis experiments are not possible, because expression of full-length ELYS is not feasible but as far as I understand, you plan to address this concern using overexpression of fragments of ELYS.

Taken together, we would like to invite you to revise your manuscript with the understanding that the referee concerns (as detailed above and in their reports) must be fully addressed and their suggestions taken on board. Please address all referee concerns in a complete point-by-point response. Acceptance of the manuscript will depend on a positive outcome of a second round of review. It is EMBO Reports policy to allow a single round of revision only and acceptance or rejection of the manuscript will therefore depend on the completeness of your responses included in the next, final version of the manuscript.

We realize that it is difficult to revise to a specific deadline. In the interest of protecting the conceptual advance provided by the work, we recommend a revision within 3 months (December 28th). Please discuss the revision progress ahead of this time with the editor if you require more time to complete the revisions.

I am also happy to discuss the revision further via e-mail or a video call, if you wish.

*****IMPORTANT NOTE:

We perform an initial quality control of all revised manuscripts before re-review. Your manuscript will FAIL this control and the handling will be delayed IF the following applies:

- 1) A data availability section providing access to data deposited in public databases is missing. If you have not deposited any data, please add a sentence to the data availability section that explains that.
- 2) Your manuscript contains statistics and error bars based on $n=2$. Please use scatter blots in these cases. No statistics should be calculated if $n=2$.

When submitting your revised manuscript, please carefully review the instructions that follow below. Failure to include requested items will delay the evaluation of your revision.*****

- 1) a .docx formatted version of the manuscript text (including legends for main figures, EV figures and tables). Please make sure that the changes are highlighted to be clearly visible.
- 2) individual production quality figure files as .eps, .tif, .jpg (one file per figure). Please download our Figure Preparation Guidelines (figure preparation pdf) from our Author Guidelines pages <https://www.embopress.org/page/journal/14693178/authorguide> for more info on how to prepare your figures.
- 3) a .docx formatted letter INCLUDING the reviewers' reports and your detailed point-by-point responses to their comments. As part of the EMBO Press transparent editorial process, the point-by-point response is part of the Review Process File (RPF), which will be published alongside your paper.
- 4) a complete author checklist, which you can download from our author guidelines (). Please insert information in the checklist that is also reflected in the manuscript. The completed author checklist will also be part of the RPF.
- 5) Please note that all corresponding authors are required to supply an ORCID ID for their name upon submission of a revised

manuscript (). Please find instructions on how to link your ORCID ID to your account in our manuscript tracking system in our Author guidelines
()

6) We replaced Supplementary Information with Expanded View (EV) Figures and Tables that are collapsible/expandable online. A maximum of 5 EV Figures can be typeset. EV Figures should be cited as 'Figure EV1, Figure EV2' etc... in the text and their respective legends should be included in the main text after the legends of regular figures.

7) Please note that a Data Availability section at the end of Materials and Methods is now mandatory. In case you have no data that requires deposition in a public database, please state so instead of refereeing to the database. See also < <https://www.embopress.org/page/journal/14693178/authorguide#dataavailability>>. Please note that the Data Availability Section is restricted to new primary data that are part of this study.

Additional information on source data and instruction on how to label the files are available .

10) Figure legends and data quantification:
The following points must be specified in each figure legend:

- the name of the statistical test used to generate error bars and P values,
 - the number (n) of independent experiments (please specify technical or biological replicates) underlying each data point,
 - the nature of the bars and error bars (s.d., s.e.m.)
- If the data are obtained from n {less than or equal to} 5, show the individual data points in addition to the SD or SEM.
- If the data are obtained from n {less than or equal to} 2, use scatter blots showing the individual data points.

See also the guidelines for figure legend preparation:
<https://www.embopress.org/page/journal/14693178/authorguide#figureformat>

11) Our journal encourages inclusion of *data citations in the reference list* to directly cite datasets that were re-used and obtained from public databases. Data citations in the article text are distinct from normal bibliographical citations and should directly link to the database records from which the data can be accessed. In the main text, data citations are formatted as follows: "Data ref: Smith et al, 2001" or "Data ref: NCBI Sequence Read Archive PRJNA342805, 2017". In the Reference list, data citations must be labeled with "[DATASET]". A data reference must provide the database name, accession number/identifiers and a resolvable link to the landing page from which the data can be accessed at the end of the reference. Further instructions are available at .

12) All Materials and Methods need to be described in the main text. We would encourage you to use 'Structured Methods', our new Materials and Methods format. According to this format, the Materials and Methods section should include a Reagents and Tools Table (listing key reagents, experimental models, software and relevant equipment and including their sources and relevant identifiers) followed by a Methods and Protocols section in which we encourage the authors to describe their methods using a step-by-step protocol format with bullet points, to facilitate the adoption of the methodologies across labs.

More information on how to adhere to this format as well as downloadable templates (.doc or .xls) for the Reagents and Tools Table can be found in our author guidelines: <
<https://www.embopress.org/page/journal/14693178/authorguide#manuscriptpreparation>>.

An example of a Method paper with Structured Methods can be found here: .

13) As part of the EMBO publication's Transparent Editorial Process, EMBO Reports publishes online a Review Process File to accompany accepted manuscripts. This File will be published in conjunction with your paper and will include the referee reports, your point-by-point response and all pertinent correspondence relating to the manuscript.

Yours sincerely,

Original Manuscript number: RC- 2023-02064

Corresponding author(s): Ralph, Kehlenbach

Dear Dr. Rembold, Dear Reviewers,

Thank you very much for the helpful comments on our manuscript and for giving us the opportunity to submit a revised version. We address and clarify many of the issues that have been raised (our responses in blue; relevant changes in the revised text are also in blue) and include new and/or modified data, as requested. Importantly, we are shifting the focus of the manuscript, starting our results-section with an approach to specifically identify proteins that interact with VAPB during mitosis (as opposed to asynchronous cells; novel Figure 1). As a matter of fact, ELYS is a *major* protein that interacts with VAPB in mitotic cells. These data are based on novel, in-depth analyses by quantitative mass spectrometry. We then continue to characterize this interaction in detail, add several controls as requested and also new data, making use of novel cells that stably express different versions of an ELYS-fragment. Importantly, the pertinent phosphomimetic S1314D-mutant of this fragment binds more tightly to endogenous VAPB, compared to the wild-type.

We also changed the title of our manuscript to “Phosphorylation of ELYS controls its interaction with VAPB at decondensing chromosomes during mitosis” to better reflect our findings.

Furthermore, we included additional authors, who were involved in the new experiments.

Here our detailed comments:

Reviewer #1 (Evidence, reproducibility and clarity (Required)):

Summary

The VAP proteins are well established as tail anchored proteins of the ER membrane. VAPs mediate co-operation between the ER and other organelles by creating a transient molecular tether with binding partners on opposing organelles to form a membrane contact site over which lipids and metabolites are exchanged. Proteins which bind VAPs generally contain a short FFAT motif, of varying sequence which binds the MSP domain of VAP. More recently the FFAT motif has been more extensively analysed in multiple different proteins and differential phosphorylation of the FFAT motif has been shown to either enhance or block VAP binding depending on the position of the phosphosite.

Recent work conducted by the authors demonstrated that a small population of VAPB is not exclusively localised to the ER and can also reach the inner nuclear membrane. They also identified ELYS as a potential interaction partner of VAPB in a screening approach. ELYS is a nucleoporin that can be found at the nuclear side of the nuclear envelope where it forms part of nuclear pore complexes. During mitosis, ELYS serves as an assembly platform that bridges an interaction between decondensing chromosomes and recruited nucleoporin subcomplexes to generate new nuclear pore complexes for post-mitotic daughter cells. In this manuscript, James et al seek to explore this enigmatic potential interaction between ELYS and VAPB to address

why VAPB may be found at the inner nuclear membrane.

Peptide binding assays and some co-immunoprecipitation experiments are used to demonstrate that interactions occur via the MSP-domain of VAPB and FFAT-like motifs within ELYS. In addition, it is demonstrated that, for the ELYS FFAT peptides, the interaction is dependent on the phosphorylation status of serine residues of a particular FFAT-motif that can either promote or reduce its affinity to VAPB. Of most relevance is a serine in the acidic tract (1314) which, when phosphorylated increases VAPB binding. This is completely in line with what is already known about the FFAT motif and so is not surprising, in particular when using a peptide in an in vitro assay.

The authors then utilise cell synchronisation techniques to provide evidence that both phosphorylation of ELYS and its binding to VAPB are heightened during mitosis. Immunofluorescence and proximity ligation assays are used to demonstrate that the proteins co-localise specifically during anaphase and at the non-core regions of segregating chromosomes.

The manuscript is concluded by investigating the effect of VAPB depletion on mitosis with some evidence to suggest that transition from meta-anaphase is delayed and defects such as lagging chromosomes are observed.

Major comments

Overall, this manuscript is well written and the data presented in Figures 1-3 convincingly show the nature of the interaction between ELYS and VAPB. Clearly the proteins interact via FFAT motifs and this interaction appears to be enhanced during mitosis. However, the work as is, relies heavily on peptide binding assays and would benefit from additional experiments to further support the results. The authors need to more clearly show that this specific phosphorylation happens during mitosis, they may have this data but it is not clearly explained.

We now performed a more detailed analysis of phosphorylation sites in ELYS - but unfortunately, we could not detect the relevant sites (foremost S1314), perhaps because the corresponding tryptic fragment is too large to be detected by our methods (there are other sites that had been identified before but escaped detection). Nevertheless, our new data using cells that stably express wild-type or mutant fragments of ELYS suggest that phosphorylation of this very site indeed promotes binding to VAPB, also in the complex milieu of a cell or a cell lysate (novel Figure 5D-F). As requested, we also performed binding experiments with a longer ELYS-fragment purified from bacteria (instead of just peptides), which further strengthen our results (novel Figure 5A-C).

In addition, the data that VAPB-ELYS interaction contributes to temporal progression of mitosis (as per the title) is not sufficiently clear. VAPB silencing appears to have some impact on

mitosis but this is not the same thing. So this section needs to be strengthened before this statement can be made.

We added additional data to this section (see also below) and changed the title of the manuscript to better convey our findings.

The authors claim that the study "suggests an active role of VAPB in recruiting membrane fragments to chromatin and in the biogenesis of a novel nuclear envelope during mitosis". Given the data presented in Figures 4 and 5, this appears to be rather speculative with little evidence to support it, so data should be provided or this statement toned down. Currently, without additional supporting data the authors may wish to revise the overarching conclusions of the study and change the title.

We agree and toned down our statement a bit. As mentioned above, we now slightly changed the focus of the manuscript, starting with the observation that ELYS is a *major* binding partner of VAPB in mitosis, and also changed the title. We now suggest that VAPB plays a role in mitosis upon recruitment to or release from ELYS at the non-core region of the chromatin in a phosphorylation-dependent manner. Alternative scenarios are presented in the discussion.

Specific points.

Peptide pull down assays clearly show which FFAT-like motifs are important in facilitating binding. The co-immunoprecipitation systems used in Figure 2 also provide useful information on the interaction in a cell context. The authors should combine these findings by introducing full length ELYS mutants with altered FFAT-like motifs into their stably expressing GFP-VAPB HeLa cell line and then performing Co-IPs to help identify which FFAT motif/s drive the mitotic interaction. Other mutants of ELYS harboring either phosphomimetic or phospho-resistant residues may also be introduced to further investigate mechanisms of the molecular switch in a cellular environment to support the work currently done with peptides alone. This is an obvious gap in the work which, based on the other data the authors have shown, should presumably be straightforward and would also lead directly into the next major point.

We tried several times, but did not succeed in obtaining stable cells expressing full-length ELYS (wild-type or mutants), which would be required for a faithful biochemical analysis. Instead, we generated cells expressing ELYS-fragments (wild-type and the relevant mutants), which had an effect on the cell cycle (new Figure 8A). Furthermore, our co-immunoprecipitation experiments show that the phosphomimetic mutation (ELYS-1018-1642-S1314D) promotes binding of endogenous VAPB. For the S1314A-mutant, which cannot be phosphorylated at this position, reduced binding was observed, compared to the wild-type-fragment (novel Figure 5E, F). Interestingly, an effect on cell cycle progression upon knockout of VAPB was recently published during the review process of our manuscript (Faria Assoni et al., 2023; this paper is now also cited in the Discussion).

- Whilst silencing VAPB does appear to delay mitosis, no reference is made to ELYS throughout Figure 5 nor as part of its associated discussion. Given that VAPB has more than 250 proposed

binding partners, the observed aberration of mitotic progression could result from a huge number of indirect processes. Further work is needed to link the experiment specifically to the VAPB-ELYS interaction and not just loss of VAPB. We would suggest generating a complementation system where ELYS is either knocked out or silenced and then wild-type ELYS and an ELYS FFAT mutant (which cannot interact with VAPB), and/or a phospho-mutant (whose interaction cannot be regulated during mitosis) are introduced. Then the observed effects can be better attributed to the VAPB-ELYS interaction and not just loss of VAPB.

As mentioned above, cells expressing full-length ELYS could not be obtained, despite several trials. Knocking down VAPB did not affect ELYS localization. To further address a possible role of VAPB-ELYS interactions during mitosis, we used our novel cells stably expressing ELYS-fragments and analyzed them by flow cytometry (novel Figure 8A). The wild-type ELYS-fragment had a clear effect on the cell cycle and the FFAT2-motif of ELYS seemed to modulate this effect. The precise role of individual sites, however, is difficult to analyze, because ELYS contains so many possible phosphorylation sites.

- The immunofluorescence and PLA results in Figure 4 could be strengthened by including other ER markers. This would show that co-localisation of ELYS at the non-core region is specific to VAPB protein, not any ER protein or rather than an artefact of the ER being pushed out of the organelle exclusion zone during mitosis and therefore 'bunching' at the periphery of the nuclear envelope. It would be worthwhile repeating these experiments with candidates such as VAPA, other ER membrane proteins or at least GFP-KDEL, to make this phenomenon more convincing. As part of this the authors should ideally generate a complemented ELYS KO (see point above) to avoid the residual activity attributed to endogenous background in the PLA Figure 4E.

We included other ER and NE markers (OSBPL9, LBR, emerin) in our PLA experiments, as suggested (novel Figures 7). We performed PLA using VAPB or ELYS knock downs and observed a decrease in the PLA signal (EV2). We also tried VAPA and GFP-KDEL in PLAs, but could not obtain reliable results, probably due to the quality of the antibodies used.

- Authors should clarify if the phosphorylation events (in particular S1314) only occur or are increased during mitosis. This may be data they have from the MS experiment in Figure 3 or it could also be shown using a phospho-antibody (although this can be challenging if a suitable antibody cannot be made).

See above. We repeated the MS-experiment, identified a few more sites but could not detect the relevant peptide (Discussion: "We note, however, that these sites were not among the phosphosites identified by us, possibly because the corresponding tryptic fragment (GNSSVSITS(1314)DETTLEYQDAPS(1326)PEDLEETVFTASKPK) is very large and may have escaped detection"). We also note that we did not identify the majority of previously described phospho-sites (>150) and concede that we cannot prove that S1314 is the only relevant phospho-site in ELYS. We believe, however, that our data show that it does indeed play a major role in controlling the interaction with VAPB.

The data in Dataset Table EV5 give some hints as to the relative levels of individual phosphopeptides (and make the point that there is more phosphorylation when the mitotic samples are analyzed). Since our peptide of interest is not in the list, however, we do not further elaborate on this.

- The authors should clarify why they need to do these semi in-vitro assays with purified GST-VAPB-MSP on beads and then lysates added and not just a standard co-IP. If this is simply signal intensity due to a very small proportion of VAPB binding to ELYS then this is fine but this should be stated and it should be made clear that ELYS is not a major binding partner - most of VAPB is on the ER. Otherwise, this is misleading.

We are using purified GST-VAPB-MSP now for the initial identification of specific binding partners of VAPB (novel Figure 1). Later, this approach gives us the opportunity to better control our experiment (using defined amounts of immobilized protein for wild-type and a mutant (Figure 2). Our data clearly show that in mitotic lysates, ELYS is actually a *prominent* binding partner of VAPB. As a matter of fact, the pattern of interacting proteins is very different in the two types of lysates. We tried to make this point clearer.

I estimate that the suggested alterations above would incur approximately 3-6 months of additional experimental work, depending on if KO cell lines were required.

Minor comments

- To show that the observed interactions and potential role of VAPB-ELYS interaction is universal it would be useful to have at least a subset of experiments also shown in another cell line or system - this is now also a requirement for some journals.

We performed similar experiments in HEK 293T cells and obtained similar results; novel Figure EV1B).

- Consider re-wording the title of the manuscript to better reflect the data presented within the study. Alternatively, provide further evidence that VAPB-ELYS interactions directly affect temporal progression of mitosis to validate this claim, as discussed above.

We changed the title (also according to the slightly changed focus of our planned manuscript).

- Quantification of blots in Figure 2A could allow measurement of relative binding affinities between VAPB-ELYS throughout the cell cycle. The same could be applied to the effect of phosphorylation on binding affinity in Figure 2D.

We performed these quantifications for three experiments and include them as new panels (novel Figures 2B and 2F).

- The cells used are never clearly mentioned in the text - I assume this is always in HeLa but this should be added in all cases for clarity

Yes, HeLa. We made this clearer in the text now.

- Page 8: "As shown in Fig. 2A, a large proportion of GFP-VAPB was precipitated under our experimental conditions." - I don't understand how this is shown in this figure as the non-bound fraction is not shown?

We compared the amount of precipitated protein to the input (10%) and concluded that a substantial portion of the total was indeed precipitated.

- Please provide some controls to demonstrate the extent to which the samples used are asyn, G1/M or M.

We show blots now (modified Figure 2A), detecting specific marker proteins.

- Page 9 - why are Phos-tag gels not shown as this would make this result more convincing?

We do show Phos-tag gels now (novel Figure Appendix S1).

- Figure 3A - I find the SDS-PAGE gel confusing. Why not show the whole gel and why is the band size apparently reduced in the mitotic fraction when previously it was increased (by phosphorylation)? It would also be useful to see if there were any other band shifts.

We now show a much larger portion of the gel (novel Figure 1). The most likely explanation here is that by our approach we enrich for phosphorylated versions of ELYS, which are less abundant in the lysate of interphase cells (and which also contains a certain percentage of lysed mitotic cells). Hence, the proteins migrate similarly in SDS-PAGE under both conditions.

- "FFAT-2 of ELYS is regulated by phosphorylation" The way you have setup the experiment leads the reader to think you are going to show which sites are differentially phosphorylated in mitosis, but then this is not the case - so there seems no purpose to doing the experiment this way. If you used TMT MS approach you would be able to potentially quantify the change in phosphorylation at the FFAT motif sites in mitosis. Otherwise what is the purpose of using these 2 samples, mitotic and AS?

Thank you very much for this comment. We repeated the MS-experiment to detect phosphorylation sites. Dataset Table EV5 now provides some information about the relative levels of phosphorylation in mitosis and in asynchronous cells (of the identified peptides). As mentioned above, however, we did not detect the peptide containing S1314.

- For all of the antibodies used, in particular for the PLA, please provide evidence of validation of the antibodies.

We added these data to the supplements (Fig. EV2 and 3).

- Just a minor point to consider - In the methods for your lysis buffer you use 400mM NaCl - might this slightly reduce the VAPB-FFAT interaction? Worth considering reducing this?

Indeed, we optimized conditions for these experiments. 400 mM is only used for lysis; the buffer is then diluted 3.75 times for the binding experiments.

- "The rather small difference observed between the wild-type and the mutant protein observed in this experiment probably results from the presence of endogenous VAPB in the stable cell

lines, which could form dimers with the exogeneous HA-tagged versions." If this is the case then please demonstrate that this is happening, or use the KO approach in the major points above. We repeated these experiments and provide better quantifications now to make our point (novel Figure 7). The sentence cited above was removed.

- "we now show that the proteins can indeed interact with each other, without the need for additional bridging factors (Figs. 1 and 3)." You show that the peptides can bind - but this is not the same thing as the peptide in the full context of the protein - so this should be toned down or removed.

We agree and now used larger protein fragments of ELYS (full-length ELYS could not be expressed, as mentioned above). Of course, the IP-experiments (Figure 5B-C) could still reflect an indirect interaction; the purified fragments, however, rather argue in favor of direct binding of VAPB to ELYS (Figure 5E-F).

- "Remarkably, this region is highly conserved between species, suggesting that it is important for protein functions (data not shown)". Please show the alignments so the reader can judge for themselves. It is conserved in ALL species and the phosphosites are also conserved??

The region and the phosphosites are conserved in vertebrates. We show these data now in Figure EV4D.

- "In our experiments, knockdown of VAPA alone did not lead to a delay in mitosis (data not shown). " Why not show this data - as this is a very interesting and potentially important observation? Also add the validation of knockdown of VAPA.

We include these data now, as suggested. Our new Fig. EV4A-B show a minor delay in the VAPA-knock down cells, compared to the VAPB-knock down cells.

- I find the end to the discussion to the paper rather abrupt. It would be interesting to discuss further how VAPB, but not apparently VAPA reaches the INM and if so why this function is required of an ER adaptor and not another more obvious adaptor protein. In short - why would VAPB be performing this role?

[REDACTED: The author's response with unpublished data]. VAPB as an ER protein could become relevant during mitosis in particular. We restructured the discussion and tried to better explain these points.

****Referees cross-commenting****

I agree with the comments of the other reviewers, and they are very much in line with my own review. We all seem convinced that VAPB binds ELYS via a pFFAT, and that this interaction is enhanced during mitosis. However the role of this interaction in mitotic progression remains unclear and based on this data should not be claimed in the title or discussion of the paper.

Reviewer #1 (Significance (Required)):

Overall, if the manuscript could be improved with the suggested changes, then this could be a considerable conceptual advance in how we understand the VAP proteins, showing functions beyond those as an ER adaptor. This would be significant for the field.

In the context of the existing literature the work does not advance our knowledge of FFAT-VAP interactions, this has already been shown, but it would give a nice example of how this can be regulated during mitosis and how VAP can contribute beyond just as an ER adaptor at membrane contact sites.

There would be a wide audience in the cell biology field and more widely as mutations in VAPB cause a form of ALS, and many people are working in this area.

My field of expertise is in organelle cell biology and membrane contact sites.

Reviewer #2 (Evidence, reproducibility and clarity (Required)):

Summary:

In this study, James et al. follow up on their prior discovery that the ER contact site protein VAPB localizes to the nuclear envelope and is a putative binding partner of the nucleoporin ELYS, which coordinates nuclear envelope reformation (NER) with nuclear pore complex (NPC) biogenesis at mitotic exit and is also a constituent of the nuclear facing Y-complexes of mature NPCs. Using a series of complementary biochemical approaches the authors 1) demonstrate that VAPB and ELYS directly interact, 2) map the binding sites on ELYS that are sufficient to bind VAPB, 3) show that mutations that disrupt VAPB-FFAT motif binding also abrogate binding to ELYS including of the full-length protein, 4) define mitotic phosphorylation sites on VAPB-bound ELYS, 5) demonstrate that phosphorylation of ELYS, specifically at the FFAT2 motif, is required for binding to VAPB, and 6) demonstrate that the phosphorylation of ELYS that regulates VAPB binding occurs in mitosis. Turning to cell biology, the authors find that VAPB, which is an established ER protein, has some preference for non-core regions during NER (like ELYS). In addition, PLA analysis suggests that the interaction of VAPB and ELYS is most robust during anaphase and is somewhat disrupted when the binding of VAPB to FFAT motifs is lost due to targeted mutation. Last, the authors demonstrate that depletion of VAPB leads to metaphase delay and lagging chromosomes.

Major comments:

The data supporting direct binding of peptides encoding the FFAT 1 and FFAT2 motif derived from ELYS to VAPB in a manner similar to other FFAT sequences is strong, as is the effect of phosphorylation of FFAT1 on the strength of this interaction.

The evidence supporting the mitotic-specific nature of the ELYS-VAPB interaction is strong, and that this interaction is direct, is also strong, and was rigorously tested using a combination of endogenous expression, heterologous expression, and recombinant protein approaches.

Moreover, the sensitivity of this interaction to established mutations in VAPB abrogating FFAT interactions reinforces the outlined underlying biochemical interaction mechanism. The essentiality of ELYS phosphorylation (and therefore the mechanism underlying the mitotic specificity of the interaction) is also strongly supported by the data using phosphatase treatment.

Although it is an intuitive model, whether the cell biological evidence support the simplest view that the ELYS-VAPB complex bridges the nuclear envelope to chromatin during NER in late anaphase / at mitotic exit is far less solid and, at a minimum, alternative models should be considered/discussed.

For example, how a delay in metaphase in the siVAPB condition is consistent with a role in NER, which occurs exclusively post-metaphase, is unclear.

Is it not possible that the VAPB-ELYS complex is regulated by phosphorylation during mitotic progression such that VAPB and/or ELYS can only exert their biological effects when released from the complex?

In other words, might ELYS be licensed to act in NER only when it is released from VAPB, which could prevent premature NER/NPC biogenesis?

Subtleties of when during mitosis the phosphorylation occurs is challenging, and it could be that the anaphase A to anaphase B transition, when many mitotic entry phosphorylation events begin to be reversed, could be relevant here.

We thank the reviewer for the comments and suggestions. Definitely, alternative explanations for the observed effects are possible – and they are not mutually exclusive. We addressed these issues in our modified discussion.

Along these lines, in Fig. 5 how the VAPB knock-down does or does not recapitulate the phenotype of an ELYS knock-down in this cell type (and the effect of the combination, to address epistasis) is needed for context, as is whether VAPB knock-down affects ELYS distribution in mitosis. ELYS knock-down would also be very beneficial for the PLA analysis to establish the "floor" of measurable signal.

ELYS knock down has strong effects on mitosis (Rasala et al., 2006), whereas we observed comparatively mild effects upon VAPB knockdown (Figure 8). VAPB knock down did not affect the distribution of ELYS (Figure EV2A). ELYS knock down cells for PLAs are also shown in Fig. EV2B.

Last, it is also possible that VAPB has other roles in mitosis that should be acknowledged - for example although it is in yeast, it is relevant that a VAPB orthologue Scs2 is required for normal nuclear envelope expansion in mitosis by regulating SUMOylation (Ptak, Saik et al., JCB, 2021 and Saik et al., JCB, 2023) - this work should be referenced as well.

We now cite those important publications, including the very recent one by Saik and coworkers.

Of course, the ideal experiment would be one in which an ELYS knock-down is complemented with a resistant form that encodes the S to A mutations in the FFAT2 region to assess its

localization and to see if it can complement the knock-out function of ELYS in post-mitotic NPC assembly or, as suggested by a sequestration model, it can drive the same metaphase delay seen upon VAPB knock-down. This is technically challenging for sure, particularly given the size of the ELYS gene, but it would address the cell biological function of this interaction in the most direct manner.

This would be the ideal experiment, indeed. As mentioned above, we tried to express full-length ELYS – and did not succeed so far.

Several other observations that could warrant further comment or study include 1) is there a VAPB signal at the metaphase poles as suggested by Fig. 4A and, if so, could this represent a distinct mitotic function?;

Indeed, we think that these are ER-cisternae (Lu et al., 2011). This is now mentioned in the text.

2) Does the HA-VAPB KD/MD mutant localize differently in mitosis compared to the WT - it appears that it might be less enriched in non-core regions (Fig. 4E)?;

We think there is no significant difference between the two versions of VAPB (and mention this now in the text).

3) does VAPB alter post-mitotic NPC biogenesis/number?

We did not detect significant differences using STED-microscopy. These data are now presented in Appendix S3.

Minor comments:

I would suggest avoiding the use of "novel" when describing newly assembled NPCs or post-mitotic nuclear envelope reformation, as its other meaning of "non-standard" makes this wording confusing.

It is unclear whether when the authors state that ELYS localizes "to the nuclear side of the nuclear envelope" they are referring to the nuclear aspect of the NPC and/or a separate pool at the INM - please edit to clarify.

We changed this according to the suggestions.

More descriptive y-axes for the plots in Fig. 4F and 5F and related legends would be useful; although the details are in the methods section, it would be nice not to have to hunt them down. Also, please clarify the meaning of blue and orange points in Fig. 4F.

We changed this according to the suggestions (novel figure 7B and 8).

Reviewer #2 (Significance (Required)):

General assessment: The biochemical analysis is rigorous and compelling and establishes the mitotic-specific interaction of VAPB and ELYS including detailed information about the binding

interface and its regulation by phosphorylation. The new insight provided into the function of this VAPB-ELYS interaction is somewhat less well developed as the current manuscript, in its current form, does not yet mechanistically define the function of the VAPB-ELYS interaction in mitosis.

Advance: Conceptually, to the best of my knowledge, the idea that VAPB contributes to mitosis in mammalian cells is novel and is therefore impactful and will motivate further work. As the authors connect VAPB biochemically to ELYS, an established factor that promotes the coordination of NER and NPC biogenesis, this interaction is likely to be mechanistically important, although the specific details by which this interaction facilitate normal mitotic progression is not yet clear.

Audience: This work will be of interest to a broad swath of cell biologists including those interested in NPCs, the nuclear envelope, the ER, membrane remodeling, and chromosome segregation.

My expertise is in nuclear envelope dynamics, nuclear pore complexes, and chromatin organization.

Reviewer #3 (Evidence, reproducibility and clarity (Required)):

This MS contains carefully carried out and well controlled experiments describing a new pFFAT in ELYS. There is a similarly convincing demonstration of functionally relevant colocalisation by proximity ligation assay (PLA), particularly that both ELYS and VAP are nuclear envelope proteins in interphase without interacting (neg control in Fig 4D).

Major Issue: Functional significance

A key conclusion is that experiments prove that "ELYS serves as the crucial initiation factor for post-mitotic NPC-assembly" (p5). However, evidence for this is lacking as this would require reconstitution of NPC assembly with a mutant form of ELYS carefully changing the FFAT motif (e.g. 1321A 1324E) and exclusion of other probable VAP targets in experiments with mutant VAP.

Our statement refers to previous publications. We try to better explain this now.

VAPs are among the proteins with the highest number of documented interactors (see Huttlin 2015/7 etc, e.g. PMID 26186194), so knocking down VAP may have pleiotropic effects and quite indirect read-outs in many aspects of cell function.

This is true, of course, and we detect a large number of interactors in our novel Figure 1. We addressed this issue in detail in our new discussion.

In addition, for this work specifically there are other NE proteins that are known interactors of VAP: Emerin (EMD) and LBR both interact with VAP (high-throughput data, VAPA and VAPB).

EMD has a motif similar to the canonical phospho-FFAT: 98 SYFTTRT 104. LBR has no motif. These findings should not be overlooked in this work. For example, was the interaction with emerin (page 4) sensitive to mutating VAP or ELYS? Could the effect seen in Figure 5 result from interactions with proteins other than ELYS?

This is of course a very valid point. In our MS-dataset (novel Figure 1), however, we did not detect emerin or LBR as interaction partners of the MSP-domain of VAPB. Of course, this does not exclude the possibility that the proteins interact via different regions, possibly also supported by the transmembrane domain of VAPB. Indeed, emerin and LBR are in close proximity to ELYS as shown by PLAs (Figure 7A-B) – but this proximity seems to be independent of the MSP-domain of VAPB (or at least of the mutations (KD/MD) introduced at one of the critical sites; Figure 7A-B).

Further experiments should be carried out to justify all statements in the current MS of functional significance. Instead of doing more experiments, an alternative for the authors would be to describe the current set of results more cautiously. However, that would require changing much of the impact of the current MS, from the title onwards.

See comments to reviewers #1 and #2.

Moderate Issue: VAPA

From the start of the Introduction and some elements of the Discussion, include VAPA in equal measure with VAPB. When describing interactions of ELYS with VAP note that Huttlin et al., reported interactions twice for each of VAPA and VAPB. When describing own results (James et al. 2019) and those of others (Saiz-Ros et al., 2019) that focused on VAPB, clarify if the authors' view is that VAPA would (or would not) have the same interaction.

Is there any evidence that only VAPB is on NE?

In general, VAPA is considered to have very similar interaction partners as VAPB. [REDACTED: The author's response with unpublished data]. Indeed, VAPA can also be found at the nuclear envelope as well. However, depletion of VAPA did not have a strong effect on mitotic progression. We will mention this in the results section (Fig. EV4) and also in the discussion.

Note that some refs in the Introduction relate to VAPA: Mesmin (not VAPB); ACBD5: although article titles refer to VAPB, early work (10.1083/jcb.201607055) showed almost identical involvement of VAPA.

We modified this part of the introduction accordingly.

Also, this redundancy likely explains "function of VAPB in mitosis is not essential," (in Discussion). The lack of effect of VAPA knock-down may indicate that in these cells VAPB is dominant, but does not exclude a role for VAPA when VAPB is reduced. That might be tested by depleting both. Even following that, there is MOSPD2 to consider

We did perform the suggested experiment (depleting VAPB and VAPA), but did not observe a stronger effect compared to depleting VAPB alone. In the discussion, we addressed these issues (including MOSPD2) again.

Other aspects of the writing

"two amino acid residues are crucial for the interaction (VAPB K87 and M89)." This is wrong. Many residues are critical, these are merely 2 of possibly >10 that were chosen by Kaiser et al (2005) to create their non-binder.. Others have used different mutations to block FFAT binding.

We changed the text accordingly.

"They may exhibit a certain binding preference to specific members of the VAP ... family...". I cannot think of any example. I note no citation is given.

We add a citation here Cabukusta et al., 2020.

When listing many or all MSP proteins, the text should state that MOSPD2 is uniquely close to VAPA/B. CFAP65 is typically not mentioned in the VAP-like lists as it does not have any of the conserved sequence that binds FFAT. If however the authors wish to include all human MSP domain protein, they should also include Hydin.

We changed the introduction accordingly (and do not mention CFAP65 anymore).

Slightly wrong to cite De Vos et al., 2012 about PTPIP51's FFAT as that paper makes no mention of the motif. Better pick Di Mattia (again)

We changed these citations, as suggested.

On VAPB (and also A) on INM: there are references to be cited esp. relating to intranuclear Scs2 in yeast (Brickner et al 2004, Ptak et al 2021)

We added this reference.

Citations for VAP at ER-mito contacts "De Vos et al., 2012; Gómez-Suaga et al., 2019; Stoica et al., 2014)". These all refer to the same bridging protein, PTPIP51. Reduce to one citation.

Done

Then mention other proteins at the same site VPS13A, mitoguardin(MIGA)-2 ...

We added a reference for MIGA-2.

"The domain interacts with characteristic peptide sequences ..." add citation to this sentence

We added an appropriate reference.

"Several variants of such motifs have been described: (i)" ... "(ii)": (i) and (ii) are entirely unlinked. Delete these and also "Several variants of such motifs have been described." Which is repeated later

Done

"FFAT-like motifs come in different flavors and may even lack the two phenylalanine residues (Murphy and Levine, 2016)": while motifs can tolerate variation at both positions, this text is misleading as it implies much more variation than is known. The 1st F can only be conservatively substituted (Y).

We thank the reviewer for bringing those issues to our attention (we are still rather new to this field and we are still learning...).

Minor aspects in Results:

ORP1L peptide as positive control: cite Kaiser 2005

We added this reference.

Was phosphoproteomics done in such a way as to find peptides that have both S1314 and S1326?

See comments to reviewers #1 and #2.

Figure 4D, row 2: Comment on intranuclear staining in Prophase (at approx 4 o'clock) of both ELYS & VAP that is PLA positive

This is now Figure 6D. This signal could result from ER-cisternae (Lu et al., 2011). We mention this in the legend now.

Referees cross-commenting

I agree with this point from Reviewer #1. We all agree that the main issue can be resolved experimentally to determine the effect of subtle point mutations in ELYS. Both other reviewers have done a good job in finding issues with the experiments that can also be addressed.

Reviewer #3 (Significance (Required)):

This work documents an interaction between the protein ELYS, that is involved in the reformation of nuclear pore complexes after mitosis, and the ER membrane protein VAPB. The interactions was previously known through high-throughput studies, along with many 100's of others for VAP, but here it is studied in detail and with care, identifying how the motif is induced by phosphorylation of ELYS. The two proteins are co-localised using convincing proximity ligation assays. This biochemistry and cell biological localisation is well done.

Functional experiments then show that VAP (in this case VAPB) knock-down affects mitosis and chromosome segregation. While the result is incontrovertible, it has many possible interpretations, mainly because VAP has hundreds of interactions, including with multiple proteins involved in mitosis beyond just ELYS. This means that there are major limitations on how the interaction and co-localisation should be interpreted, reducing the advance associated with the current manuscript to incremental, and the limiting the audience to specialized.

We agree. The large number of known VAPB-interactors make it very difficult to interpret any phenotype associated with VAPB-depletion. We tried to make the point, however, that ELYS is the only protein described so far that becomes particularly prominent as an interaction partner of VAPB in mitosis (and in particular at the molecular detail presented in our manuscript). Nevertheless, we are now very careful in our interpretations.

Thank you very much again for handling this manuscript and the very helpful comments.

Sincerely

Ralph Kehlenbach

List of changes in figures and tables:

Figure 1

- Figure 3A is now figure 1A and replaced the Coomassie gel image
- added new figures 1B-E

Figure 2

- added new plot B with quantification of ELYS protein level
- added new plot F with quantification of ELYS binding

Figure 3

- Figure 1 is now figure 3 (no changes)

Figure 4

- Figure 3 is now figure 4. We added a more descriptive figure legend for panel F

Figure 5

Figure 5 is new

Figure 6

- Figure 4 is now figure 6 A-D; C has a more details

Figure 7

- Figure 4E, F were modified and are now in Figure 7. We also changed the y-axis title for plots in B

Figure 8

- Figure 8A is new. Other panels are from old figure 5 (no further changes).

- new Figure EV1
- new Figure EV2
- new Figure EV3
- new Figure EV4

- new Appendix Figure S1
- new Appendix Figure S2
- new Appendix Figure S3

- new Dataset Tables EV 1-5
- updated Appendix tables S1 and S2

Dear Prof. Kehlenbach

Thank you for the submission of your revised manuscript to EMBO reports. As I informed you last week, we have meanwhile received all referee reports and I include them again below my signature. All three referees are very positive about the study and request only minor changes.

Being back in the office, I have now also gone through the manuscript from the editorial side and there are a few things that we need before we can proceed with the official acceptance of your study.

- Please reduce the number of keywords to 5.
 - Please update the 'Conflict of interest' paragraph to our new 'Disclosure and competing interests statement' and place it after the Acknowledgement section.
For more information see
<https://www.embopress.org/page/journal/14693178/authorguide#conflictsofinterest>
 - Just a kind reminder not to forget removing the reviewer access from the Data Availability section.
 - Appendix: Please add page numbers on the title page (table of Content). The legends for each figure/table need to be removed from the manuscript as they are already provided in the Appendix.
 - Appendix Figure S3: Please add a scale bar and define its size in the figure legend. The legend contains a mismatch with the panel label (sint instead of si nt). The quantification lacks a description in the legend. Please define the bars and error bars and the nature of the replicates (biological/independent or technical).
 - Movies: the correct nomenclature for the file names, legends and callouts should be Movie EV1 and Movie EV2. The legends need to be removed from the manuscript file and each should be provided as readme.txt file, which needs to be zipped with the corresponding video and uploaded as one folder.
 - Datasets: please change the nomenclature from Table EV# to Dataset EV#. The legends are provided in a separate tab of the .xls file called Legend or Dataset EV#. Please also add the name "Dataset EV#" to the legend in the file (as you did for Table EV5).
 - The list of abbreviations need to be removed from the manuscript file. Please define all abbreviations the first time they are used in the text.
 - Our production/data editors have asked you to clarify several points in the figure legends (see below). Please incorporate these changes in the manuscript and return the revised file with tracked changes with your final manuscript submission.
- A) Please note that a separate 'Data Information' section is required in the legends of figures 1b-c; 2d-e; 6a, c-d.
B) Please note that the error bars are not defined in the legends of figures 2b, f; 4d, f; 5c, f; 7b; 8a, e, h; EV 4b.
C) Please note that the scale bar needs to be defined for figure 7a; EV 2a-b; EV 3.
- As a standard procedure, we edit the title and abstract of manuscripts to make them more accessible to a general readership. Please find the edited suggestions below my signature.

- Finally, EMBO Reports papers are accompanied online by A) a short (1-2 sentences) summary of the findings and their significance, B) 2-3 bullet points highlighting key results and C) a synopsis image that is 550x300-600 pixels large (width x height) in PNG for JPG format. You can either show a model or key data in the synopsis image. Please note that the size is rather small and that text needs to be readable at the final size. Please send us this information along with the revised manuscript.

- On a different note, I would like to alert you that EMBO Press offers a new format for a video-synopsis of work published with us, which essentially is a short, author-generated film explaining the core findings in hand drawings, and, as we believe, can be very useful to increase visibility of the work. This has proven to offer a nice opportunity for exposure i.p. for the first author(s) of the study. Please see the following link for representative examples and their integration into the article web page:

https://www.embopress.org/video_synopses
<https://www.embopress.org/doi/full/10.15252/emj.2019103932>

With kind regards,

Abstract and title:

Phosphorylation of ELYS promotes its interaction with VAPB at decondensing chromosomes during mitosis

ELYS is a nucleoporin that localizes to the nuclear side of the nuclear pore complex (NPC) in interphase cells. In mitosis, it serves as an assembly platform that interacts with chromatin and then with nucleoporin subcomplexes to initiate post-mitotic NPC assembly. Here we identify ELYS as a major binding partner of the membrane protein VAPB during mitosis. In mitosis, ELYS becomes phosphorylated at many sites, including a predicted FFAT (two phenylalanines in an acidic tract) motif, which mediates interaction with the MSP (major sperm protein) domain of VAPB. Binding assays using recombinant proteins or cell lysates and co-immunoprecipitation experiments show that VAPB binds the FFAT motif of ELYS in a phosphorylation-dependent manner. In anaphase, the two proteins co-localize to the non-core region of the newly forming nuclear envelope. Depletion of VAPB results in prolonged mitosis, slow progression from meta- to anaphase, and in chromosome segregation defects. Together, our results suggest a role of VAPB in mitosis upon recruitment to or release from ELYS at the non-core region of the chromatin in a phosphorylation-dependent manner.

Referee #1:

Addressing my previous comments:

I appreciate the authors' efforts to address the comments of all the reviewers. Generally their revisions make the manuscript publishable, with new experiments and text changes that attempt to address the major point of functional relevance.

I have looked at the considerable amount of new text and material, with the following MINOR comments on the MS as a whole:

Introduction

"Exchanging these amino acids to aspartic acid largely abolishes the interaction of VAPB to FFAT-dependent target proteins (Kaiser et al., 2005)."

This is oddly over-specific - the whole sentence could be dropped, with the citation added to the previous sentence.

New text page 3:

There are problems with some of this

"FFAT-like motifs come in different flavors, e.g. ... or with a neutral instead of an acidic tract (MOSPD1 and 3 binding to FFNT-motifs; (Cabukusta et al., 2020))" This is wrong - the FFNTs' are such a different flavor that they are not the same and have a different name. I'm not sure why the authors include them - it is not strictly needed, as the motifs differ significantly (NB the MS would have to explain the name in full - not just abbreviate).

"... examples for protein-protein interaction partners of VAPB are the protein tyrosine phosphatase-interacting protein-51 (PTPIP51), connecting ER- and mitochondria (Stoica et al., 2014)" I find it strange to cite this paper as it has zero mention of FFATs.

What this section should focus on instead is to describe the FFAT-like motif as an enlarged definition of the FFAT motif originally described (Loewen et al 2003, PMID: 12727870), almost every element of which turns out to be non-essential, with functionally relevant motifs that vary at multiple positions that can best be assessed by a position-specific scoring matrix (for example see PMID: 29873773).

Results

Overall: There is an excess of rhetorical words describing the results: interestingly, strikingly, remarkably (x9 in total in the MS). The biological findings should largely speak for themselves and do not need to be hyped so much.

p10 lines 5-7:

"Three FFAT motifs were predicted ... all of which lack the hydrophobic phenylalanine residues." This reads as if none of the motifs have the key required residues. However, they all have an aromatic residue at the key position 2 (2 Ys and 1 F). Note that Y2 has been accepted as a variant of F2 since the inception of these motifs, despite the fixation of the name "FFAT" on Fs (see Loewen et al, PMID: 12727870, Fig 2D).

p13 and p16:

The text should explain what the authors have told us in their rebuttal about the inability of approaches to re-express full-length ELYS. This is the place to do that, and the topic should be revisited briefly in the Discussion. If omitted then some other group will attempt the same thing only to fail, so it's a negative result worth including.

p17: "also depleted the VAPB-related protein VAPA ... (supplementary Figure EV4A and B)"

Did the authors try a double depletion? If not, at minimum discuss it and the possible result. Note: it is not right to describe VAPA like this; instead the accurate term is a (close) paralog. This applies too in the Discussion, to replace "very similar" in "VAPA, a protein very similar in sequence and localization to VAPB"

Very minor

p6 typo: motive -> motif

General grammar: commas are included many times where not needed. Should only be to separate ideas. Example "scenario," (p17).

p19: " In our assays, phosphorylation ..." this sentence refers to fig 4 not fig 3.

Referee #2:

In general the vast majority of my points have been satisfactorily addressed, or a significant attempt to address them has been made but for technical reasons this has not been possible (e.g. the phospho-MS to directly demonstrate the phosphorylation of FFA motifs in mitosis). I find the revised manuscript is significantly improved, in particular the apparent change in binding partners of VAPB to now favour ELYS binding during mitosis is much more apparent. The vast majority of the relevant controls are now included and I find the data overall more convincing. I suggest a few minor improvements and minimal further quantifications/controls to improve the newly added sections.

Overall, barring a significant raft of new experiments to further probe the role of the VAPB-ELYS interaction and the timing of the phosphorylation events (presumably the subject of another paper) I think this is highly novel work which is suitable for publication.

Abstract -

"In mitosis, ELYS becomes phosphorylated at many sites, including a predicted FFAT (two phenylalanines in an acidic tract) motif,"

I am still unclear if any proteomics data show phosphorylation of these FFAT motifs? The authors clearly attempted this experiment but due to technical reasons (as they mention in the response) could not detect this peptide. So, can they confidently say that this happens? If not then this direct statement should be removed/toned down for clarity.

Minor points:

"Tetracycline-induced expression of the wild-type-version of the ELYS-fragment, however, clearly increased this proportion from 21% to 39%. Interestingly, the S1314A-version of the fragment had a somewhat larger, the S1314D-version a somewhat smaller effect. Two other fragments (S1326A and S1326D) were rather similar to the wild-type fragment in this assay."

For the S1314A version - to me this looks very similar to WT? Are any of these differences statistically significant, saying they are "rather similar" seems a little inconclusive.

Are expression levels of all these constructs the same - this should be shown.

"VAPB-depleted cells displayed a slower progression through mitosis compared to the control cells, with a clear delay right after the release. We also depleted the VAPB-related protein VAPA, which can also localize to the INM and observed only a minor effect on mitotic progression (supplementary Figure EV4A and B)."

Are these differences statistically significant? Is only VAPB significant?

Silencing of the proteins does not appear so effective - could this impact the results. I.e. is VAPA silenced to a lesser extent than

VAPB?

Referee #3:

Overall, this revised manuscript from James et al. addresses my major concerns about the first submission that was refereed through Review Commons. Strengths of the manuscript include technically compelling evidence of a previously unknown, phospho-dependent ELYS-VAPB interaction. The new Figure 5 data provides further support for the core biochemical interaction and its sensitivity to the phosphorylation state of ELYS, which places the interaction in the context of mitosis. My other comments were addressed with changes to the text and the addition of citations. I am therefore supportive of the publication of this work, which will be of broad interest to multiple fields.

Referee #1:

Addressing my previous comments:

I appreciate the authors' efforts to address the comments of all the reviewers. Generally their revisions make the manuscript publishable, with new experiments and text changes that attempt to address the major point of functional relevance.

I have looked at the considerable amount of new text and material, with the following MINOR comments on the MS as a whole:

Introduction

"Exchanging these amino acids to aspartic acid largely abolishes the interaction of VAPB to FFAT-dependent target proteins (Kaiser et al., 2005)."

This is oddly over-specific - the whole sentence could be dropped, with the citation added to the previous sentence.

New text page 3:

There are problems with some of this

"FFAT-like motifs come in different flavors, e.g. ... or with a neutral instead of an acidic tract (MOSPD1 and 3 binding to FFNT-motifs; (Cabukusta et al., 2020))" This is wrong - the FFNTs' are such a different flavor that they are not the same and have a different name. I'm not sure why the authors include them - it is not strictly needed, as the motifs differ significantly (NB the MS would have to explain the name in full - not just abbreviate).

"... examples for protein-protein interaction partners of VAPB are the protein tyrosine phosphatase-interacting protein-51 (PTPIP51), connecting ER- and mitochondria (Stoica et al., 2014)" I find it strange to cite this paper as it has zero mention of FFATs.

What this section should focus on instead is to describe the FFAT-like motif as an enlarged definition of the FFAT motif originally described (Loewen et al 2003, PMID: 12727870), almost every element of which turns out to be non-essential, with functionally relevant motifs that vary at multiple positions that can best be assessed by a position-specific scoring matrix (for example see PMID: 29873773).

Results

Overall: There is an excess of rhetorical words describing the results: interestingly, strikingly, remarkably (x9 in total in the MS). The biological findings should largely speak for themselves and do not need to be hyped so much.

p10 lines 5-7:

"Three FFAT motifs were predicted ... all of which lack the hydrophobic phenylalanine residues." This reads as if none of the motifs have the key required residues. However, they all have an aromatic residue at the key position 2 (2 Ys and 1 F). Note that Y2 has been accepted as a variant of F2 since the inception of these motifs, despite the fixation of the name "FFAT" on Fs (see Loewen et al, PMID: 12727870, Fig 2D).

p13 and p16:

The text should explain what the authors have told us in their rebuttal about the inability of approaches to re-express full-length ELYS. This is the place to do that, and the topic should be revisited briefly in the Discussion. If omitted then some other group will attempt the same thing only to fail, so it's a negative result worth including.

p17: "also depleted the VAPB-related protein VAPA ... (supplementary Figure EV4A and B)"

Did the authors try a double depletion? If not, at minimum discuss it and the possible result. Note: it is not right to describe VAPA like this; instead the accurate term is a (close) paralog. This applies too in the Discussion, to replace "very similar" in "VAPA, a protein very similar in sequence and localization to VAPB"

Very minor

p6 typo: motive -> motif

General grammar: commas are included many times where not needed. Should only be to separate ideas. Example "scenario," (p17).

p19: " In our assays, phosphorylation ..." this sentence refers to fig 4 not fig 3.

Referee #2:

In general the vast majority of my points have been satisfactorily addressed, or a significant attempt to address them has been made but for technical reasons this has not been possible (e.g. the phospho-MS to directly demonstrate the phosphorylation of FFA motifs in mitosis). I find the revised manuscript is significantly improved, in particular the apparent change in binding partners of VAPB to now favour ELYS binding during mitosis is much more apparent. The vast majority of the relevant controls are now included and I find the data overall more convincing. I suggest a few minor improvements and minimal further quantifications/controls to improve the newly added sections.

Overall, barring a significant raft of new experiments to further probe the role of the VAPB-ELYS interaction and the timing of the phosphorylation events (presumably the subject of another paper) I think this is highly novel work which is suitable for publication.

Abstract -

"In mitosis, ELYS becomes phosphorylated at many sites, including a predicted FFAT (two phenylalanines in an acidic tract) motif,"

I am still unclear if any proteomics data show phosphorylation of these FFAT motifs? The authors clearly attempted this experiment but due to technical reasons (as they mention in the response) could not detect this peptide. So, can they confidently say that this happens? If not then this direct statement should be removed/toned down for clarity.

Minor points:

"Tetracycline-induced expression of the wild-type-version of the ELYS-fragment, however, clearly increased this proportion from 21% to 39%. Interestingly, the S1314A-version of the fragment had a somewhat larger, the S1314D-version a somewhat smaller effect. Two other fragments (S1326A and S1326D) were rather similar to the wild-type fragment in this assay."

For the S1314A version - to me this looks very similar to WT? Are any of these differences statistically significant, saying they are "rather similar" seems a little inconclusive.

Are expression levels of all these constructs the same - this should be shown.

"VAPB-depleted cells displayed a slower progression through mitosis compared to the control cells, with a clear delay right after the release. We also depleted the VAPB-related protein VAPA, which can also localize to the INM and observed only a minor effect on mitotic progression (supplementary Figure EV4A and B)."

Are these differences statistically significant? Is only VAPB significant?

Silencing of the proteins does not appear so effective - could this impact the results. I.e. is VAPA silenced to a lesser extent than VAPB?

Referee #3:

Overall, this revised manuscript from James et al. addresses my major concerns about the first submission that was refereed through Review Commons. Strengths of the manuscript include technically compelling evidence of a previously unknown, phospho-dependent ELYS-VAPB interaction. The new Figure 5 data provides further support for the core biochemical interaction and its sensitivity to the phosphorylation state of ELYS, which places the interaction in the context of mitosis. My other comments were addressed with changes to the text and the addition of citations. I am therefore supportive of the publication of this work, which will be of broad interest to multiple fields.

Rev_Com_number: RC-2023-02064

New_manu_number: EMBOR-2023-58173V2

Corr_author: Kehlenbach

Title: Phosphorylation of ELYS controls its interaction with VAPB at decondensing chromosomes during mitosis

Dear Dr. Rembold, Dear Reviewers,

Please find our comments below.

Sincerely,
Ralph Kehlenbach

- Please reduce the number of keywords to 5.

done

- Please update the 'Conflict of interest' paragraph to our new 'Disclosure and competing interests statement' and place it after the Acknowledgement section.

For more information see

<https://www.embopress.org/page/journal/14693178/authorguide#conflictsofinterest>

done

- Just a kind reminder not to forget removing the reviewer access from the Data Availability section.

done

- Appendix: Please add page numbers on the title page (table of Content). The legends for each figure/table need to be removed from the manuscript as they are already provided in the Appendix.

done

- Appendix Figure S3: Please add a scale bar and define its size in the figure legend. The legend contains a mismatch with the panel label (sint instead of si nt). The quantification lacks a description in the legend. Please define the bars and error bars and the nature of the replicates (biological/independent or technical).

done

- Movies: the correct nomenclature for the file names, legends and callouts should be Movie EV1 and Movie EV2.

done The legends need to be removed from the manuscript file and each should be provided as readme.txt file, which needs to be zipped with the corresponding video and uploaded as one folder.

done

- Datasets: please change the nomenclature from Table EV# to Dataset EV#.

done

The legends are provided in a separate tab of the .xls file called Legend or Dataset EV#. Please also add the name "Dataset EV#" to the legend in the file (as you did for Table EV5).

done

- The list of abbreviations need to be removed from the manuscript file. Please define all abbreviations the first time they are used in the text.

done

- Our production/data editors have asked you to clarify several points in the figure legends (see below). Please incorporate these changes in the manuscript and return the revised file with tracked changes with your final manuscript submission.

(changes in blue in the figure legends)

A) Please note that a separate 'Data Information' section is required in the legends of figures 1b-c; 2d-e; 6a, c-d.

done

B) Please note that the error bars are not defined in the legends of figures 2b, f; 4d, f; 5c, f; 7b; 8a, e, h; EV 4b.

done

C) Please note that the scale bar needs to be defined for figure 7a; EV 2a-b; EV 3.

done

- As a standard procedure, we edit the title and abstract of manuscripts to make them more accessible to a general readership. Please find the edited suggestions below my signature.

We made those changes to the title and the abstract. Thank you very much for the suggestions

- Finally, EMBO Reports papers are accompanied online by A) a short (1-2 sentences) summary of the findings and their significance, B) 2-3 bullet points highlighting key results and C) a synopsis image that is 550x300-600 pixels large (width x height) in PNG for JPG format. You can either show a model or key data in the synopsis image. Please note that the size is rather small and that text needs to be readable at the final size. Please send us this information along with the revised manuscript.

We will upload these files ("synopsis" and "summary sentence bullet points").

- On a different note, I would like to alert you that EMBO Press offers a new format for a video-synopsis of work published with us, which essentially is a short, author-generated film explaining the core findings in hand drawings, and, as we believe, can be very useful to increase visibility of the work. This has proven to offer a nice opportunity for exposure i.p. for the first author(s) of the study. Please see the following link for representative examples and their integration into the article web page:

<https://www.embopress.org/doi/full/10.15252/emj.2019103932>

(may be for the next manuscript...)

With kind regards,

Abstract and title:

Phosphorylation of ELYS promotes its interaction with VAPB at decondensing chromosomes during mitosis

ELYS is a nucleoporin that localizes to the nuclear side of the nuclear pore complex (NPC) in

interphase cells. In mitosis, it serves as an assembly platform that interacts with chromatin and then with nucleoporin subcomplexes to initiate post-mitotic NPC assembly. Here we identify ELYS as a major binding partner of the membrane protein VAPB during mitosis. In mitosis, ELYS becomes phosphorylated at many sites, including a predicted FFAT (two phenylalanines in an acidic tract) motif, which mediates interaction with the MSP (major sperm protein) domain of VAPB. Binding assays using recombinant proteins or cell lysates and co-immunoprecipitation experiments show that VAPB binds the FFAT motif of ELYS in a phosphorylation-dependent manner. In anaphase, the two proteins co-localize to the non-core region of the newly forming nuclear envelope. Depletion of VAPB results in prolonged mitosis, slow progression from meta- to anaphase, and in chromosome segregation defects. Together, our results suggest a role of VAPB in mitosis upon recruitment to or release from ELYS at the non-core region of the chromatin in a phosphorylation-dependent manner.

Referee #1:

Addressing my previous comments:

I appreciate the authors' efforts to address the comments of all the reviewers. Generally their revisions make the manuscript publishable, with new experiments and text changes that attempt to address the major point of functional relevance.

I have looked at the considerable amount of new text and material, with the following MINOR comments on the MS as a whole:

Introduction

"Exchanging these amino acids to aspartic acid largely abolishes the interaction of VAPB to FFAT-dependent target proteins (Kaiser et al., 2005)."

This is oddly over-specific - the whole sentence could be dropped, with the citation added to the previous sentence.

We changed this, as suggested.

New text page 3:

There are problems with some of this

"FFAT-like motifs come in different flavors, e.g. ... or with a neutral instead of an acidic tract (MOSPD1 and 3 binding to FFNT-motifs; (Cabukusta et al., 2020))" This is wrong - the FFNTs' are such a different flavor that they are not the same and have a different name. I'm not sure why the authors include them - it is not strictly needed, as the motifs differ significantly (NB the MS would have to explain the name in full - not just abbreviate).

We modified this, as suggested.

"... examples for protein-protein interaction partners of VAPB are the protein tyrosine phosphatase-interacting protein-51 (PTPIP51), connecting ER- and mitochondria (Stoica et al., 2014)" I find it strange to cite this paper as it has zero mention of FFATs.

We cite another paper now.

What this section should focus on instead is to describe the FFAT-like motif as an enlarged definition of the FFAT motif originally described (Loewen et al 2003, PMID: 12727870), almost every element of which turns out to be non-essential, with functionally relevant motifs that vary at multiple positions that can best be assessed by a position-specific scoring matrix (for example see PMID: 29873773).

We modified this and added Loewen et al. as a reference.

Results

Overall: There is an excess of rhetorical words describing the results: interestingly, strikingly,

remarkably (x9 in total in the MS). The biological findings should largely speak for themselves and do not need to be hyped so much.

We removed many of those...

p10 lines 5-7:

"Three FFAT motifs were predicted ... all of which lack the hydrophobic phenylalanine residues." This reads as if none of the motifs have the key required residues. However, they all have an aromatic residue at the key position 2 (2 Ys and 1 F). Note that Y2 has been accepted as a variant of F2 since the inception of these motifs, despite the fixation of the name "FFAT" on Fs (see Loewen et al, PMID: 12727870, Fig 2D).

We modified this, as suggested.

p13 and p16:

The text should explain what the authors have told us in their rebuttal about the inability of approaches to re-express full-length ELYS. This is the place to do that, and the topic should be revisited briefly in the Discussion. If omitted then some other group will attempt the same thing only to fail, so it's a negative result worth including.

We mentioned this in the text now (p. 13).

p17: "also depleted the VAPB-related protein VAPA ... (supplementary Figure EV4A and B)"

Did the authors try a double depletion? If not, at minimum discuss it and the possible result. Note: it is not right to describe VAPA like this; instead the accurate term is a (close) paralog. This applies too in the Discussion, to replace "very similar" in "VAPA, a protein very similar in sequence and localization to VAPB"

We tried the double depletion (and mention this in the text now). The efficiency, however, was low for the double knockdown.

Very minor

p6 typo: motive -> motif

done

General grammar: commas are included many times where not needed. Should only be to separate ideas. Example "scenario," (p17).

We tried to do a better job here and removed many of those commas...

p19: "In our assays, phosphorylation ..." this sentence refers to fig 4 not fig 3.

We changed this accordingly.

Referee #2:

In general the vast majority of my points have been satisfactorily addressed, or a significant attempt to address them has been made but for technical reasons this has not been possible (e.g. the phospho-MS to directly demonstrate the phosphorylation of FFA motifs in mitosis). I find the revised manuscript is significantly improved, in particular the apparent change in binding partners of VAPB to now favour ELYS binding during mitosis is much more apparent. The vast majority of the relevant controls are now included and I find the data overall more convincing. I suggest a few minor improvements and minimal further quantifications/controls to improve the newly added sections.

Overall, barring a significant raft of new experiments to further probe the role of the VAPB-ELYS interaction and the timing of the phosphorylation events (presumably the subject of another paper) I think this is highly novel work which is suitable for publication.

Abstract -

"In mitosis, ELYS becomes phosphorylated at many sites, including a predicted FFAT (two

phenylalanines in an acidic tract) motif,"

I am still unclear if any proteomics data show phosphorylation of these FFAT motifs? The authors clearly attempted this experiment but due to technical reasons (as they mention in the response) could not detect this peptide. So, can they confidently say that this happens? If not then this direct statement should be removed/toned down for clarity.

The relevant site (S1314) was identified in Ullah et al, 2016 (mentioned on pages 11 and 12).

Minor points:

"Tetracycline-induced expression of the wild-type-version of the ELYS-fragment, however, clearly increased this proportion from 21% to 39%. Interestingly, the S1314A-version of the fragment had a somewhat larger, the S1314D-version a somewhat smaller effect. Two other fragments (S1326A and S1326D) were rather similar to the wild-type fragment in this assay."

For the S1314A version - to me this looks very similar to WT? Are any of these differences statistically significant, saying they are "rather similar" seems a little inconclusive.

We mention a p-value (0,0231) now in the legend to Figure 8A (difference between S1314A and S1314D).

Are expression levels of all these constructs the same - this should be shown.

These constructs express to very similar levels. This is shown in Figure 5E.

"VAPB-depleted cells displayed a slower progression through mitosis compared to the control cells, with a clear delay right after the release. We also depleted the VAPB-related protein VAPA, which can also

localize to the INM and observed only a minor effect on mitotic progression (supplementary Figure EV4A and B)."

Are these differences statistically significant? Is only VAPB significant?

Only VAPB is significant.

Silencing of the proteins does not appear so effective - could this impact the results. I.e. is VAPA silenced to a lesser extent than VAPB?

Silencing levels were similar for both proteins (yet indeed not complete, see Figure EV4). We mention this in the text now.

Referee #3:

Overall, this revised manuscript from James et al. addresses my major concerns about the first submission that was refereed through Review Commons. Strengths of the manuscript include technically compelling evidence of a previously unknown, phospho-dependent ELYS-VAPB interaction. The new Figure 5 data provides further support for the core biochemical interaction and its sensitivity to the phosphorylation state of ELYS, which places the interaction in the context of mitosis. My other comments were addressed with changes to the text and the addition of citations. I am therefore supportive of the publication of this work, which will be of broad interest to multiple fields.

Prof. Ralph Kehlenbach
Universitätsmedizin Göttingen
Molecular Biology
Humboldtallee 23
Göttingen 37073
Germany

Dear Ralph,

I am very pleased to accept your manuscript for publication in the next available issue of EMBO reports. Thank you for your contribution to our journal and congratulations on the successful publication of your work.

Kind regards,

Martina

Rev_Com_number: RC-2023-02064

New_manu_number: EMBOR-2023-58173V3

Corr_author: Kehlenbach

Title: Phosphorylation of ELYS promotes its interaction with VAPB at decondensing chromosomes during mitosis